# Universal Neural Optimal Transport

Jonathan Geuter [1 2 *]   Gregor Kornhardt [3 *]   Ingimar Tomasson [3 *]   Vaios Laschos [4]

## Abstract

Optimal Transport (OT) problems are a corner-stone of many applications, but solving them is computationally expensive. To address this problem, we propose UNOT (Universal Neural Optimal Transport), a novel framework capable of accurately predicting (entropic) OT distances and plans between discrete measures for a given cost function. UNOT builds on Fourier Neural Operators, a universal class of neural networks that map between function spaces and that are discretization-invariant, which enables our network to process measures of variable resolutions. The network is trained adversarially using a second, generating network and a self-supervised bootstrapping loss. We ground UNOT in an extensive theoretical framework. Through experiments on Euclidean and non-Euclidean domains, we show that our network not only accurately predicts OT distances and plans across a wide range of datasets, but also captures the geometry of the Wasserstein space correctly. Furthermore, we show that our network can be used as a state-of-the-art initialization for the Sinkhorn algorithm with speedups of up to $7.4\times$, significantly outperforming existing approaches.

## 1. Introduction

Optimal Transport (Villani, 2009; Peyré & Cuturi, 2019) plays an increasing role in various areas in machine learning, such as domain adaptation (Courty et al., 2017), single-cell genomics (Schiebinger et al., 2019), imitation learning (Dadashi et al., 2020), imaging (Schmitz et al., 2018), dataset adaptation (Alvarez-Melis & Fusi, 2021), and signal processing (Kolouri et al., 2017). Oftentimes, an entropic

*Equal contribution [1]Harvard John A. Paulson School of Engineering and Applied Sciences [2]Kempner Institute at Harvard University [3]Department of Mathematics, Technische Universität Berlin, Germany [4]Weierstrass Institute, Berlin, Germany. Correspondence to: Jonathan Geuter <jonathan.geuter@gmx.de>.

*Proceedings of the 42nd International Conference on Machine Learning*, Vancouver, Canada. PMLR 267, 2025. Copyright 2025 by the author(s).

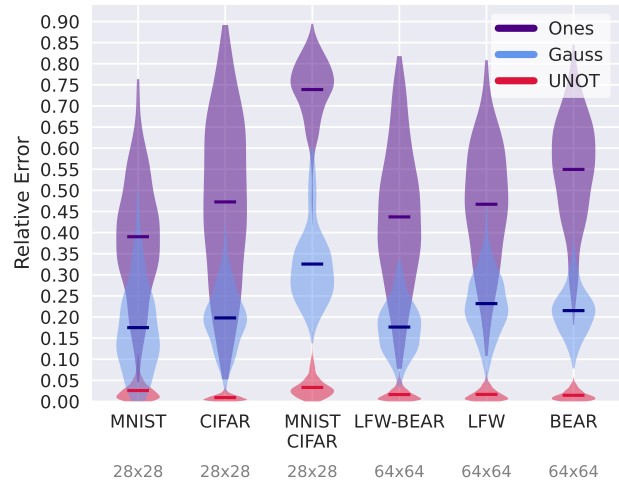

*Figure 1.* Errors on the OT distance after *a single* Sinkhorn iteration for the default initialization (Ones), the Gaussian one (Thornton & Cuturi, 2022), and ours (UNOT), for $c(\boldsymbol{x}, \boldsymbol{y}) = \|\boldsymbol{x} - \boldsymbol{y}\|^2$.

regularizer is added, as this allows for efficient computation of the solution via the Sinkhorn algorithm (Cuturi, 2013). The entropic OT problem between probability measures $\mu \in \mathcal{P}(\mathcal{X})$, $\nu \in \mathcal{P}(\mathcal{Y})$ on Polish spaces $\mathcal{X}, \mathcal{Y}$, given a cost function $c : \mathcal{X} \times \mathcal{Y} \to \mathbb{R} \cup \{\infty\}$, is defined as

$$\mathrm{OT}_\epsilon(\mu, \nu) = \inf_{\pi \in \Pi(\mu, \nu)} \int_{\mathcal{X} \times \mathcal{Y}} c \, \mathrm{d}\pi - \epsilon KL(\pi \| \mu \otimes \nu), \quad (1)$$

where $\Pi(\mu, \nu)$ is the set of all *transport plans* (i.e. measures on $\mathcal{X} \times \mathcal{Y}$ that admit $\mu$ resp. $\nu$ as their marginals), $KL(\pi \| \mu \otimes \nu) = \int \log(\pi / \mu \otimes \nu) \mathrm{d}\pi$ is the KL divergence of $\pi$ from $\mu \otimes \nu$, and $\epsilon > 0$ is a regularizing coefficient.[1]

Many of these applications require solving problem (1) repeatedly, such as in single-cell perturbations (Bunne et al., 2022; 2023b;a), Natural Language Processing (Xu et al., 2018), flow matching with OT couplings (Tong et al., 2024; Pooladian et al., 2023), or even seismology (Engquist & Froese, 2013). However, solving OT problems is computationally expensive, and fast approximation methods are an active area of research. Variations of the transport problem, such as (generalized) sliced Wasserstein distances (Kolouri et al., 2015; 2019), reduce computational complexity at

---

[1]For a background on OT, see Appendix A.

the cost of accuracy via random projections. Two previous works are aimed at predicting good initializations for the Sinkhorn algorithm, which can iteratively solve problem (1). In (Thornton & Cuturi, 2022), initializations are computed from OT problems between Gaussians. (Amos et al., 2023) train a neural network to predict transport plans and costs via the entropic *dual OT problem* (Section 2). While their framework shares some similarities with ours (see Section 5), it is inherently limited to measures of fixed dimension from the training dataset. Instead, we present a *Universal Neural OT* (UNOT) solver which, given discrete measures $\mu \in \mathcal{P}(\mathcal{X})$ and $\nu \in \mathcal{P}(\mathcal{Y})$ of variable resolution (viewed as *discretizations* of continuous measures; see Section 3.1), can accurately predict the OT cost and plan associated with problem (1). To this end, we leverage Fourier Neural Operators (FNOs) (Kovachki et al., 2024), a discretization-invariant class of neural networks that can process inputs of variable sizes. An FNO $S_\phi$ is trained to predict a solution to the dual OT problem (Section 2) given two measures $(\mu, \nu)$, from which the primal problem (1) can be solved. Training is self-supervised with an adversarial *generator* network $G_\theta$ which creates training distributions $G_\theta(z) = (\mu, \nu)$ from $z \sim \rho_z = \mathcal{N}(0, I)$ (see Section 3.4). We want to highlight that in contrast to most neural OT frameworks, such as (Bunne et al., 2023a; Uscidda & Cuturi, 2023; Korotin et al., 2023), we *generalize across OT problems* (given a fixed cost). GeONet (Gracyk & Chen, 2024) also uses Neural Operators to learn Wasserstein geodesics. In Section 4, we show that UNOT significantly outperforms GeONet on approximating geodesics, despite not being trained on them.

**Our contributions are as follows:**

- present UNOT, the first neural OT solver capable of generalizing across datasets and input dimensions

- introduce a generator $G_\theta$ (Section 3.3) which can provably generate any discrete distribution (of fixed dimension) during training (in fact, we prove this result for a very general class of residual networks, see Theorem 3 and Corollary 4)

- propose a self-supervised bootstrapping loss which provably minimizes the loss against the ground truth dual potentials (Proposition 5)

- show that UNOT can accurately predict OT distances across various datasets, costs, and domains of different dimensions up to a few percent error, and that it accurately captures the geometry of the Wasserstein space by approximating barycenters (Section 4)

- approximate Wasserstein geodesics through barycenters and OT plans predicted by UNOT (Section 4)

- demonstrate how UNOT sets a new state-of-the-art for initializing the Sinkhorn algorithm while maintaining its desirable properties, such as parallelizability and differentiability (Section 4)

## 2. Background

We give a brief overview of optimal transport, and how it relates to UNOT. For a more thorough introduction, see Appendix A.1 or (Peyré & Cuturi, 2019; Villani, 2009).

**Notation.** We write vectors in bold ($x$, $u$, etc.) and matrices in capitals ($X$, $U$, etc.). $1_n \in \mathbb{R}^n$ denotes the all-ones vector, $\Delta^{n-1}$ the simplex in $\mathbb{R}^n$, and all elements in $\Delta^{n-1}$ with positive entries are denoted by $\Delta_{>0}^{n-1}$. $\mathcal{P}(\mathcal{X})$ denotes the space of probability measures on $\mathcal{X}$, and $\mathcal{P}_p(\mathcal{X})$ the set of probability measures with finite $p$-th moments. For $\mu \in \mathcal{P}(\mathcal{X})$ and a map $T$, we denote by $T_\# \mu$ the *pushforward* of $\mu$ under $T$, i.e. the measure $\mu \circ T^{-1}$. $S^2 = \{x \in \mathbb{R}^3 : \|x\| = 1\}$ is the unit sphere in $\mathbb{R}^3$.

### 2.1. Optimal Transport

**Unregularized Optimal Transport.** The unregularized problem takes the form $\inf_{\pi \in \Pi(\mu,\nu)} \int_{\mathcal{X} \times \mathcal{Y}} c(x,y) \mathrm{d}\pi(x,y)$, akin to (1) without the regularization term. In the case where $\mathcal{X} = \mathcal{Y}$, $c(x,y) = d(x,y)^p$ for $p \geq 1$ and a metric $d$ on $\mathcal{X}$, the *Wasserstein-$p$ distance* is defined as

$$W_p(\mu,\nu) = \inf_{\pi \in \Pi(\mu,\nu)} \left( \int_{\mathcal{X} \times \mathcal{X}} d(x,y)^p \mathrm{d}\pi(x,y) \right)^{\frac{1}{p}}, \quad (2)$$

which is indeed a distance on the space $\mathcal{P}_p(\mathcal{X})$ of Borel measures with finite $p$-th moments (Villani, 2009).

**Dual Optimal Transport Problem.** The regularized Kantorovich problem (1) admits a dual formulation:

$$\sup_{f \in L^1(\mu), g \in L^1(\nu)} \int_{\mathcal{X}} f(x) \mathrm{d}\mu(x) + \int_{\mathcal{Y}} g(y) \mathrm{d}\nu(y) - \imath^\epsilon(f,g), \quad (3)$$

where

$$\imath^\epsilon(f,g) = \epsilon \int_{\mathcal{X} \times \mathcal{Y}} e^{\frac{1}{\epsilon}(f(x)+g(y)-c(x,y))} - 1 \mathrm{d}\mu(x) \mathrm{d}\nu(y).$$

It can be shown that if $c \in L^1(\mu \otimes \nu)$, the values of the primal (1) and the dual (3) coincide (Nutz, 2022).

**Discrete Optimal Transport.** We want to apply UNOT to *discretizations* of measures $\mu \in \mathcal{P}(\mathcal{X})$, $\nu \in \mathcal{P}(\mathcal{Y})$ (see Section 3.1). To this end, consider measures $\mu$ and $\nu$ that are supported on finitely many points $x_1, ..., x_m \in \mathcal{X}$, $y_1, ..., y_n \in \mathcal{Y}$ resp. i.e. $\mu = \sum_{i=1}^m a_i \delta_{x_i}$, $\nu = \sum_{j=1}^n b_j \delta_{y_j}$. By abusing notation, we can write $\mu \in \mathbb{R}_{\geq 0}^m$ and $\nu \in \mathbb{R}_{\geq 0}^n$.[2] Note that the dual potentials in problem (3) are elements in $L^1(\mu)$ resp. $L^1(\nu)$, hence we can abuse notation again and consider the potentials $f \in \mathbb{R}^m$ and $g \in \mathbb{R}^n$ to be vectors as well. This point of view gives rise to *discrete optimal*

---

[2]Whenever we view a discrete measure or a function as a vector, we will use bold characters.

*transport*. We set $C \in \mathbb{R}^{m \times n}$ via $C_{ij} = c(x_i, y_j)$, and view transport plans as matrices $\Pi \in \mathbb{R}^{m \times n}$ (see Appendix A.1 for a more thorough introduction to discrete OT). The following proposition shows that the plan $\Pi$ can be recovered from the dual vectors $\boldsymbol{f}$ and $\boldsymbol{g}$ (Peyré & Cuturi, 2019).

**Proposition 1.** *Define the Gibbs kernel $K = \exp(-C/\epsilon)$. The unique solution $\Pi$ of the discrete OT problem is given by*

$$\Pi = \operatorname{diag}(\boldsymbol{u}) K \operatorname{diag}(\boldsymbol{v}) \qquad (4)$$

*for two positive scaling vectors $\boldsymbol{u}$ and $\boldsymbol{v}$ unique up to a scaling constant (i.e. $\lambda \boldsymbol{u}$, $\frac{1}{\lambda}\boldsymbol{v}$ for $\lambda > 0$). Furthermore, $(\boldsymbol{u}, \boldsymbol{v})$ are linked to a solution $(\boldsymbol{f}, \boldsymbol{g})$ of the dual problem via*

$$(\boldsymbol{u}, \boldsymbol{v}) = (\exp(\boldsymbol{f}/\epsilon), \exp(\boldsymbol{g}/\epsilon)).$$

In Section 3.1, we show how the solution to the entropic dual between discrete $(\boldsymbol{\mu}_n, \boldsymbol{\nu}_n)$ converges to the solution of the continuous dual (3) as $(\boldsymbol{\mu}_n, \boldsymbol{\nu}_n)$ converge to continuous measures $(\mu, \nu)$ in some way, which will be crucial for the design of our network $\mathrm{S}_{\boldsymbol{\phi}}$.

### 2.2. The Sinkhorn Algorithm

The Sinkhorn Algorithm 1 can iteratively solve the discrete dual problem and was introduced in (Cuturi, 2013). It requires an initialization $\boldsymbol{v}^0 \in \mathbb{R}^n$, which is typically set to $1_n$, and $\boldsymbol{\mu}$ and $\boldsymbol{\nu}$ to be positive everywhere.

---

**Algorithm 1** Sinkhorn($\boldsymbol{\mu}, \boldsymbol{\nu} > 0, K = \exp(-C/\epsilon), \epsilon, \boldsymbol{v}^0$)

1: **for** $l = 0, ..., N$ **do**
2: $\quad \boldsymbol{u}^{l+1} \leftarrow \boldsymbol{\mu}./K\boldsymbol{v}^l$
3: $\quad \boldsymbol{v}^{l+1} \leftarrow \boldsymbol{\nu}./K^\top \boldsymbol{u}^{l+1}$
4: **end for**
5: $\Pi \leftarrow \operatorname{diag}(\boldsymbol{u}^l) K \operatorname{diag}(\boldsymbol{v}^l), \operatorname{OT}_\epsilon(\boldsymbol{\mu}, \boldsymbol{\nu}) \leftarrow \langle C, \Pi \rangle$
6: **return** $\boldsymbol{u}, \boldsymbol{v}, \Pi, \operatorname{OT}_\epsilon(\boldsymbol{\mu}, \boldsymbol{\nu})$

---

In the algorithm, $./$ is to be understood as element-wise division. Sinkhorn and Knopp (Sinkhorn & Knopp, 1967) showed that the iterates $\boldsymbol{u}^l$ and $\boldsymbol{v}^l$ from the algorithm converge to the vectors $\boldsymbol{u}$ and $\boldsymbol{v}$ from Proposition 1.

### 2.3. Predicting Dual Potentials

Given discrete measures $\boldsymbol{\mu}$ and $\boldsymbol{\nu}$, UNOT should ultimately be used to approximate the associated transport plan and cost. However, given an optimal dual potential $\boldsymbol{v}$, the corresponding potential $\boldsymbol{u}$ can be computed as

$$\boldsymbol{u} = \boldsymbol{\mu}./K\boldsymbol{v}, \qquad (5)$$

which also holds at convergence of the Sinkhorn algorithm. Thus, solving for the $m \times n$-dimensional plan $\Pi$ can be reduced to a $n$-dimensional problem over $\boldsymbol{v}$. Since computations in the log space tend to be more stable (Peyré &

Cuturi, 2019), we will instead let UNOT predict the dual potential $\boldsymbol{g} = \epsilon \log(\boldsymbol{v})$, i.e.

$$\mathrm{S}_{\boldsymbol{\phi}}(\boldsymbol{\mu}, \boldsymbol{\nu}) = \boldsymbol{g}, \quad \boldsymbol{\mu} \in \mathcal{P}(\mathcal{X}), \boldsymbol{\nu} \in \mathcal{P}(\mathcal{Y}).$$

The prediction $\boldsymbol{g}$ can then be used to solve the entropic OT problem via the relationship (5) and Proposition 1, or to initialize the Sinkhorn algorithm via $\boldsymbol{v}^0 = \exp(\boldsymbol{g}/\epsilon)$.

Note that the solution to the entropic dual is not unique (see Proposition 1). How we account for this non-uniqueness is explained in Section 3.4. However, when endowing $\mathbb{R}^m \times \mathbb{R}^n$ with the equivalence relation $(\boldsymbol{u}_1, \boldsymbol{v}_1) \sim (\boldsymbol{u}_2, \boldsymbol{v}_2) \Leftrightarrow \exists \lambda > 0 : (\boldsymbol{u}_1, \boldsymbol{v}_1) = (\lambda \boldsymbol{u}_2, \frac{1}{\lambda}\boldsymbol{v}_2)$ (i.e. accounting for the non-uniqueness of the dual solution), the map $(\boldsymbol{\mu}, \boldsymbol{\nu}) \mapsto \boldsymbol{v}$, mapping two measures to the associated dual potential in the quotient space, is Lipschitz continuous (Carlier et al., 2022), which supports its learnability by a neural network.

## 3. Universal Neural Optimal Transport

Consider the OT problem between two (grayscale) images, encoded as vectors in $\boldsymbol{\mu}_n, \boldsymbol{\nu}_n \in \mathbb{R}^n$. These can be viewed as discrete measures on $\mathcal{P}([0,1]^2)$, which discretize continuous measures $\mu, \nu \in \mathcal{P}([0,1]^2)$, where the discretization depends on the resolution of the image, and the continuous measures correspond to the images at "infinite" resolution. UNOT should predict the corresponding dual potential $\boldsymbol{g}_n \in \mathbb{R}^n$ solving (3) *independent* of the resolution $n$.[3] In Section 3.1, we establish a convergence result for the dual potentials as $n \to \infty$, which justifies the use of Neural Operators (Kovachki et al., 2024) as a parametrization of $\mathrm{S}_{\boldsymbol{\phi}}$; also see Section 3.2. Furthermore, as we want UNOT to work across datasets, we require a generator $\mathrm{G}_{\boldsymbol{\theta}}$ that can provably generate any pair of distributions during training (Section 3.3). In Section 3.4, we construct an adversarial training objective for $\mathrm{S}_{\boldsymbol{\phi}}$ and $\mathrm{G}_{\boldsymbol{\theta}}$. Further details about hyperparameter and architecture choices can be found in Appendix C. The implementation and model weights are available at `https://github.com/GregorKornhardt/UNOT`.

### 3.1. Convergence of Dual Potentials

In this section, we prove convergence of the discrete dual potentials $\boldsymbol{g}_n$ as $n$ goes to infinity. For brevity, this section is kept informal; see Appendix B for a formal treatment. Assume now that $\mathcal{X} = \mathcal{Y} \subseteq \mathbb{R}^N$ is compact, and $c(x, y)$ is Lipschitz continuous in both its arguments. For absolutely continuous $\mu, \nu \in \mathcal{P}(\mathcal{X})$, denote by $(\mu_n)_{n \in \mathbb{N}}, (\nu_n)_{n \in \mathbb{N}} \subset \mathcal{P}(\mathcal{X})$ *discretizing sequences* of $\mu$ and $\nu$ (formally defined in

---

[3]Note that while we consider images as an example, the learning task is the same for any setting where discrete measures of varying resolution share an underlying continuous cost function, which arises in settings such as single-cell genomics, fluid dynamics, point cloud processing, or economics.

Appendix B). While a solution $(f_n, g_n)$ of the discrete dual problem between $\mu_n$ and $\nu_n$ is only defined $\mu_n$ - resp. $\nu_n$ - a.e., it can be canonically extended to all of $\mathcal{X}$ (Feydy et al., 2018) (see Appendix B for details). The following proposition shows that the extended potentials $(f_n, g_n)$ converge to the solution $(f, g)$ of the continuous entropic problem.

**Proposition 2.** *(Informal) Let $(\mu_n)_{n \in \mathbb{N}}$, $(\nu_n)_{n \in \mathbb{N}}$ be discretizing sequences for absolutely continuous $\mu, \nu \in \mathcal{P}(\mathcal{X})$. Let $(f_n, g_n)$ be the (unique) extended dual potentials of $(\mu_n, \nu_n)$ such that $f_n(x_0) = 0$ for some $x_0 \in \mathcal{X}$ and all $n$. Let $(f, g)$ be the (unique) dual potentials of $(\mu, \nu)$ such that $f(x_0) = 0$. Then $f_n$ and $g_n$ converge uniformly to $f$ and $g$ on all of $\mathcal{X}$.*

A formal version and its proof can be found in Appendix B. This proposition is crucial in designing our network $S_\phi$, as we discuss in the following section.

### 3.2. Fourier Neural Operators

Fourier Neural Operators (FNOs) (Kovachki et al., 2024) are neural networks mapping between infinite-dimensional function spaces. More precisely, a neural operator is a map $F : \mathcal{A} \to \mathcal{U}$ between Banach spaces $\mathcal{A}$ and $\mathcal{U}$ of functions $a \in \mathcal{A} : \mathcal{D}_a \to \mathbb{R}^{d'_a}$ and $u : \mathcal{D}_u \to \mathbb{R}^{d'_u}$ respectively, for bounded domains $\mathcal{D}_a \subset \mathbb{R}^{d_a}$ and $\mathcal{D}_u \subset \mathbb{R}^{d_u}$. An input function $a \in \mathcal{A}$ evaluated at points $\boldsymbol{x}_1, ..., \boldsymbol{x}_n \in \mathbb{R}^{d_a}$ can be encoded as a vector $\boldsymbol{a} = [a(\boldsymbol{x}_1), ..., a(\boldsymbol{x}_n)] \in \mathbb{R}^{n \times d'_a}$; the same applies to the output function $u \in \mathcal{U}$, which can be written as $\boldsymbol{u} = [u(\boldsymbol{y}_1), ..., u(\boldsymbol{y}_m)] \in \mathbb{R}^{m \times d'_u}$, corresponding to the values at $\boldsymbol{y}_1, ..., \boldsymbol{y}_m \in \mathbb{R}^{d_u}$. At its core, an FNO applies a sequence of $L$ "kernel layers" to the input vector $\boldsymbol{a}$. In each of these layers, a fixed number of Fourier features of the discrete Fourier transform of the input is computed, the features are transformed by a ($\mathbb{C}$-valued) linear layer (we use a two-layer network in practice instead, as we found it to improve performance), and then mapped back by the inverse Fourier transform. Importantly, neural operators are by construction discretization-invariant when inputs and outputs correspond to discretizations of underlying functions. This is exactly what Proposition 2 guarantees: the dual potentials corresponding to measures $\mu_n$ and $\nu_n$ converge uniformly to the continuous potentials corresponding to the limiting distributions $\mu$ and $\nu$ as the resolution of $\mu_n$ and $\nu_n$ increases. Hence, FNOs are a natural choice of architecture in our setting. More details on FNOs, and how we implemented $S_\phi$, can be found in Appendix A.5.

### 3.3. Generating Measures for Training

UNOT is trained on pairs of distributions generated by a generator network $G_\theta$ of the following form:

$$G_\theta : \mathbb{R}^d \to \mathcal{P}(\mathcal{X}) \times \mathcal{P}(\mathcal{X})$$
$$\boldsymbol{z} \sim \rho_{\boldsymbol{z}} \mapsto R\left[\text{ReLU}\left(\text{NN}_\theta(\boldsymbol{z}) + \lambda I_{d,d'}(\boldsymbol{z})\right) + \delta\right], \quad (6)$$

where $\rho_{\boldsymbol{z}} = \mathcal{N}(0, I_d)$ is a Gaussian prior, $\text{NN}_\theta$ is a trainable neural network (in practice, we use a 5-layer fully connected MLP, see Appendix C), $I_{d,d'}$ is an interpolation operator matching the generator's output dimension $d'$ and acting as a skip connection reminiscent of ResNets (He et al., 2016), and $\lambda > 0$ is a constant for the skip connection. $\delta > 0$ is a small constant needed to generate our targets with the Sinkhorn algorithm, as outlined in Section 3.4. R denotes renormalizing to two probability measures and downsampling them to random dimensions in a set range, such that $S_\phi$ trains on measures of varying resolutions, which is known to improve NO training (Li et al., 2024a). More specifically, if we write $[\boldsymbol{x}_1, \boldsymbol{x}_2] = \text{ReLU}(\text{NN}_\theta(\boldsymbol{z}) + \lambda I_{d,d'}(\boldsymbol{z})) + \delta$ for two vectors $\boldsymbol{x}_1$ and $\boldsymbol{x}_2$ of equal size (say both with $n$ samples), R first maps them to $[\boldsymbol{x}_1/\sum_i(\boldsymbol{x}_1)_i, \boldsymbol{x}_2/\sum_i(\boldsymbol{x}_2)_i]$ and then uses 2D bilinear interpolation to downsample them to $m$ samples each. The generator is universal in the following sense:

**Theorem 3.** *Let $0 < \lambda \leq 1$ and $G_\theta : \mathbb{R}^d \to \mathbb{R}^d$ be defined via*

$$G_\theta(\boldsymbol{z}) = \text{ReLU}(\text{NN}_\theta(\boldsymbol{z}) + \lambda \boldsymbol{z}),$$

*where $\boldsymbol{z} \sim \rho_{\boldsymbol{z}} = \mathcal{N}(0, I)$, and where $\text{NN}_\theta : \mathbb{R}^d \to \mathbb{R}^d$ is Lipschitz continuous with $\text{Lip}(\text{NN}_\theta) = L < \lambda$. Then $G_\theta$ is Lipschitz continuous with $\text{Lip}(q) < L + \lambda$, and $\tilde{G}(z) := \text{NN}_\theta(\boldsymbol{z}) + \lambda \boldsymbol{z}$ is invertible on $\mathbb{R}^d$. Furthermore, for any $\boldsymbol{x} \in \mathbb{R}^d_{\geq 0}$ it holds*

$$\rho_{G_{\theta \# \rho_{\boldsymbol{z}}}}(\boldsymbol{x}) \geq \frac{1}{(L + \lambda)^d} \mathcal{N}\left(\tilde{G}_\theta^{-1}(\boldsymbol{x}) | 0, I\right).$$

*In other words, $G_{\theta \# \rho_{\boldsymbol{z}}}$ has positive density at any non-negative $\boldsymbol{x} \in \mathbb{R}^d_{\geq 0}$.*

This shows that any pair of discrete probability measures $(\boldsymbol{\mu}, \boldsymbol{\nu})$ of joint dimension $d$ can be generated by $G_\theta$. A direct consequence of the theorem is an extension to functions that are compositions of functions $\tilde{G}_\theta$ as above, which covers a wide class of ResNets. Both proofs can be found in Appendix B.

**Corollary 4.** *Let $\tilde{G}_\theta = \tilde{G}_{\theta_1} \circ \tilde{G}_{\theta_1} \circ ... \circ \tilde{G}_{\theta_R}$ be a composition of functions $\tilde{G}_{\theta_i}$, each of which is of the form as in Theorem 3. Let $\boldsymbol{z} \sim \rho_{\boldsymbol{z}} = \mathcal{N}(0, I)$. Then*

$$\rho_{\tilde{G}_{\theta \# \rho_{\boldsymbol{z}}}}(\boldsymbol{x}) \geq \frac{1}{(L + \lambda)^{Rd}} \mathcal{N}\left(\tilde{G}_\theta^{-1}(\boldsymbol{x}) | 0, I\right)$$

*for any $\boldsymbol{x} \in \mathbb{R}^d$. As in Theorem 3, this also holds for any $\boldsymbol{x} \in \mathbb{R}^d_{\geq 0}$ if $\tilde{G}_\theta$ is followed by a ReLU activation.*

Although the more general Corollary 4 is not needed for our purposes, it might be of independent interest to the research community. Note that the generator in Theorem 3 does not exactly match our generator's architecture. A discussion of how the theorem relates to our setting, as well as further details on the generator, can be found in Appendix C.

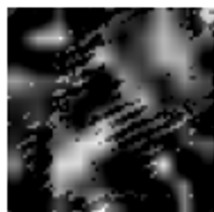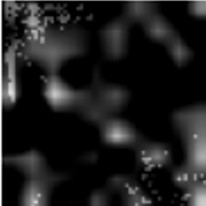

*Figure 2.* Generated pair of training samples (lighter=more mass).

Figure 2 shows a pair of samples generated by $G_{\theta}$. The generator seems to layer highly structured shapes with more blurry ones. More examples, as well as an analysis of the performance of $S_{\phi}$ on samples generated by $G_{\theta}$ over the course of training, can be found in Appendix D.6.

### 3.4. UNOT Training Algorithm

Given a pair of distributions $(\boldsymbol{\mu}, \boldsymbol{\nu}) = G_{\theta}(\boldsymbol{z})$ (in this section, we will remove the subscript $n$ for clarity), $S_{\phi}(\boldsymbol{\mu}, \boldsymbol{\nu}) =: \boldsymbol{g}_{\phi}$ should predict the true dual potential $\boldsymbol{g}$ associated with $\boldsymbol{\mu}$ and $\boldsymbol{\nu}$. Hence, we could simply compute the true potential $\boldsymbol{g}$ with the Sinkhorn algorithm and use $L_2(\boldsymbol{g}_{\phi}, \boldsymbol{g}) := \|\boldsymbol{g}_{\phi} - \boldsymbol{g}\|_2^2$ as our training loss. However, it would be prohibitively expensive to run the Sinkhorn algorithm until convergence. Hence, we instead employ a bootstrapping loss on the prediction $\boldsymbol{g}_{\phi}$. Let $\tau_k : (\boldsymbol{\mu}, \boldsymbol{\nu}, \boldsymbol{g}_{\phi}) \mapsto \boldsymbol{g}_{\tau_k}$ denote running the Sinkhorn algorithm on $(\boldsymbol{\mu}, \boldsymbol{\nu})$ with initialization $\boldsymbol{v}^0 = \exp(\boldsymbol{g}_{\phi}/\epsilon)$ for a very small number of iterations $k$, i.e. warmstarting the Sinkhorn algorithm with the current prediction $\boldsymbol{g}_{\phi}$, and returning $\epsilon \log \boldsymbol{v} = \boldsymbol{g}_{\phi}$.[4] To ensure uniqueness and improve training, we shift $\boldsymbol{g}_{\tau_k}$ to have zero sum; this corresponds to the non-uniqueness of the dual potentials, see Proposition 1. Minimizing $L_2(\boldsymbol{g}_{\phi}, \boldsymbol{g}_{\tau_k})$ implies minimizing the ground truth loss $L_2(\boldsymbol{g}_{\phi}, \boldsymbol{g})$ against the true potential $\boldsymbol{g}$.

**Proposition 5.** *For two discrete measures $(\boldsymbol{\mu}, \boldsymbol{\nu})$ with $n$ particles, let $\boldsymbol{g}$ be an optimal dual potential, $\boldsymbol{g}_{\phi} = S_{\phi}(\boldsymbol{\mu}, \boldsymbol{\nu})$, and $\boldsymbol{g}_{\tau_k} = \tau_k(\boldsymbol{\mu}, \boldsymbol{\nu}, \boldsymbol{g}_{\phi})$. Without loss of generality, assume that $\sum_i \boldsymbol{g}_i = \sum_i \boldsymbol{g}_{\tau_k i} = 0$. Then*

$$L_2(\boldsymbol{g}_{\phi}, \boldsymbol{g}) \leq c(K, k, n) \, L_2(\boldsymbol{g}_{\phi}, \boldsymbol{g}_{\tau_k})$$

*for some constant $c(K, k, n) > 1$ depending only on the Gibbs kernel $K$, $k$ and $n$.*

The proposition shows that minimizing $L_2(\boldsymbol{g}_{\phi}, \boldsymbol{g}_{\tau_k})$ implies minimizing $L_2(\boldsymbol{g}_{\phi}, \boldsymbol{g})$, i.e. the loss between the prediction and the ground truth potential. The proof is based on the Hilbert projective metric (Peyré & Cuturi, 2019) and can be found in Appendix B.

**Training objective.** Having defined the loss for $S_{\phi}$, as well

---

<hr>

[4]The Sinkhorn algorithm requires input measures to be positive; this is the reason we add the constant $\delta > 0$ in the generator.

---

**Algorithm 2** UNOT Training Algorithm

1:  **in** cost $c$, reg parameter $\epsilon$, prior $\rho_{\boldsymbol{z}}$, learning rates $\{\alpha_i\}_i$, $\{\beta_i\}_i$, Sinkhorn target generator $\tau_k$
2:  **for** $i = 1, 2, ..., T$ **do**
3:     $\boldsymbol{z} \leftarrow \text{sample}(\rho_{\boldsymbol{z}})$
4:     $(\boldsymbol{\mu}, \boldsymbol{\nu}) \leftarrow G_{\theta}(\boldsymbol{z})$
5:     **for** mini-batch $(\boldsymbol{\mu}^b, \boldsymbol{\nu}^b)$ in $(\boldsymbol{\mu}, \boldsymbol{\nu})$ **do**
6:       $\boldsymbol{g}_{\phi} \leftarrow S_{\phi}(\boldsymbol{\mu}^b, \boldsymbol{\nu}^b)$
7:       $\boldsymbol{g}_{\tau_k} \leftarrow \tau_k(\boldsymbol{\mu}^b, \boldsymbol{\nu}^b, \boldsymbol{g}_{\phi})$
8:       $\phi \leftarrow \phi - \alpha_i \nabla_{\phi} L_2(\boldsymbol{g}_{\tau_k}, \boldsymbol{g}_{\phi})$
9:     **end for**
10:   **for** mini-batch $\boldsymbol{z}^b$ in $\boldsymbol{z}$ **do**
11:     $(\boldsymbol{\mu}_{\theta}, \boldsymbol{\nu}_{\theta}) \leftarrow G_{\theta}(\boldsymbol{z}^b)$
12:     $\boldsymbol{g}_{\theta} \leftarrow S_{\phi}(\boldsymbol{\mu}_{\theta}, \boldsymbol{\nu}_{\theta})$
13:     $\boldsymbol{g}_{\tau_k} \leftarrow \tau_k(\boldsymbol{\mu}_{\theta}, \boldsymbol{\nu}_{\theta}, \boldsymbol{g}_{\theta})$
14:     $\theta \leftarrow \theta + \beta_i \nabla_{\theta} L_2(\boldsymbol{g}_{\tau_k}, g_{\theta})$
15:   **end for**
16: **end for**

---

as the target generation procedure, the training objective for $S_{\phi}$ and $G_{\theta}$ consists of $S_{\phi}$ trying to minimize the loss $L_2(\boldsymbol{g}_{\phi}, \boldsymbol{g})$, while $G_{\theta}$ attempts to maximize it, similar to the training objective in GANs (Goodfellow et al., 2014). Putting everything together, our adversarial training objective for $S_{\phi}$ and $G_{\theta}$ reads

$$\max_{\theta} \min_{\phi} \mathbb{E}_{\boldsymbol{z} \sim \rho_{\boldsymbol{z}}} \left[ L_2 \left( \tau_k \left( G(\boldsymbol{z}), S(G(\boldsymbol{z})) \right), S_{\phi}(G_{\theta}(\boldsymbol{z})) \right) \right], \tag{7}$$

where S and G without subscripts denote no gradient tracking, as the target is not backpropagated through. The training algorithm can be seen in Algorithm 2. In practice, training will be batched, which we omitted for clarity. Note that vectors $\boldsymbol{g}$ with subscripts $\theta$ or $\phi$ are backpropagated through with respect to these parameters, whereas target vectors (with subscript $\tau_k$) are not.

## 4. Experiments

**Training Details.** We test UNOT in three different settings: **a)** with $c(\boldsymbol{x}, \boldsymbol{y}) = \|\boldsymbol{x} - \boldsymbol{y}\|_2^2$ on the unit square $\mathcal{X} = [0, 1]^2$; **b)** with $c(\boldsymbol{x}, \boldsymbol{y}) = \|\boldsymbol{x} - \boldsymbol{y}\|_2$ on $[0, 1]^2$; **c)** with the *spherical distance* $c(\boldsymbol{x}, \boldsymbol{y}) = \arccos(\langle \boldsymbol{x}, \boldsymbol{y} \rangle)$ on the unit sphere $S^2 = \{\boldsymbol{x} \in \mathbb{R}^3 : \|\boldsymbol{x}\| = 1\}$. For each of these settings, we train a separate model on 200M samples $\boldsymbol{z}$, where training samples $(\boldsymbol{\mu}, \boldsymbol{\nu})$ are between $10 \times 10$ and $64 \times 64$ dimensional (randomly downsampled in $G_{\theta}$). Training takes around 35h on an H100 GPU. $S_{\phi}$ is an FNO with 26M parameters optimized with AdamW (Loshchilov & Hutter, 2019); $G_{\theta}$ is a fully connected MLP with 272k parameters optimized with Adam (Kingma & Ba, 2017). In the spherical setting $c(\boldsymbol{x}, \boldsymbol{y}) = \arccos(\langle \boldsymbol{x}, \boldsymbol{y} \rangle)$ we parametrize $S_{\phi}$ with a *Spherical FNO* (SFNO) (Bonev et al., 2023) instead, which is essentially an FNO adapted to the sphere; for more details

_Table 1._ Mean number of iterations needed to achieve 0.01 relative error on the OT distance for $c(\boldsymbol{x}, \boldsymbol{y}) = \|\boldsymbol{x} - \boldsymbol{y}\|^2$.

|  | UNOT (OURS) | ONES | GAUSS |
|---|---|---|---|
| MNIST | **3 ± 5** | 16 ± 9 | 10 ± 7 |
| CIFAR | **3 ± 6** | 80 ± 22 | 52 ± 19 |
| CIFAR-MNIST | **4 ± 4** | 32 ± 15 | 13 ± 9 |
| LFW | **7 ± 8** | 78 ± 20 | 35 ± 14 |
| BEAR | **4 ± 6** | 41 ± 16 | 25 ± 13 |
| LFW-BEAR | **4 ± 6** | 53 ± 18 | 29 ± 13 |

_Table 2._ Relative speedup of Sinkhorn with UNOT and cost $c(\boldsymbol{x}, \boldsymbol{y}) = \|\boldsymbol{x} - \boldsymbol{y}\|^2$. Time in s to achieve 0.01 relative error on the OT distance.

|  | UNOT (OURS) | ONES | SPEEDUP |
|---|---|---|---|
| MNIST | **$1.2 \cdot 10^{-3}$** | $1.5 \cdot 10^{-3}$ | 1.25 |
| CIFAR | **$9.5 \cdot 10^{-4}$** | $7.1 \cdot 10^{-3}$ | 7.4 |
| CIFAR-MNIST | **$1.3 \cdot 10^{-3}$** | $2.7 \cdot 10^{-3}$ | 2.07 |
| LFW | **$3.0 \cdot 10^{-3}$** | $1.5 \cdot 10^{-2}$ | 5 |
| BEAR | **$2.6 \cdot 10^{-3}$** | $1.0 \cdot 10^{-2}$ | 3.8 |
| LFW-BEAR | **$2.7 \cdot 10^{-3}$** | $1.2 \cdot 10^{-2}$ | 4.4 |

on FNOs and SFNOs see Appendix A.5. We highlight that $S_\phi$ is relatively small, such that its runtime vanishes compared to the runtime of even just a few Sinkhorn iterations, making it much cheaper to run than Sinkhorn (see Section 4.1). We set $k$ (the number of Sinkhorn iterations in $\tau_k$) to 5, and $\epsilon = 0.01$. Additional training details can be found in Appendix C.

We demonstrate the performance of the three models on various tasks, such as predicting transport distances, initializing the Sinkhorn algorithm, computing Sinkhorn divergence barycenters, and approximating Wasserstein geodesics. For the Euclidean settings a) and b) (from above), we view images as discrete distributions on the unit square, and test on MNIST (28×28), grayscale CIFAR10 (28×28), the teddy bear class from the Google Quick, Draw! dataset (64×64), and Labeled Faces in the Wild (LFW, 64×64), as well as cross-datasets CIFAR-MNIST and LFW-Bear (where $\boldsymbol{\mu}$ comes from one dataset and $\boldsymbol{\nu}$ from the other). For the spherical setting c), we project these images onto the unit sphere in $\mathbb{R}^3$ (for details, see Appendix D.1). Unless otherwise noted, we perform a single Sinkhorn iteration on $\boldsymbol{g} = S_\phi(\boldsymbol{\mu}, \boldsymbol{\nu})$ in all experiments in order to compute the second potential $\boldsymbol{f}$. Errors are averaged over 500 samples. Additional experiments, including a sweep over input sizes $10 \times 10$ to $64 \times 64$, as well as variants of UNOT for fixed input dimension or variable $\epsilon$, can be found in Appendix D.

### 4.1. Predicting Transport Distances

We compare the convergence of the Sinkhorn algorithm in terms of relative error on the transport distance $\text{OT}_\epsilon(\boldsymbol{\mu}, \boldsymbol{\nu})$ for our learned initialization $\boldsymbol{v}^0 = \exp(S_\phi(\boldsymbol{\mu}, \boldsymbol{\nu})/\epsilon)$ to

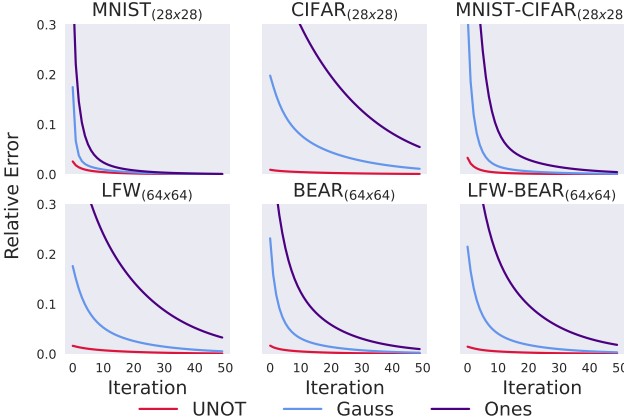

_Figure 3._ Relative error on the OT distance for Sinkhorn with our initialization (UNOT), compared to the default (Ones) and Gaussian initialization (Gauss) (Thornton & Cuturi, 2022).

the default initialization $1_n$ and the Gaussian initialization from (Thornton & Cuturi, 2022), which is based on closed-form solutions for Gaussian distributions. Note that the Gaussian initialization is only valid for $c(\boldsymbol{x}, \boldsymbol{y}) = \|\boldsymbol{x} - \boldsymbol{y}\|^2$, hence we omit it when $c(\boldsymbol{x}, \boldsymbol{y}) = \|\boldsymbol{x} - \boldsymbol{y}\|$ or $c(\boldsymbol{x}, \boldsymbol{y}) = \arccos(\langle \boldsymbol{x}, \boldsymbol{y} \rangle)$.[5] We do not compare to Meta OT (Amos et al., 2023) here, as their approach is inherently dataset dependent and breaks down when testing on out-of-distribution data.[6] For completeness, we include a detailed comparison in Appendix D.2, which shows that UNOT significantly outperforms Meta OT on all datasets except MNIST, the training dataset of Meta OT. Surprisingly, UNOT also almost matches Meta OT on MNIST, despite not having seen any MNIST samples during training, while Meta OT was explicitly trained on them.

Figure 1 (Section 1) shows the relative error on $\text{OT}_\epsilon(\boldsymbol{\mu}, \boldsymbol{\nu})$ after _a single_ Sinkhorn iteration for $c(\boldsymbol{x}, \boldsymbol{y}) = \|\boldsymbol{x} - \boldsymbol{y}\|^2$, and Figure 4 shows the same plot for $c(\boldsymbol{x}, \boldsymbol{y}) = \|\boldsymbol{x} - \boldsymbol{y}\|$ on the square, and $c(\boldsymbol{x}, \boldsymbol{y}) = \arccos(\langle \boldsymbol{x}, \boldsymbol{y} \rangle)$ on the sphere. In Figure 3, we plot the relative error on the OT distance over the number of Sinkhorn iterations for $c(\boldsymbol{x}, \boldsymbol{y}) = \|\boldsymbol{x} - \boldsymbol{y}\|^2$ (for the equivalent plots for the other cost functions, please see Appendix D.7), demonstrating that UNOT can be used as a state-of-the-art initialization. Table 1 shows the average number of Sinkhorn iterations needed to achieve 0.01 relative error on $\text{OT}_\epsilon(\boldsymbol{\mu}, \boldsymbol{\nu})$ for $c(\boldsymbol{x}, \boldsymbol{y}) = \|\boldsymbol{x} - \boldsymbol{y}\|^2$. In Table 2 we show the relative speedup achieved by initializing the Sinkhorn algorithm with UNOT implemented in JAX over the default initialization (on a batch size of 64 in `float32`

---

[5]If the cost function is not $\|\boldsymbol{x} - \boldsymbol{y}\|^2$, the Gaussian initialization is not theoretically justified. Empirically, we noted that it behaves similar to the default initialization in these cases.

[6]We note that it should be possible to finetune UNOT on specific datasets as well; however, we have not tested this.

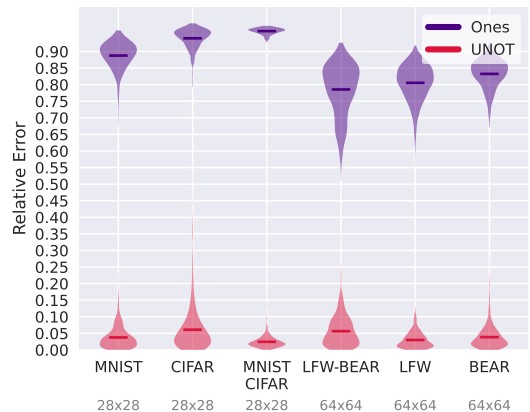
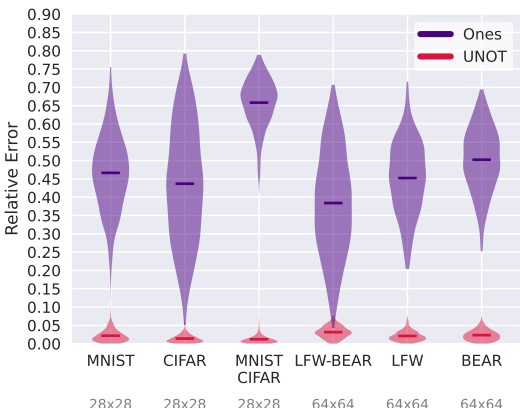

*Figure 4.* Error on the OT distance after a single Sinkhorn iteration with UNOT vs. the default initialization (Ones) for cost $\|\boldsymbol{x} - \boldsymbol{y}\|$ on the square $[0,1]^2$ (left) and $\arccos(\langle \boldsymbol{x}, \boldsymbol{y} \rangle)$ on the sphere $\{\boldsymbol{x} \in \mathbb{R}^3 : \|\boldsymbol{x}\| = 1\}$ (right).

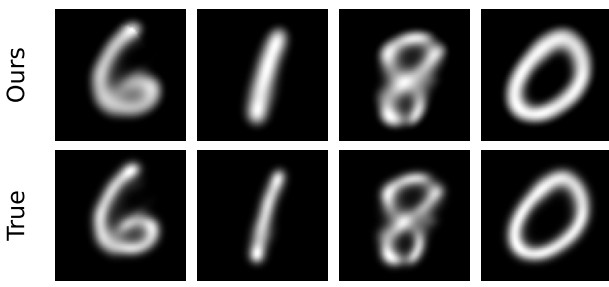

*Figure 5.* Sinkhorn divergence barycenters computed with UNOT via eq. (10) (top) vs. ground truth (bottom) of between 5 to 10 MNIST samples of the same digit per barycenter.

on an NVIDIA 4090). We achieve an average speedup of 3.57 on $28 \times 28$ datasets and 4.4 on $64 \times 64$ datasets.[7] For comparison, the relative speedup achieved in (Amos et al., 2023) was 1.96 (for a model trained only on MNIST).

### 4.2. Sinkhorn Divergence Barycenters

The *Wasserstein barycenter* for a set of measures $\{\nu_1, ..., \nu_N\} \subset \mathcal{P}_2(\mathcal{X})$ and $\lambda \in \Delta^{n-1}$ is defined as

$$\mu = \underset{\mu' \in \mathcal{P}_2(X)}{\arg\min} \sum_i \lambda_i W_2^2(\mu', \nu_i). \tag{8}$$

---

[7]We did not optimize the network $S_\phi$ much for efficiency, and more efficient implementations likely exist. Note that FNOs process complex numbers, but PyTorch is heavily optimized for real number operations. With kernel support for complex numbers, UNOT will likely be much faster. In addition, computation times can vary significantly across hardware, batch sizes, precision, etc.

To make this problem tractable, consider the Sinkhorn divergence barycenter

$$\mu = \underset{\mu' \in \mathcal{P}_2(X)}{\arg\min} \sum_i \lambda_i \, \mathrm{SD}_\epsilon(\mu', \nu_i), \tag{9}$$

where the *Sinkhorn divergence*[8] between $\mu$ and $\nu$ is

$$\mathrm{SD}_\epsilon(\mu, \nu) = \mathrm{OT}_\epsilon(\mu, \nu) - \frac{1}{2}\mathrm{OT}_\epsilon(\mu, \mu) - \frac{1}{2}\mathrm{OT}_\epsilon(\nu, \nu).$$

Now for discrete measures $\boldsymbol{\mu}, \boldsymbol{\nu}$, denote by $(\boldsymbol{f}, \boldsymbol{g})$ the dual potentials for $\mathrm{OT}_\epsilon(\boldsymbol{\mu}, \boldsymbol{\nu})$, and by $\boldsymbol{p}$ that for $\mathrm{OT}_\epsilon(\boldsymbol{\mu}, \boldsymbol{\mu})$.[9] Writing $\boldsymbol{\mu} = \sum_i \boldsymbol{a}_i \delta_{x_i}$ for some $\boldsymbol{a} \in \mathbb{R}^n$, the gradient of (9) w.r.t $\boldsymbol{a}$ is given by (cf. (Feydy et al., 2018)):

$$\nabla_{\boldsymbol{a}} \mathrm{SD}_\epsilon(\boldsymbol{\mu}, \boldsymbol{\nu}) = \boldsymbol{f} - \boldsymbol{p}. \tag{10}$$

Hence, we can solve (9) with (projected) gradient descent, where $S_\phi$ predicts $\boldsymbol{f}$ and $\boldsymbol{p}$ in (10).[10] Further details and a pseudocode can be found in Appendix A.2. Throughout this section, we set $c(\boldsymbol{x}, \boldsymbol{y}) = \|\boldsymbol{x} - \boldsymbol{y}\|^2$, and always run 200 gradient steps using gradients from (10). Figure 5 shows UNOT barycenters vs. the true barycenters (computed with the POT library) of between 5 and 10 MNIST samples of the same digit per barycenter. In Appendix D.7, we also provide quantitative results for barycenters with different initializations. In Figure 6, we show barycenters computed between four shapes. UNOT accurately predicting barycenters demonstrates it captures the geometry of the Wasserstein space beyond predicting distances.

---

[8]It can be seen as a *debiased* version of $\mathrm{OT}_\epsilon(\mu, \nu)$, and we use it as an approximation of the squared Wasserstein distance.

[9]If both measures are identical, the dual potentials can be chosen to be identical as well.

[10]To be precise, this solves the barycenter problem on the discrete space $\{x_1, ..., x_n\}$.

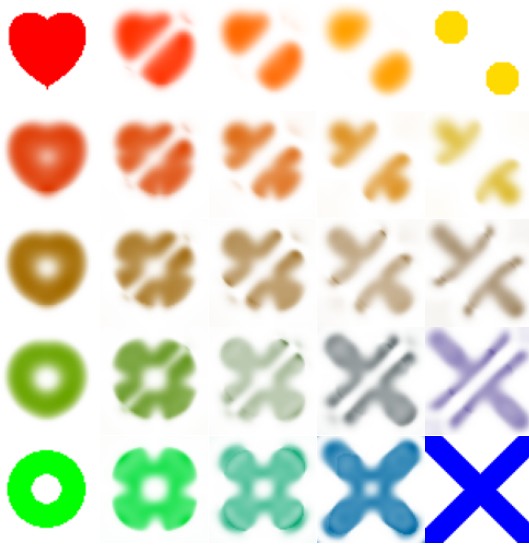

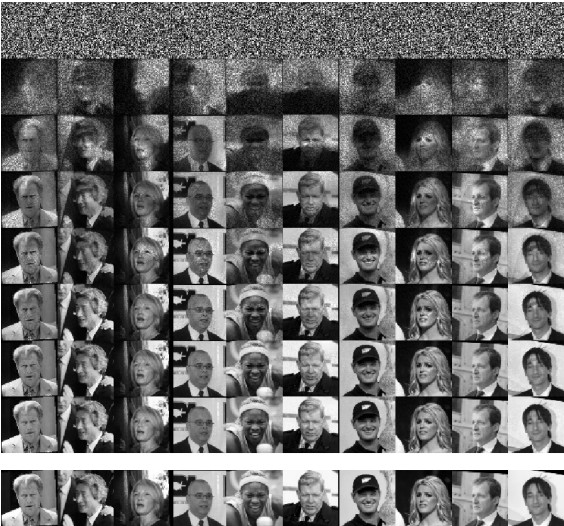

*Figure 6.* UNOT barycenters computed between four shapes (corners) by linearly interpolating $\lambda = (\lambda_1, \lambda_2, \lambda_3, \lambda_4)$ from eq. (9) between the four unit vectors, and solving via eq. (10) with UNOT.

### 4.3. Calculating Geodesics

Let $\mu, \nu \in \mathcal{P}_2(\mathcal{X})$ be two measures such that $\nu = T_{\#}\mu$ for an *optimal transport map* $T : \mathcal{X} \to \mathcal{X}$ (which exists for the non-entropic optimal transport problem under certain conditions, see Appendix A.1). The *Wasserstein geodesic* between $\mu$ and $\nu$, also called *McCann interpolation*, is the constant-speed geodesic between $\mu$ and $\nu$ and given by

$$\mu_t : [0, 1] \to \mathcal{P}_2(\mathcal{X}), \quad t \mapsto [(1 - t)\mathrm{Id} + tT]_{\#}\mu.$$

It can be interpreted as the shortest path between $\mu$ and $\nu$. The Wasserstein barycenter (8) between $(\mu, (1 - t))$ and $(\nu, t)$ (i.e. where $1 - t$ and $t$ are the weights $\lambda_i$ from equation (8)) turns out to be equal to $\mu_t$ (Agueh & Carlier, 2011). This gives us two methods to approximate the Wasserstein geodesic between $\mu$ and $\nu$: Either by iteratively computing barycenters as in Section 4.2, or by computing the (entropic) transport plan from equation (4) as an approximation to $T$ (we are leaving out some technicalities for brevity here, which can be found in Appendix A.3). We compare the geodesics computed by UNOT to the ground truth geodesic (obtained from the true OT plan), as well as to GeONet (Gracyk & Chen, 2024), a recently proposed framework that also uses Neural Operators to learn Wasserstein geodesics *directly* by parametrizing a coupled PDE system encoding the optimality conditions of the dynamic OT problem. Akin to (Amos et al., 2023), GeONet is inherently dataset dependent. Figure 8 shows the McCann interpolation between two MNIST digits using the ground truth OT plan, the OT plan computed by UNOT, barycenters computed by UNOT, and the GeONet geodesic, where we use the UNOT model trained with $c(\boldsymbol{x}, \boldsymbol{y}) = \|\boldsymbol{x} - \boldsymbol{y}\|^2$ again. We see that despite

*Figure 7.* Sinkhorn divergence particle flow between distributions of images, from noise to LFW (64x64). Gradients computed via eq. (11) and (10) with UNOT. Bottom row is target images.

GeONet being *trained to predict geodesics on MNIST*, while UNOT *does not train on geodesics, nor on MNIST*, both geodesics computed by UNOT are significantly closer to the ground truth than the GeONet geodesic.

### 4.4. Wasserstein on Wasserstein Gradient Flow

Oftentimes in machine learning, the distributions of interest are not images, but *distributions over images*, such as in generative modeling. In this experiment, we show that UNOT can successfully transport distributions over images as well. Let $\hat{\mu}, \hat{\nu} \in \mathcal{P}_2((\mathcal{P}_2([0, 1]^2), c), W_2)$, i.e. the space of distributions over images equipped with the Wasserstein distance (and $\mathcal{P}([0, 1]^2)$ being equipped with $c(\boldsymbol{x}, \boldsymbol{y}) = \|\boldsymbol{x} - \boldsymbol{y}\|^2$). Denote by $\hat{\mathrm{SD}}_{\epsilon}(\hat{\mu}, \hat{\nu})$ the Sinkhorn divergence between $\hat{\mu}$ and $\hat{\nu}$, where we use $\mathrm{SD}_{\epsilon}(\mu, \nu)$ as the ground cost between $\mu, \nu \in \mathcal{P}_2([0, 1]^2)$ as an approximation of $W_2^2(\mu, \nu)$. Writing $\hat{\mu} = \frac{1}{n}\sum_i^n \delta_{\mu_i}$, $\hat{\nu} = \frac{1}{n}\sum_j^n \delta_{\nu_j}$ for $\mu_i, \nu_j \in \mathcal{P}_2([0, 1]^2)$, we let UNOT approximate the particle flow $\frac{\partial}{\partial t}\hat{\mu}_t = -\nabla_{\hat{\mu}_t}[\hat{\mathrm{SD}}_{\epsilon}(\hat{\mu}_t, \hat{\nu})]$, for which we can derive the gradient via (see (Li et al., 2024b)):

$$\frac{\partial \hat{\mathrm{SD}}_{\epsilon}(\hat{\mu}, \hat{\nu})}{\partial \mu_k} = \sum_j \frac{\partial \mathrm{SD}_{\epsilon}(\mu_k, \nu_j)}{\partial \mu_k}\Pi_{kj}, \quad k = 1, ..., n,$$

(11)

where $\Pi_{kj}$ is an optimal transport plan between $\mu_k$ and $\nu_j$. These gradients can be approximated by UNOT via equation (10) as before; further details can be found in Appendix A.4. In Figure 7, we plot the particle flow from Gaussian noise $\hat{\mu}$ to a distribution $\hat{\nu}$ over 10 images, where we visualize $\hat{\mu}_t$ after every 10 gradient steps (using AdamW (Loshchilov & Hutter, 2019) with gradients computed via equation (10)). We can see that the UNOT flow converges quickly.

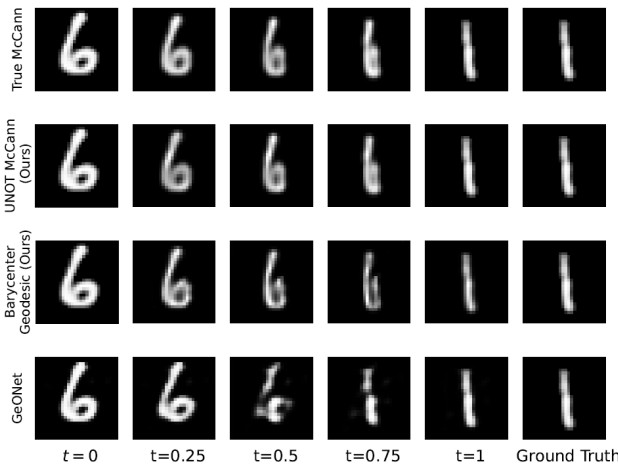

Figure 8. McCann interpolations computed with the true OT plan, UNOT OT plan, UNOT barycenters, and GeONet (top to bottom).

## 5. Related Work

**Neural OT.** Typically, neural OT approaches aim at solving individual instances of (high-dimensional) OT problems. In (Korotin et al., 2023), a maximin formulation for the dual problem is derived and two networks, parametrizing the transport plan and the dual potential resp., are trained adversarially. In (Bunne et al., 2023a), transport maps between continuous input distributions conditioned on a context variable are learned. Another interesting recent paper (Uscidda & Cuturi, 2023) suggests a universal regularizer, called the *Monge gap*, to learn OT maps and distances. Unlike these works, we focus on *generalizing across* OT problems.

**Initializing Sinkhorn.** There exists very little literature on initializing the Sinkhorn algorithm. (Thornton & Cuturi, 2022) propose using dual vectors recovered from the unregularized 1D optimal transport problem, or from closed-form transport maps in a Gaussian (mixture) setting, and were able to significantly speed up convergence. (Amos et al., 2023) propose a neural approach, training a single network to predict the optimal dual potential $f$ of the discrete dual problem, and their loss is simply the (negative) dual objective (3). This approach works well when training on low-dimensional datasets such as MNIST, and is elegant as it does not require ground truth potentials, i.e. is fully unsupervised, but it is not able to generalize to out-of-distribution data, and can only be used for input measures of fixed size.

**OT for Machine Learning.** Leveraging OT to formulate new machine learning methods has seen a surge in popularity in recent years, and it has been applied to a wide range of problems. Relevant works include the celebrated Wasserstein GAN (Arjovsky et al., 2017), multi-label learning (Frogner et al., 2015), inverse problems in physics (Engquist & Yang, 2019), point cloud processing (Geuter et al., 2025; Fishman et al., 2025), or few-shot image classifica-

tion to compute distances between images (Zhang et al., 2020). In flow matching OT can be used to straighten paths (Lipman et al., 2023; Tong et al., 2024; Pooladian et al., 2023). Approximating Wasserstein gradient flows with the JKO scheme has been explored in numerous works (Alvarez-Melis & Fusi, 2021; Alvarez-Melis et al., 2022; Bunne et al., 2022; Choi et al., 2024). The theory of Wasserstein gradient flows has also been used to study learning dynamics in various settings, such as for overparametrized two-layer networks (Chizat & Bach, 2018) or simplified transformers (Geshkovski et al., 2024).

**Generative Adversarial Networks.** GANs (Goodfellow et al., 2014), like other types of generative models, aim at generating samples from a distribution $\rho_{\text{data}}$, given access to a finite number of samples. In contrast, we do not have access to samples from the target distribution. However, our loss function (7) shares similarities with the adversarial GAN loss. Given prior samples $z \sim \rho_z$ and data samples $x \sim \rho_{\text{data}}$, the GAN objective for a generator G is

$$\min_{G} \max_{D} \mathbb{E}_{x \sim \rho_{\text{data}}} \left[ \log D(x) \right] + \mathbb{E}_{z \sim \rho_z} \left[ \log(1 - D(G(z))) \right],$$

where D is the *discriminator*, which predicts the probability that a sample came from the target distribution rather than the generator. Note that while our generator *maximizes* the objective, the GAN generator *minimizes* it.

## 6. Discussion

We presented UNOT, a neural OT solver capable of solving entropic OT problems universally across datasets, for a given cost function. Leveraging Neural Operators, UNOT can process distributions of varying resolutions supported on grids. UNOT's training involves a *generator* network $G_{\theta}$ producing synthetic training samples for the predictive network $S_{\phi}$, where both networks are trained jointly via a self-supervised adversarial loss. $S_{\phi}$ predicts the potential of the dual OT problem, and our training objective provably minimizes the loss w.r.t. the ground truth potentials. We show that UNOT is universal in the sense that the generator can create any discrete distributions during training, and empirically verify this through experiments on Euclidean and non-Euclidean image datasets of varying resolutions. UNOT consistently predicts OT distances up to 1-3% relative error, and approximates barycenters and geodesics in Wasserstein space by solving for the OT plan. Furthermore, we demonstrate that UNOT can be used as a state-of-the-art initialization for the Sinkhorn algorithm, achieving speedups of up to $7.4\times$. Current limitations are that UNOT does not extrapolate well to measures with significantly higher resolutions than the training samples, nor generalizes to cost functions other than the training cost. Scaling UNOT to higher resolutions, as well as applying it to other data modalities or non-uniform grids, are interesting directions for future research.

## Acknowledgements

For this work, VL has been funded by Deutsche Forschungsgemeinschaft (DFG) - Project-ID 318763901 - SFB1294. GK acknowledges funding within the BMBF project VI-Screen-PRO (Teilvorhaben: Mathematische Bildverarbeitung und maschinelles Lernen) VC3-23/13N17309.

## Impact Statement

This paper presents work whose goal is to advance the field of Machine Learning. There are many potential societal consequences of our work, as optimal transport has a vast range of applications, but none of these we feel must be specifically highlighted here.

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

## A. Background

### A.1. Optimal Transport

In this section, we recall some properties of optimal transport. First, we define the unregularized continuous problems for completeness.

**Problem A.1** (Kantorovich Optimal Transport Problem). For $\mu \in \mathcal{P}(\mathcal{X})$, $\nu \in \mathcal{P}(\mathcal{Y})$, and a cost $c : \mathcal{X} \times \mathcal{Y} \to \mathbb{R} \cup \{\infty\}$ the Kantorovich problem takes the form

$$\inf_{\pi \in \Pi(\mu,\nu)} \int c(x,y) \, \mathrm{d}\pi(x,y) \tag{12}$$

The infimum in (12) is called the *transport cost*, and the minimizer $\pi$, if it exists, the *optimal transport plan*.

The continuous dual problem is similar to the regularized dual (3). For a more thorough overview of OT, we refer the reader to (Villani, 2009; Peyré & Cuturi, 2019; Chewi et al., 2024).

**Problem A.2** (Dual Optimal Transport Problem). For $\mu, \nu$ and $c$ as before, the dual problem reads

$$\sup_{\substack{f \in L^1(\mu), g \in L^1(\nu) \\ f+g \leq c}} \int_{\mathcal{X}} f(x)\mathrm{d}\mu(x) + \int_{\mathcal{Y}} g(y)\mathrm{d}\nu(y),$$

where $f + g \leq c$ is to be understood as $f(x) + g(y) \leq c(x,y)$ for all $x, y$.

An important concept in optimal transport are *transport maps*.

**Definition A.3** (Transport Maps). A map $T : \mathcal{X} \to \mathcal{Y}$ is called a *transport map* between $\mu$ and $\nu$ if $\nu = T_{\#}\mu$. If there exists an optimal transport plan $\pi$ such that $\pi = (\mathrm{Id}, T)_{\#}\mu$, $T$ is called an *optimal transport map*.

Of course, not every transport plan admits a transport map; however, every transport map yields an optimal transport plan via $\pi = (Id, T)_{\#}\mu$. For sufficient conditions for the existence of both transport plans and maps, we refer the reader to (Villani, 2009).

In the paper we mentioned that if $\mu$ and $\nu$ are supported on finitely many points, one can rewrite the problems A.1 and A.2 with vectors. We now define the discrete problems carefully.

**Problem A.4** (Discrete Optimal Transport Problem). For two discrete measures $\boldsymbol{\mu} \in \Delta^{m-1}$ and $\boldsymbol{\nu} \in \Delta^{n-1}$, and a cost matrix $C \in \mathbb{R}^{m \times n}$, the discrete OT problem is defined as

$$L(\boldsymbol{\mu}, \boldsymbol{\nu}) := \min_{\Pi \in \Pi(\boldsymbol{\mu},\boldsymbol{\nu})} \langle C, \Pi \rangle$$

Here, $\Pi(\boldsymbol{\mu}, \boldsymbol{\nu})$ denotes the set of all *transport plans* between $\boldsymbol{\mu}$ and $\boldsymbol{\nu}$, i.e. matrices $\Pi \in \mathbb{R}_{\geq 0}^{m \times n}$ s.t. $\Pi 1_n = \boldsymbol{\mu}$ and $\Pi^\top 1_m = \boldsymbol{\nu}$. The problem has a dual formulation:

**Problem A.5** (Discrete Dual Optimal Transport Problem). For two discrete measures $\boldsymbol{\mu} \in \Delta^{m-1}$ and $\boldsymbol{\nu} \in \Delta^{n-1}$, and a cost matrix $C \in \mathbb{R}^{m \times n}$, the discrete dual OT problem is defined as

$$D(\boldsymbol{\mu}, \boldsymbol{\nu}) := \max_{\substack{\boldsymbol{f} \in \mathbb{R}^m, \, \boldsymbol{g} \in \mathbb{R}^n \\ \boldsymbol{f}+\boldsymbol{g} \leq C}} \langle \boldsymbol{f}, \boldsymbol{\mu} \rangle + \langle \boldsymbol{g}, \boldsymbol{\nu} \rangle$$

Here, $\boldsymbol{f} + \boldsymbol{g} \leq C$ is to be understood as $\boldsymbol{f}_i + \boldsymbol{g}_j \leq C_{ij}$ for all $i \in [\![m]\!]$, $j \in [\![n]\!]$. In the special case where $\mathcal{X} = \mathcal{Y}$ and $C$ corresponds to a metric, i.e. $C_{ij} = d(x_i, y_j)$, the *Wasserstein distance of order $p$ between $\boldsymbol{\mu}$ and $\boldsymbol{\nu}$* for $p \in [1, \infty)$ is defined as:

$$W_p(\boldsymbol{\mu}, \boldsymbol{\nu}) = \left( \min_{\pi \in \Pi(\boldsymbol{\mu},\boldsymbol{\nu})} \sum_{i,j} C_{ij}^p \Pi_{ij} \right)^{\frac{1}{p}} .$$

This definition coincides with the definition from the paper for continuous measures, if they are supported on finitely many points.

For completeness, we also state the entropically regularized primal and dual problem in the discrete case. The discrete problem is typically formulated with an entropy term instead of the KL divergence as in equation (1), but the two can be shown to be equivalent (Chewi et al., 2024).

**Definition A.6** (Entropy). For a matrix $P = [p_{ij}]_{ij} \in \mathbb{R}^{m \times n}$, we define its entropy $H(P)$ as

$$H(P) := -\sum_{i=1}^{m} \sum_{j=1}^{n} p_{ij} \log p_{ij}$$

if all entries are positive, and $H(P) := -\infty$ if at least one entry is negative. For entries $p_{ij} = 0$, we use the convention $0 \log 0 = 0$, as $x \log x \xrightarrow{x \to 0} 0$.

The entropic optimal transport problem is defined as follows.

**Problem A.7** (Entropic Discrete Optimal Transport Problem). For $\epsilon > 0$, the entropic optimal transport problem is defined as:

$$\mathrm{OT}_\epsilon(\boldsymbol{\mu}, \boldsymbol{\nu}) := \min_{\Pi \in \Pi(\boldsymbol{\mu}, \boldsymbol{\nu})} \langle C, \Pi \rangle - \epsilon \, \mathrm{H}(\Pi).$$

Note that this is identical to the unregularized optimal transport problem, except that the unregularized one does not contain the regularization term $-\epsilon H(\Pi)$. As the objective in Problem A.7 is $\epsilon$-strongly convex, the problem admits a unique solution (Peyré & Cuturi, 2019).

The *Gibbs kernel* is defined as $K = \exp(-C/\epsilon)$. Then the entropic dual problem reads:

**Problem A.8** (Entropic Discrete Dual Problem).

$$\max_{\boldsymbol{f}_\epsilon \in \mathbb{R}^m, \, \boldsymbol{g}_\epsilon \in \mathbb{R}^n} \langle \boldsymbol{f}_\epsilon, \boldsymbol{\mu} \rangle + \langle \boldsymbol{g}_\epsilon, \boldsymbol{\nu} \rangle - \epsilon \left\langle e^{\boldsymbol{f}_\epsilon/\epsilon}, K e^{\boldsymbol{g}_\epsilon/\epsilon} \right\rangle.$$

Again, without the regularization term $-\epsilon \left\langle e^{\boldsymbol{f}_\epsilon/\epsilon}, K e^{\boldsymbol{g}_\epsilon/\epsilon} \right\rangle$, this equals the regular optimal transport dual; note, however, that the unregularized dual is subject to the constraint $\boldsymbol{f} + \boldsymbol{g} \leq C$.

In both the continuous, as well as the discrete setting, there is *duality*, i.e. the optima of the primal and dual problems coincide. In addition, the optimizers are intrinsically linked, akin to Proposition 1 for the discrete entropic problem. We refer the reader to (Villani, 2009; Peyré & Cuturi, 2019) for more details.

### A.2. Calculating the Barycenter

Recall the Sinkhorn divergence barycenter for a set of discrete measures $\{\boldsymbol{\nu}_1, ..., \boldsymbol{\nu}_N\} \subset \mathcal{P}_2(\mathcal{X})$,

$$\boldsymbol{\mu} = \underset{\boldsymbol{\mu}' \in \mathcal{P}_2(X)}{\arg\min} \sum_i \alpha_i \, \mathrm{SD}_\epsilon(\boldsymbol{\mu}', \boldsymbol{\nu}_i).$$

For a solution $(\boldsymbol{f}, \boldsymbol{g})$ to the dual problem A.8 between two measures $\boldsymbol{\mu} = \sum_i \boldsymbol{a}_i \delta_{x_i}$ and $\boldsymbol{\nu} = \sum_j \boldsymbol{b}_j \delta_{y_j}$, it holds

$$\mathrm{SD}_\epsilon(\boldsymbol{\mu}, \boldsymbol{\nu}) = \langle \boldsymbol{\mu}, \boldsymbol{f} - \boldsymbol{p} \rangle + \langle \boldsymbol{\nu}, \boldsymbol{g} - \boldsymbol{q} \rangle,$$

see (Feydy et al., 2018). Here, $\boldsymbol{p}$ and $\boldsymbol{q}$ are the optimal potentials for $(\boldsymbol{\mu}, \boldsymbol{\mu})$ and $(\boldsymbol{\nu}, \boldsymbol{\nu})$ resp. (if both measures in the OT problem are the same, the dual potentials can be chosen to be equal).

From this identity, we immediately get

$$\nabla_{\boldsymbol{a}} \, \mathrm{SD}_\epsilon(\boldsymbol{\mu}, \boldsymbol{\nu}) = \boldsymbol{f} - \boldsymbol{p}.$$

Note that this is not a gradient with respect to the measure $\boldsymbol{\mu}$; instead, we view $\boldsymbol{\mu}$ as a vector, and compute the gradient w.r.t. the entries in that vector. This means we essentially compute the barycenter on the discrete space $\{x_1, ..., x_n\}$.

A pseudocode for how to approximate the Sinkhorn divergence barycenter with UNOT is given in Algorithm 3. Note that instead of using softmax to project back to probability measures, one could also just rescale; however, softmax proved better in practice. We also run a single Sinkhorn iteration on the output of $\mathrm{S}_\phi$ in practice, as it improved visual quality of the barycenters; however, this is not strictly needed.

---

**Algorithm 3** Barycenter Computation

---

1: **in** set of measures $\{\boldsymbol{\nu}_i\}_i \subset \Delta^{n-1}$, initial $\boldsymbol{\mu}_0 \in \Delta^{n-1}$, weights $\boldsymbol{\lambda} \in \Delta^{n-1}$
2: $\boldsymbol{\mu} \leftarrow \boldsymbol{\mu}_0$
3: **for** $i = 1, 2, ..., T$ **do**
4:     $\boldsymbol{p} \leftarrow \mathrm{S}_\phi(\boldsymbol{\mu}, \boldsymbol{\mu})$
5:     **for** $\boldsymbol{\nu}_i$ in $\{\boldsymbol{\nu}_i\}_i$ **do**
6:        $\boldsymbol{f}_i \leftarrow \mathrm{S}_\phi(\boldsymbol{\nu}_i, \boldsymbol{\mu})$          //switch the order of the arguments to get $\boldsymbol{f}_i$ instead of $\boldsymbol{g}_i$
7:     **end for**
8:     $\boldsymbol{\mu} \leftarrow \mathrm{softmax}\left(\boldsymbol{\mu} - \sum_i \boldsymbol{\lambda}_i(\boldsymbol{f}_i - \boldsymbol{p}_k)\right)$
9: **end for**

---

**Barycenters can be far when Transport Distances are close.** We now give a simple example that illustrates that merely predicting transport distances accurately does *not* necessarily imply predicting barycenters accurately, at least in the nonregularized setting. Let $\boldsymbol{\mu}$ be the measure with mass $1/2$ at the two points $(0, -1)$ and $(0, 1)$ in $\mathbb{R}^2$. Let $\boldsymbol{\nu}_1$ be the measure with mass $1/2$ at each of the points $(-1, 0)$ and $(1, -\epsilon)$ for a small $\epsilon > 0$, and $\boldsymbol{\nu}_2$ a measure with mass $1/2$ at each of the two points $(-1, 0)$ and $(1, \epsilon)$. Then as $\epsilon$ goes to 0, the transport distances between between $\boldsymbol{\mu}$ and $\boldsymbol{\nu}_1$ resp. $\boldsymbol{\mu}$ and $\boldsymbol{\nu}_2$ become arbitrarily close. However, the unique Wasserstein-$p$ barycenter (for $p > 1$) between $\boldsymbol{\mu}$ and $\boldsymbol{\nu}_1$ has mass $1/2$ at each of the points $(-1/2, 1/2)$ and $(1/2, -(1+\epsilon)/2)$, whereas the barycenter between $\boldsymbol{\mu}$ and $\boldsymbol{\nu}_2$ has mass $1/2$ at each of the points $(-1/2, -1/2)$ and $(1/2, (1+\epsilon)/2)$, so no matter how small $\epsilon$ gets, the barycenters will always be far apart.

### A.3. Geodesics

In Section 4.3, we saw that the McCann interpolation between two measures $\mu, \nu \in \mathcal{P}_2(\mathcal{X})$ is a constant-speed geodesic. In this section, we provide additional background on constant-speed geodesics, and establish a connection between constant-speed geodesics in $\mathcal{P}_2([0, 1]^2)$ and the notion of *strong-$\epsilon$ quasi-geodesics* in the discretized space $\mathcal{P}(\frac{[[n]]^2}{n})$. This makes our approximation of geodesics as in Section 4.3 more rigorous.

First, we recall the definition of constant-speed geodesics.

**Definition A.9.** A curve $\omega : [0, 1] \to (\mathcal{P}_2(\mathcal{X}), W_2)$ is called *constant-speed geodesic* between $\omega(0)$ and $\omega(1)$ if it satisfies

$$W_2(\omega(t), \omega(s)) = |t - s| W_2(\omega(0), \omega(1)), \quad \forall t, s \in [0, 1].$$

It turns out that for convex $\mathcal{X} \subset \mathbb{R}^d$, constant-speed geodesics are equivalent to push-forwards under transport plans, and if the starting point $\omega(0)$ is absolutely continuous, this is equal to the McCann interpolation.

**Theorem A.10.** *Let $\mathcal{X} \subset \mathbb{R}^d$ be convex. Then a curve $\omega : [0, 1] \to (\mathcal{P}_2(\mathcal{X}), W_2)$ is a constant-speed geodesic between $\omega(0)$ and $\omega(1)$ if and only if it is of the form*

$$\omega(t) = (p_t)_\# \pi$$

*for an optimal transport plan $\pi$ between $\omega(0)$ and $\omega(1)$, where the interpolation $p_t$ is given by $p_t(x, y) = (1 - t)x + ty$. If, in addition, $\omega(0)$ is absolutely continuous, then we can write*

$$\omega(t) = [(1 - t)\,\mathrm{Id} + tT]_\# \omega(0)$$

*for an optimal transport map $T$ from $\omega(0)$ to $\omega(1)$.*

This theorem holds, in fact, for any Wasserstein-$p$ space for $p > 1$, see (Santambrogio, 2016).

Now, denote by $\mu_t$ the McCann interpolation between $\mu$ and $\nu$. As mentioned in Section 4.2, we can express $\mu_t$ as the following barycenter:

$$\mu_t = \operatorname*{arg\,min}_{\mu' \in \mathcal{P}_2(\mathcal{X})} \left((1 - t)W_2^2(\mu', \mu) + tW_2^2(\mu', \nu)\right),$$

which we approximate by the Sinkhorn Divergence barycenter

$$\mu_t = \operatorname*{arg\,min}_{\mu' \in \mathcal{P}_2(\mathcal{X})} \left( (1-t) \operatorname{SD}_\epsilon(\mu', \mu) + t \operatorname{SD}_\epsilon(\mu', \nu) \right), \tag{13}$$

which is justified by the fact that Sinkhorn Divergences converge to the OT cost as $\epsilon \to 0$, and that they are reliable loss functions, in the sense that weak convergence of a sequence of measures is equivalent to convergence of the Sinkhorn divergence, see (Feydy et al., 2018). As also shown in (Feydy et al., 2018), the gradient of the Sinkhorn Divergence w.r.t. the vector $a$, when writing a discrete measure $\mu$ as $\mu = \sum_i a_i \delta_{x_i}$, is given by

$$\nabla_a \operatorname{SD}_\epsilon(\mu, \nu) = f - p. \tag{14}$$

However, if we now do a simple gradient descent on (13) using (14), we are not actually computing the barycenter on the space $\mathcal{P}_2(\mathcal{X})$ anymore, as we only consider gradients w.r.t. $a$, which does not allow particles to *move*, but merely to *teleport mass* to other particles. In particular, if $\mathcal{X}$ is a discrete space, there exist no constant-speed geodesics between different points anymore, as can easily be seen from the following example. Let $\mu_0 = \delta_{x_0}$ and $\mu_1 = \delta_{x_1}$ be two Dirac measures for some $x_0, x_1 \in X$. Assume there would exist a constant speed geodesic $\omega$ joining $\mu_0$ and $\mu_1$. Then for $t > 0$,

$$W_2(\omega(t), \omega(0)) = t W_2(\omega(1), \omega(0)).$$

However, since the space is discrete, this implies that $x_0 = x_1$, i.e. the only constant-speed geodesics are constant. We therefore work with the following approximation of geodesics.

**Definition A.11** (Quasi-Isometry). Let $(\mathcal{X}_1, d_1)$ and $(\mathcal{X}_2, d_2)$ be metric spaces. $f : \mathcal{X}_1 \to \mathcal{X}_2$ is called a $(\lambda, \epsilon)$-quasi-isometry if there exist $\lambda \geq 0$ and $\epsilon > 0$ such that for all $x, y \in X_1$

$$\frac{1}{\lambda} d_1(x, y) - \epsilon \leq d_2(f(x), f(y)) \leq \lambda d_1(x, y) + \epsilon$$

If in addition there exists a $C > 0$ such that for all $z \in \mathcal{X}_2$ there exist an $x \in \mathcal{X}_1$ such that $d_2(f(x), z) \leq C$, f is called quasi-isometry.

We can then use this to define quasi-geodesics. (Bonciocat & Sturm, 2009) introduced a similar concept called h-rough geodesics, for which they just used the upper bound.

**Definition A.12** (Strong-$\epsilon$ Quasi-Geodesics). A strong-$\epsilon$ quasi-geodesic in a metric space $(\mathcal{X}, d)$ is a map $\gamma : [0, 1] \to \mathcal{X}$ such that for all $s, t \in [0, 1]$,

$$d(\gamma_0, \gamma_1)|t - s| - \epsilon \leq d(\gamma_t, \gamma_s) \leq d(\gamma_0, \gamma_1)|t - s| + \epsilon.$$

Now let $\mathcal{X} = [0, 1]^2$, and denote by $\frac{[[n]]^2}{n} \subset [0, 1]^2$ the discrete space consisting of all $x_i$ of the form $x_i = \left( \frac{1}{2n}, \frac{1}{2n} \right) + k \left( \frac{1}{n}, 0 \right) + j \left( 0, \frac{1}{n} \right)$, for $k, j = 0, ..., n - 1$. We can then show that $(\mathcal{P}_2([0, 1]^2), W_2)$ is quasi-isometric to $(\mathcal{P}(\frac{[[n]]^2}{n}), W_2)$.

**Proposition A.13.** *The metric space $(\mathcal{P}_2([0, 1]^2), W_2)$ is $(1, \frac{1}{\sqrt{2}n})$-quasi-isometric to $(\mathcal{P}(\frac{[[n]]^2}{n}), W_2)$, i.e. there exist an $f : (\mathcal{P}_2([0, 1]^2), W_2) \to (\mathcal{P}(\frac{[[n]]^2}{n}), W_2)$ such that for all $\mu, \nu \in \mathcal{P}_2([0, 1])$ it holds that*

$$W_2(\mu, \nu) - \frac{1}{\sqrt{2}n} \leq W_2(f(\mu), f(\nu)) \leq W_2(\mu, \nu) + \frac{1}{\sqrt{2}n}.$$

*Proof.* We split the space $[0, 1]^2$ into squares via $N(x_i) := (x_i + [-\frac{1}{2n}, \frac{1}{2n}]^2)$. We define $f : \mathcal{P}([0, 1]^2) \to \mathcal{P}(\frac{[[n]]^2}{n})$ by

$$f(\mu) = \sum_{x_i \in X} \left( \int_{N(x_i)} d\mu \right) \delta_{x_i}.$$

By triangle inequality, we have

$$W_2(f(\mu), f(\nu)) \leq W_2(\mu, \nu) + W_2(\mu, f(\mu)) + W_2(\nu, f(\nu))$$
$$W_2(\mu, \nu) \leq W_2(f(\mu), f(\nu)) + W_2(\mu, f(\mu)) + W_2(\nu, f(\nu))$$

For any measure $\mu \in \mathcal{P}([0,1])$, denoting by $T : [0,1]^2 \to \frac{[[n]]^2}{n}$ the map that sends each point to the corresponding midpoint $x_i$, we get

$$W_2^2(\mu, f(\mu)) \leq \int_{[0,1]^2} |T(x) - x|^2 \mathrm{d}\mu \leq \int_{[0,1]^2} \frac{2}{4n^2} \mathrm{d}\mu = \frac{2}{4n^2}.$$

Therefore we have a $(1, \frac{1}{\sqrt{2}n})$-quasi-isometry between both spaces.

We also need to show that there exist a $C > 0$ such that for all $\mu \in \mathcal{P}(\frac{[[n]]^2}{n})$ there exist a $\nu \in \mathcal{P}([0,1]^2)$ with

$$W_2(f(\nu), \mu) < C.$$

Choosing $C = \frac{1}{\sqrt{2}n}$ and $\nu = \mu$ concludes the proof. $\qquad\square$

We immediately get the following corollary.

**Corollary A.14.** *Constant-speed geodesics $\mathcal{P}_2([0,1]^2)$ are strong-$\epsilon$ quasi-geodesics in $\mathcal{P}([[n]]^2/n)$.*

This justifies doing gradient descent on (13) using the discrete space gradient (14) to approximate the geodesic, as we can approximate the constant-speed geodesic with a strong-$\epsilon$-quasi-geodesic in the discrete space.

### A.4. Wasserstein on Wasserstein Distance

In this section, we provide additional details on how to solve the particle flow

$$\frac{\partial}{\partial t}\hat{\mu}_t = -\nabla_{\hat{\mu}_t}[\hat{\mathrm{SD}}_\epsilon(\hat{\mu}_t, \hat{\nu})] \tag{15}$$

from Section 4.4. Recall that $\hat{\mu}, \hat{\nu} \in \mathcal{P}_2(\mathcal{P}_2([0,1]^2, c), W_2)$, $\hat{\mu} = \frac{1}{n}\sum_i \delta_{\mu_i}$, $\hat{\nu} = \frac{1}{n}\sum_j \delta_{\nu_j}$ for $\mu_i, \nu_j \in \mathcal{P}_2([0,1]^2)$. From (Li et al., 2024b), we get that

$$\frac{\partial \hat{\mathrm{SD}}_\epsilon(\hat{\mu}, \hat{\nu})}{\partial \mu_k} = \sum_j \frac{\partial \mathrm{SD}_\epsilon(\mu_k, \nu_j)}{\partial \mu_k} \Pi_{kj},$$

where $\Pi_{kj}$ is an optimal transport plan between $\mu_k$ and $\nu_j$. Now as in the previous section, we can approximate $\frac{\partial \mathrm{SD}_\epsilon(\mu_k, \nu_j)}{\partial \mu_k} = \boldsymbol{f}_{kj} - \boldsymbol{p}_k$, where $\boldsymbol{f}_{kj}$ is the dual potential from $\mathrm{OT}_\epsilon(\mu_k, \nu_j)$, and $\boldsymbol{p}_k$ that of $\mathrm{OT}_\epsilon(\mu_k, \mu_k)$. As before, we can approximate these gradients with UNOT, which lets us solve (15) with a simple gradient descent scheme, as shown in Section A.2. As in Section A.2, we add a single Sinkhorn iteration on the predictions made by $\mathrm{S}_\phi$ as it improves visual quality, but this is not strictly necessary.

### A.5. Fourier Neural Operators

In this section, we describe FNOs in more detail. The main breakthrough for Neural Operators came in the combination with approximating solutions to partial differential equations (PDEs) (Li et al., 2020; 2021; Goswami et al., 2022). Many problems, including PDEs, can be numerically solved by discretizing infinite-dimensional input and output functions. Neural Operators are a class of neural networks that parametrize functions $F : \mathcal{A} \to \mathcal{U}$, where $\mathcal{A}$ and $\mathcal{U}$ are Banach spaces whose elements are functions $a : \mathcal{D}_a \to \mathbb{R}^{d_a}$ and $u : \mathcal{D}_u \to \mathbb{R}^{d'_u}$ respectively, for bounded domains $\mathcal{D}_a \subset \mathbb{R}^{d_a}$ and $\mathcal{D}_u \subset \mathbb{R}^{d_u}$. One of the main advantages of Neural Operators is that they can generalize over different grid discretizations, unlike traditional neural networks, which makes them particularly well-suited for solving PDEs,[11] and they are universal approximators for continuous operators acting on Banach spaces (Kovachki et al., 2024). While our space $\mathcal{P}([0,1]^2)$ is not

---

[11]For example, an FNO could be used to solve PDEs of the form $\Delta u = a$ with Dirichlet boundary conditions, for which we get a unique solution $u$ for every $a$. The FNO then maps each $a \in \mathcal{A}$ to the corresponding solution $u \in \mathcal{U}$.

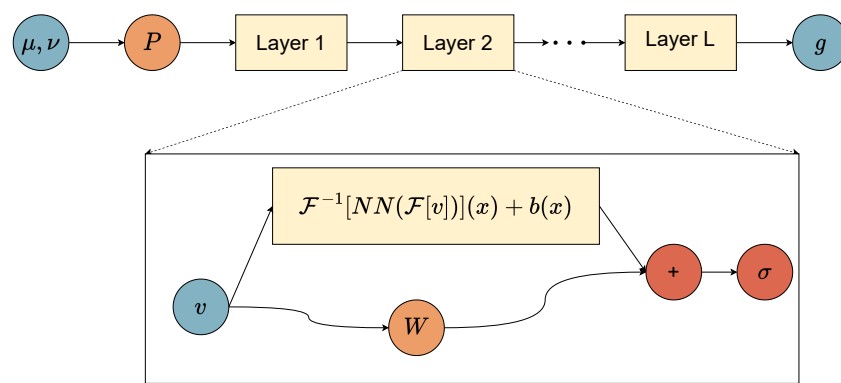

*Figure 9.* Fourier Neural Operator architecture, adapted from (Kovachki et al., 2024). The input measures $(\boldsymbol{\mu}, \boldsymbol{\nu})$ are passed through a point-wise lifting operator $P$ which is then followed by $L$ Fourier operators and point-wise non-linearity operators. After the last Fourier layer, we project back to the output potential $g$ with a point-wise operator $Q$.

technically a Banach space, the space of finite signed measures with the total variation norm is, and $\mathcal{P}([0,1]^2)$ is a subset. We note that approximation theory for Neural Operators usually

A neural operator usually has the following form:

$$F : \mathcal{A} \to \mathcal{U}$$
$$a \mapsto Q \circ B_L \circ .... \circ B_1 \circ P(a),$$

which in our setting becomes

$$S_{\boldsymbol{\phi}} : \mathcal{P}([0,1]^2) \times \mathcal{P}([0,1]^2) \to L^1([0,1]^2)$$
$$(\boldsymbol{\mu}, \boldsymbol{\nu}) \mapsto Q \circ B_L \circ .... \circ B_1 \circ P(\boldsymbol{\mu}, \boldsymbol{\nu}).$$

Here, $P$ is a *lifting map*, $B_i$ are the *kernel layers*, and $Q$ is a *projection* back to the target space.

Different versions of neural operators have been proposed, which mostly differ in how the kernel layers $B_i$ are defined. Our network $S_{\boldsymbol{\phi}}$ is parametrized as a *Fourier Neural Operator* (FNO) (Kovachki et al., 2024), where the kernel layers act on Fourier features of the inputs. We outline details for all the layers in the following.

- **Lifting** ($P$). The lifting map is a pointwise map $\{a : \mathcal{D}_a \to \mathbb{R}^{d'_a}\} \mapsto \{v_0 : \mathcal{D}_0 \to \mathbb{R}^{d_{v_0}}\}$, which maps the input $a$ to a function $v_0$ by mapping points in $\mathbb{R}^{d'_a}$ to points in $\mathbb{R}^{d_{v_0}}$. We use a 2D convolutional layer for $P$, and in our setting, $\mathcal{D}_a = [0,1]^2 \times [0,1]^2$, as we can view elements in $\mathcal{P}([0,1]^2)$ as maps $[0,1]^2 \to \mathbb{R}$ when dealing with discretizations of measures.

- **Iterative Fourier Layer** ($B_i$). The network has $L$ Fourier layers $B_i$. In each of them, we map $\{v_i : \mathcal{D}_i \to \mathbb{R}^{d_{v_i}}\} \mapsto \{v_{i+1} : \mathcal{D}_{i+1} \to \mathbb{R}^{d_{v_{i+1}}}\}$ by first applying the (discrete) Fourier transform $\mathcal{F}$ from which we select a fixed number of Fourier features, then a neural network $NN$ on these features, and then the inverse Fourier transform $\mathcal{F}^{-1}$. Note that the Fourier features are complex, hence the network $NN$ is also complex (with multiplications in $\mathbb{C}$). Each Fourier layer also contains a *bypass layer*, which is similar to a skip connection, but contains a layer $W$ which is typically a 2D convolution; cmp Figure 9. Hence, the output of the Fourier layer is given by $\sigma(\mathcal{F}^{-1}(NN(\mathcal{F}(v)) + b + Wv)$, where $\sigma$ is an activation.

- **Projection** ($Q$). The projection $Q$ is the analogue to the lifting layer, mapping the hidden representation to the output function $\{v_L : \mathcal{D}_L \to \mathbb{R}^{dv_L}\} \mapsto \{u : \mathcal{D}_u \to \mathbb{R}^{d'_u}\}$. In our setting, $\mathcal{D}_u = [0,1]^2$.

In contrast to (Kovachki et al., 2024), we found that a Fourier layer containing a two-layer neural network $NN$ instead of just a linear layer worked better in practice. Our bypass layer is still a linear layer $W$.

On the unit sphere $S^2$, we use Spherical FNOs (SFNOs) (Bonev et al., 2023) instead of regular FNOs, which respect the geometry of $S^2$. SFNOs leverage the Fourier transform on the sphere $\mathcal{F}^{S^2}$, which can be viewed as a change of basis into

an orthogonal basis of $L^2(S^2)$, instead of the regular Fourier transform $\mathcal{F}$ for flat geometries. Everything else about our architecture remains the same.

Details on hyperparameter choices can be found in Appendix C.

# B. Proofs

This section contains all proofs, as well as further technical details omitted in the paper. For convenience, we restate the statements from the paper.

We start off by rigorously restating Proposition 2. Let $\mathcal{X} \subset \mathbb{R}^N$ be a compact set. We start off with a natural definition of discretization of a continuous measure, which applies, for example, to discrete images as discretizations of an underlying "ground truth" continuous image.

**Definition B.1** (Discretization of Measures). Let $\mu \in \mathcal{P}(\mathcal{X})$ be an absolutely continuous measure, and let $\mathcal{X}_n = \{x_1^n, ..., x_n^n\} \subset \mathcal{X}$. The *discretization of $\mu$ on $\mathcal{X}_n$* is defined as the measure $\boldsymbol{\mu}_n \in \mathcal{P}(\mathcal{X})$ supported on $\mathcal{X}_n$, where

$$\boldsymbol{\mu}_n(x_i^n) = \int_{\Omega_i} \mathrm{d}\mu,$$

with

$$\Omega_i^n = \{x \in \mathcal{X} : \ \|x - x_i^n\| \leq \|x - y\| \ \forall y \in \mathcal{X}\}.$$

Note that the intersections $\Omega_i^n \cap \Omega_j^n$ have Lebesgue measure zero, so this is well-defined.

We cannot guarantee that an arbitrary sequence of discretizations $\mu_n$ converges weakly to $\mu$ as $n \to \infty$; simply consider the case where all the $x_i^n$ are identical for all $n$ and $i$. Hence, we need to ensure that the discretization is uniform over all of $\mathcal{X}$ in some way.

**Definition B.2** (Uniform Discretization). Let $\mathcal{X}_n = \{x_i^n, ..., x_n^n\}$ be subsets of $\mathcal{X}$ for all $n \in \mathbb{N}$. Then we call the sequence $(\mathcal{X}_n)_{n \in \mathbb{N}}$ a *uniform discretization* of $\mathcal{X}$ if for all $x \in \mathcal{X}$,

$$\lim_{n \to \infty} \min_{i=1,...,n} \|x - x_i^n\| = 0.$$

While this may seem like a "pointwise discretization" at first, it turns out to be uniform, as an Arzelà-Ascoli type argument shows.

**Theorem B.3.** *Let $\mathcal{X} \subset \mathbb{R}^d$ be compact, and let $\{\mathcal{X}_n\}_{n \geq 1}$ be a sequence of finite subsets of $\mathcal{X}$ with $|\mathcal{X}_n| = n$ for each $n$. The following are equivalent:*

1. $\lim\limits_{n \to \infty} \sup\limits_{x \in \mathcal{X}} \min\limits_{y \in \mathcal{X}_n} \|x - y\| = 0.$

2. $\forall x \in \mathcal{X} : \quad \lim\limits_{n \to \infty} \min\limits_{y \in \mathcal{X}_n} \|x - y\| = 0.$

*Proof.* **(1)** $\implies$ **(2)**. If $\sup_{x \in \mathcal{X}} \min_{y \in \mathcal{X}_n} \|x - y\| \to 0$, then in particular for each fixed $x \in \mathcal{X}$ we have

$$\min_{y \in \mathcal{X}_n} \|x - y\| \leq \sup_{z \in \mathcal{X}} \min_{y \in \mathcal{X}_n} \|z - y\| \longrightarrow 0.$$

**(2)** $\implies$ **(1)**. Define

$$f_n(x) = \min_{y \in \mathcal{X}_n} \|x - y\|, \qquad x \in \mathcal{X}.$$

By hypothesis (2), $f_n(x) \to 0$ for every $x \in \mathcal{X}$. Moreover for any $x, z \in \mathcal{X}$,

$$\left| f_n(x) - f_n(z) \right| = \left| \min_{y \in \mathcal{X}_n} \|x - y\| - \min_{y \in \mathcal{X}_n} \|z - y\| \right| \leq \|x - z\|,$$

so $\{f_n\}$ is equicontinuous on the compact set $\mathcal{X}$. Since $f_n \to 0$ pointwise, the Arzelà–Ascoli theorem upgrades to uniform convergence of the entire sequence (instead of just a subsequence):

$$\lim_{n \to \infty} \sup_{x \in \mathcal{X}} f_n(x) = \lim_{n \to \infty} \sup_{x \in \mathcal{X}} \min_{y \in \mathcal{X}_n} \|x - y\| = 0.$$

Hence (1) holds, completing the proof. $\qquad \square$

Note that condition (1) in Theorem B.3 is equivalent to Definition 1 of a "discrete refinement" in (Kovachki et al., 2024).

The following lemma holds.

**Lemma B.4** (Weak Convergence of Discretizations of Measures). *Let $\mu \in \mathcal{X}$ be absolutely continuous, and $(\mu_n)_{n \in \mathbb{N}}$ be a sequence of discretizations of $\mu$ supported on a uniform discretization $(\mathcal{X}_n)_{n \in \mathbb{N}}$ of $\mathcal{X}$. Then $\mu_n$ converges weakly to $\mu$.*

*Proof.* Let $f \in C_b(\mathcal{X})$ be a test function. We have to show that

$$\int_{\mathcal{X}} f \mathrm{d}\mu_n \xrightarrow{n \to \infty} \int_{\mathcal{X}} f \mathrm{d}\mu.$$

Since $\mathcal{X}$ is compact and $f \colon \mathcal{X} \to \mathbb{R}$ is continuous, by the Heine–Cantor theorem $f$ is uniformly continuous. Hence, for every $\varepsilon > 0$ there exists $\delta > 0$ such that

$$\|x - y\| < \delta \implies |f(x) - f(y)| < \varepsilon \quad \text{for all } x, y \in \mathcal{X}.$$

Since

$$\sup_{x \in \mathcal{X}} \min_{1 \le i \le n} \|x - x_i^n\| \xrightarrow{n \to \infty} 0,$$

we can choose $n'$ such that for all $n \ge n'$,

$$\sup_{x \in \mathcal{X}} \min_{1 \le i \le n} \|x - x_i^n\| < \delta.$$

In particular, for each $x \in \mathcal{X}$, there is some $x_i^n \in \mathcal{X}_n$ with $\|x - x_i^n\| < \delta$, giving

$$|f(x) - f(x_i^n)| < \varepsilon \quad \text{whenever } \|x - x_i^n\| < \delta. \tag{16}$$

Let

$$\Omega_i^n = \left\{ x \in \mathcal{X} : \|x - x_i^n\| \le \|x - x_j^n\| \text{ for all } j = 1, \ldots, n \right\}$$

as above. These sets form a partition of $\mathcal{X}$ (up to measure-zero boundaries). Then for all $n \ge n'$, we have (using equation (16)):

$$\left| \int_{\mathcal{X}} f \, \mathrm{d}\mu_n - \int_{\mathcal{X}} f \, \mathrm{d}\mu \right| = \left| \sum_{i=1}^{n} \int_{\Omega_i^n} \left( f(x_i^n) - f(x) \right) \mathrm{d}\mu(x) \right|$$

$$\le \sum_{i=1}^{n} \int_{\Omega_i^n} \left| f(x_i^n) - f(x) \right| \mathrm{d}\mu(x)$$

$$\le \sum_{i=1}^{n} \epsilon \mu(\Omega_i^n)$$

$$= \epsilon,$$

and letting $\epsilon \to 0$ finishes the proof. $\square$

In Proposition 2, we used the "canonical extension" for dual potentials. For a pair of dual variables $(f, g)$ solving the dual problem (3) between $\mu$ and $\nu$, their canonical extensions are defined by $f$ and $g$ satisfying the following conditions:

$$f(x) = -\epsilon \log \int_{\mathcal{X}} \exp \left( \frac{1}{\epsilon} \left( g(y) - c(x, y) \right) \right) \mathrm{d}\nu(y),$$

$$g(x) = -\epsilon \log \int_{\mathcal{X}} \exp \left( \frac{1}{\epsilon} \left( f(y) - c(x, y) \right) \right) \mathrm{d}\mu(y).$$

We refer to (Santambrogio, 2015; Feydy et al., 2018) for more details.

We can now state and prove a formal version of Proposition 2.

**Proposition 2.** *(Formal) Let $c(x, y) : \mathcal{X} \times \mathcal{X} \to \mathbb{R}$ be Lipschitz continuous in both its arguments, and $\mathcal{X} \subset \mathbb{R}^N$ compact. Let $(\mu_n)_{n \in \mathbb{N}}$, $(\nu_n)_{n \in \mathbb{N}}$ be discretization sequences for absolutely continuous $\mu, \nu \in \mathcal{P}(\mathcal{X})$, supported on a uniform discretization $(\mathcal{X}_n)_{n \in \mathbb{N}}$ of $\mathcal{X}$. Let $(f_n, g_n)$ be the (unique) extended dual potentials of $(\mu_n, \nu_n)$ such that $f_n(x_0) = 0$ for some $x_0 \in \mathcal{X}$ and all $n$. Let $(f, g)$ be the (unique) dual potentials of $(\mu, \nu)$ such that $f(x_0) = 0$. Then $f_n$ and $g_n$ converge uniformly to $f$ and $g$ on all of $\mathcal{X}$.*

*Proof.* By Lemma B.4, we know that $\mu_n \rightharpoonup \mu$ and $\nu_n \rightharpoonup \nu$. The statement now follows immediately from Proposition 13 in (Feydy et al., 2018). $\qquad\square$

**Theorem 3.** *Let $0 < \lambda \leq 1$ and $G_{\boldsymbol{\theta}} : \mathbb{R}^d \to \mathbb{R}^d$ be defined via*

$$G_{\boldsymbol{\theta}}(\boldsymbol{z}) = \mathrm{ReLU}\left(\mathrm{NN}_{\boldsymbol{\theta}}(\boldsymbol{z}) + \lambda \boldsymbol{z}\right),$$

*where $\boldsymbol{z} \sim \rho_{\boldsymbol{z}} = \mathcal{N}(0, I)$, and where $\mathrm{NN}_{\boldsymbol{\theta}} : \mathbb{R}^d \to \mathbb{R}^d$ is Lipschitz continuous with $\mathrm{Lip}(\mathrm{NN}_{\boldsymbol{\theta}}) = L < \lambda$. Then $G_{\boldsymbol{\theta}}$ is Lipschitz continuous with $\mathrm{Lip}(q) < L + \lambda$, and $\tilde{G}(z) := \mathrm{NN}_{\boldsymbol{\theta}}(\boldsymbol{z}) + \lambda \boldsymbol{z}$ is invertible on $\mathbb{R}^d$. Furthermore, for any $\boldsymbol{x} \in \mathbb{R}^d_{\geq 0}$ it holds*

$$\rho_{G_{\boldsymbol{\theta}\#}\rho_{\boldsymbol{z}}}(\boldsymbol{x}) \geq \frac{1}{(L + \lambda)^d} \mathcal{N}\left(\tilde{G}_{\boldsymbol{\theta}}^{-1}(\boldsymbol{x}) | 0, I\right).$$

*In other words, $G_{\boldsymbol{\theta}\#}\rho_{\boldsymbol{z}}$ has positive density at any non-negative $\boldsymbol{x} \in \mathbb{R}^d_{\geq 0}$.*

*Proof.* Since the Lipschitz constant of the sum of two functions is bounded by the sum of the Lipschitz constants of the two functions, we have

$$\mathrm{Lip}(\tilde{G}_{\boldsymbol{\theta}}) \leq L + \lambda.$$

From Theorem 1 in (Behrmann et al., 2019), it follows that $\tilde{G}_{\boldsymbol{\theta}}$ is invertible, and Lemma 2 therein implies

$$\mathrm{Lip}(\tilde{G}_{\boldsymbol{\theta}}^{-1}) \leq \frac{1}{\lambda - L}.$$

The Lipschitz continuity of $\tilde{G}_{\boldsymbol{\theta}}^{-1}$ implies that for any $\boldsymbol{h}, \boldsymbol{z} \in \mathbb{R}^d$ with $h \neq 0$, we have

$$
\begin{aligned}
\left\| \nabla \tilde{G}_{\boldsymbol{\theta}}(\boldsymbol{z}) \boldsymbol{h} \right\| &= \lim_{t \to 0} \left\| \frac{\tilde{G}_{\boldsymbol{\theta}}(\boldsymbol{z} + t\boldsymbol{h}) - \tilde{G}_{\boldsymbol{\theta}}(\boldsymbol{z})}{t} \right\| \\
&\geq \frac{1}{\mathrm{Lip}(\tilde{G}_{\boldsymbol{\theta}}^{-1})} \lim_{t \to 0} \left\| \frac{\tilde{G}_{\boldsymbol{\theta}}^{-1}(\tilde{G}_{\boldsymbol{\theta}}(\boldsymbol{z} + t\boldsymbol{h})) - \tilde{G}_{\boldsymbol{\theta}}^{-1}(\tilde{G}_{\boldsymbol{\theta}}(\boldsymbol{z}))}{t} \right\| \\
&= \frac{1}{\mathrm{Lip}(\tilde{G}_{\boldsymbol{\theta}}^{-1})} \|\boldsymbol{h}\| \\
&> 0,
\end{aligned}
$$

which shows that $\nabla \tilde{G}_{\boldsymbol{\theta}}$ is invertible everywhere. Hence, by the inverse function theorem, we get

$$\nabla \tilde{G}_{\boldsymbol{\theta}}^{-1}(\boldsymbol{x}) = \nabla \tilde{G}_{\boldsymbol{\theta}}^{-1}(\tilde{G}_{\boldsymbol{\theta}}(\tilde{G}_{\boldsymbol{\theta}}^{-1}(\boldsymbol{x}))) = (\nabla \tilde{G}_{\boldsymbol{\theta}}(\tilde{G}_{\boldsymbol{\theta}}^{-1}(\boldsymbol{x})))^{-1}$$

for any $\boldsymbol{x} \in \mathbb{R}^d$. Furthermore, similar to above, we have

$$\left\| \nabla \tilde{G}_{\boldsymbol{\theta}}(\boldsymbol{z}) \boldsymbol{e}_i \right\| = \lim_{t \to 0} \left\| \frac{\tilde{G}_{\boldsymbol{\theta}}(\boldsymbol{z} + t\boldsymbol{e}_i) - \tilde{G}_{\boldsymbol{\theta}}(\boldsymbol{z})}{t} \right\| \leq \mathrm{Lip}(\tilde{G}_{\boldsymbol{\theta}}) \lim_{t \to 0} \left\| \frac{\boldsymbol{z} + t\boldsymbol{e}_i - \boldsymbol{z}}{t} \right\| \leq L + \lambda,$$

where $\boldsymbol{e}_i$ is the $i^{\text{th}}$ unit vector. Hence, we get from Hadamard's inequality that

$$|\det \nabla \tilde{G}_{\boldsymbol{\theta}}(\boldsymbol{z})| \leq \Pi_i \left\| \nabla \tilde{G}_{\boldsymbol{\theta}}(\boldsymbol{z}) \boldsymbol{e}_i \right\| \leq \Pi_i (L + \lambda) = (L + \lambda)^d.$$

Putting everything together, by change of variables, we get for any $x \in \mathbb{R}^d$:

$$\rho_{\tilde{G}_{\theta\#}\rho_z}(x) = \rho_z(\tilde{G}_\theta^{-1}(x)) \left| \det\nabla\tilde{G}_\theta^{-1}(x) \right|$$

$$= \rho_z(\tilde{G}_\theta^{-1}(x)) \left| \det\nabla\tilde{G}_\theta(\tilde{G}_\theta^{-1}(x)) \right|^{-1}$$

$$\geq \frac{1}{(L+\lambda)^d} \rho_z(\tilde{G}_\theta^{-1}(x))$$

$$= \frac{1}{(L+\lambda)^d} \mathcal{N}(\tilde{G}_\theta^{-1}(x)|0, I).$$

Now clearly, if $x \in \mathbb{R}^d_{\geq 0}$, then

$$\rho_{G_{\theta\#}\rho_z}(x) \geq \rho_{\tilde{G}_{\theta\#}\rho_z}(x),$$

as for any $z$ with $\tilde{G}(z) = x$, we also have $G(z) = x$. Thus, we also have

$$\rho_{G_{\theta\#}\rho_z}(x) \geq \frac{1}{(L+\lambda)^d} \mathcal{N}(\tilde{G}_\theta^{-1}(x)|0, I),$$

which finishes the proof. $\qquad\square$

**Corollary 4.** *Let $\tilde{G}_\theta = \tilde{G}_{\theta_1} \circ \tilde{G}_{\theta_1} \circ ... \circ \tilde{G}_{\theta_R}$ be a composition of functions $\tilde{G}_{\theta_i}$, each of which is of the form as in Theorem 3. Let $z \sim \rho_z = \mathcal{N}(0, I)$. Then*

$$\rho_{\tilde{G}_{\theta\#}\rho_z}(x) \geq \frac{1}{(L+\lambda)^{Rd}} \mathcal{N}\left(\tilde{G}_\theta^{-1}(x)|0, I\right)$$

*for any $x \in \mathbb{R}^d$. As in Theorem 3, this also holds for any $x \in \mathbb{R}^d_{\geq 0}$ if $\tilde{G}_\theta$ is followed by a ReLU activation.*

*Proof.* Consider the case where $\tilde{G}_\theta = \tilde{G}^1_{\theta_1} \circ \tilde{G}^2_{\theta_2}$. Then for any $x \in \mathbb{R}^d$, we get from the proof of Theorem 3 above:

$$\rho_{\tilde{G}_{\theta\#}\rho_z}(x) \geq \frac{1}{(L+\lambda)^d} \rho_{\tilde{G}^2_{\theta_2\#}\rho_z}((\tilde{G}^1_{\theta_1})^{-1}(x))$$

$$\geq \frac{1}{(L+\lambda)^{2d}} \mathcal{N}\left(\left(\tilde{G}^2_{\theta_2}\right)^{-1}\left(\left(\tilde{G}^1_{\theta_1}\right)^{-1}(x)\right)|0, I\right)$$

$$= \frac{1}{(L+\lambda)^{2d}} \mathcal{N}(\tilde{G}_\theta^{-1}(x)|0, I).$$

The claim now follows by induction over the layers of $\tilde{G}_\theta$. Note that if $\tilde{G}_\theta$ is followed by a ReLU activation, this inequality also holds for any $x \in \mathbb{R}^d_{\geq 0}$, similar to Theorem 3. $\qquad\square$

Next, we prove Proposition 5. The proof is based on the *Hilbert projective metric*. For two vectors $u, v \in \mathbb{R}^n_+$, it is defined as

$$d_H(u, v) := \max_i [\log(u_i) - \log(v_i)] - \min_i [\log(u_i) - \log(v_i)],$$

and can be shown to be a distance on the projective cone $\mathbb{R}^n_+ / \sim$, where $u \sim u'$ if $u = ru'$ for some $r > 0$ (Peyré & Cuturi, 2019; Franklin & Lorenz, 1989). For $f = \log(u)$ and $g = \log(v)$, we thus define the following loss:

$$L_H(f, g) := \max_i [f_i - g_i] - \min_i [f_i - g_i].$$

**Lemma B.5.** *Let $f, g \in \mathbb{R}^n$. Then*

$$L_H(f, g) \leq \sqrt{2}\|f - g\|_2.$$

*If, in addition, $\sum_i f_i = \sum_i g_i = 0$, then*

$$\|f - g\|_2 \leq \sqrt{n}\, L_H(f, g).$$

*Proof.* Let $\boldsymbol{h} = \boldsymbol{f} - \boldsymbol{g}$. For the first inequality, observe that $\mathrm{L}_H(\boldsymbol{f}, \boldsymbol{g}) = \max_i \boldsymbol{h}_i - \min_i \boldsymbol{h}_i$. Let $j^*$ and $k^*$ be the indices achieving $\max_i \boldsymbol{h}_i$ and $\min_i \boldsymbol{h}_i$, respectively. Define the vector $\boldsymbol{e}$ such that $\boldsymbol{e}_{j^*} = 1$, $\boldsymbol{e}_{k^*} = -1$, and $\boldsymbol{e}_i = 0$ for all other $i$. Then:

$$\mathrm{L}_H(\boldsymbol{f}, \boldsymbol{g}) = \boldsymbol{e} \cdot \boldsymbol{h} \leq \|\boldsymbol{e}\|_2 \|\boldsymbol{h}\|_2 = \sqrt{2} \|\boldsymbol{f} - \boldsymbol{g}\|_2.$$

Now assume that $\sum_i \boldsymbol{f}_i = \sum_i \boldsymbol{g}_i = 0$. Set $M = \max_i \boldsymbol{h}_i$ and $m = \min_i \boldsymbol{h}_i$. If all $\boldsymbol{h}_i = 0$, both statements are trivial. Hence, assume at least one of the $\boldsymbol{h}_i$ is not zero. Since $\sum_i \boldsymbol{h}_i = \sum_i \boldsymbol{f}_i - \boldsymbol{g}_i = 0$, this implies $M > 0$ and $m < 0$. For any index $i$, $\boldsymbol{h}_i \leq M$, and thus

$$(\boldsymbol{h}_i)^2 \leq M^2 \leq (M - m)^2 = \mathrm{L}_H(\boldsymbol{f}, \boldsymbol{g})^2.$$

Summing over all indices, we have:

$$\|\boldsymbol{f} - \boldsymbol{g}\|_2^2 = \|\boldsymbol{h}\|_2^2 = \sum_{i=1}^n (\boldsymbol{h}_i)^2 \leq n \cdot \mathrm{L}_H(\boldsymbol{f}, \boldsymbol{g})^2.$$

Taking the square root yields:

$$\|\boldsymbol{f} - \boldsymbol{g}\|_2 \leq \sqrt{n}\, \mathrm{L}_H(\boldsymbol{f}, \boldsymbol{g}).$$

This finishes the proof. $\qquad\square$

**Proposition 5.** *For two discrete measures $(\boldsymbol{\mu}, \boldsymbol{\nu})$ with $n$ particles, let $\boldsymbol{g}$ be a potential solving the dual problem, $\boldsymbol{g}_\phi = \mathrm{S}_\phi(\boldsymbol{\mu}, \boldsymbol{\nu})$, and $\boldsymbol{g}_{\tau_k} = \tau_k(\boldsymbol{\mu}, \boldsymbol{\nu}, \boldsymbol{g}_\phi)$ the target. Without loss of generality, assume that $\sum_i \boldsymbol{g}_i = \sum_i \boldsymbol{g}_{\tau_k i} = 0$. Then*

$$\mathrm{L}_2(\boldsymbol{g}_\phi, \boldsymbol{g}) \leq c(K, k, n)\, \mathrm{L}_2(\boldsymbol{g}_\phi, \boldsymbol{g}_{\tau_k})$$

*for some constant $c(K, k, n) > 1$ depending only on the Gibbs kernel $K$, $k$ and $n$.*

*Proof.* We first show a similar inequality as in Proposition 5 for the Hilbert loss. A well-known fact about the Hilbert metric is that positive matrices (in our case, the Gibb's kernel $K$) act as strict contractions on positive vectors with respect to the Hilbert metric (cf. Theorem 4.1 in (Peyré & Cuturi, 2019)). More precisely, we have

$$d_H(K\boldsymbol{v}, K\boldsymbol{v}') \leq \lambda(K) d_H(\boldsymbol{v}, \boldsymbol{v}')$$

for any positive vectors $\boldsymbol{v}, \boldsymbol{v}' \in \mathbb{R}^n$, where

$$\lambda(K) := \frac{\sqrt{\eta(K)} - 1}{\sqrt{\eta(K)} + 1}, \quad \eta(K) := \max_{i,j,k,l} \frac{K_{ik} K_{jl}}{K_{jk} K_{il}}.$$

The same inequality also holds for $K^\top$ in place of $K$. Note that by definition, $\eta(K) \geq 1$, hence $0 < \lambda(K) < 1$. Now consider a starting vector $\boldsymbol{v}^0$ to the Sinkhorn algorithm, and let $\boldsymbol{v}^l$ denote the $l^{\text{th}}$ iterate of the vector. Denote by $v^\star$ the limit $\lim_{l \to \infty} \boldsymbol{v}^l$ of the algorithm. Then (letting $'/'$ denote element-wise division):

$$\begin{aligned}
d_H(\boldsymbol{v}^{l+1}, \boldsymbol{v}^\star) &= d_H\left(\boldsymbol{\nu}/K^\top \boldsymbol{u}^{l+1}, \boldsymbol{\nu}/K^\top \boldsymbol{u}^\star\right) \\
&= d_H\left(K^\top \boldsymbol{u}^{l+1}, K^\top \boldsymbol{u}^\star\right) \\
&\leq \lambda(K) d_H(\boldsymbol{u}^{l+1}, \boldsymbol{u}^\star) \\
&= \lambda(K) d_H\left(\boldsymbol{\mu}/K\boldsymbol{v}^l, \boldsymbol{\mu}/K\boldsymbol{v}^\star\right) \\
&= \lambda(K) d_H\left(K\boldsymbol{v}^l, K\boldsymbol{v}^\star\right) \\
&\leq \lambda(K)^2 d_H\left(\boldsymbol{v}^l, \boldsymbol{v}^\star\right),
\end{aligned}$$

where we used the Hilbert metric inequality twice, once on $K$ and once on $K^\top$. Iteratively applying this inequality and translating into log-space notation, this gives us

$$\mathrm{L}_H(\boldsymbol{g}_{\tau_k}, \boldsymbol{g}) \leq \lambda(K)^{2k} \mathrm{L}_H(\boldsymbol{g}_\phi, \boldsymbol{g}).$$

For now, assume that $\sum_i \boldsymbol{g}_{\phi i} = 0$. By triangle inequality,

$$\mathrm{L}_H(\boldsymbol{g}_\phi, \boldsymbol{g}) \leq \mathrm{L}_H(\boldsymbol{g}_\phi, \boldsymbol{g}_{\tau_k}) + \mathrm{L}_H(\boldsymbol{g}_{\tau_k}, \boldsymbol{g}) \leq \mathrm{L}_H(\boldsymbol{g}_\phi, \boldsymbol{g}_{\tau_k}) + \lambda(K)^{2k} \mathrm{L}_H(\boldsymbol{g}_\phi, \boldsymbol{g}),$$

which gives us

$$\mathrm{L}_H(\boldsymbol{g}_\phi, \boldsymbol{g}) \leq \frac{1}{1 - \lambda(K)^{2k}} \, \mathrm{L}_H(\boldsymbol{g}_\phi, \boldsymbol{g}_{\tau_k}) =: c(K, k) \, \mathrm{L}_H(\boldsymbol{g}_\phi, \boldsymbol{g}_{\tau_k}).$$

Combining this with Lemma B.5 yields

$$\|\boldsymbol{g}_\phi - \boldsymbol{g}\|_2 \leq \sqrt{n} L_H(\boldsymbol{g}_\phi, \boldsymbol{g}) \leq \sqrt{n} c(K, k) L_H(\boldsymbol{g}_\phi, \boldsymbol{g}_{\tau_k}) \leq 2\sqrt{n} c(K, k) \, \|\boldsymbol{g}_\phi - \boldsymbol{g}_{\tau_k}\|_2 = c(K, k, n) \, \|\boldsymbol{g}_\phi - \boldsymbol{g}_{\tau_k}\|_2, \quad (17)$$

from which the claim follows by squaring both sides. We are left with proving the general case when $\sum_i \boldsymbol{g}_{\phi_i} \neq 0$. Write $\boldsymbol{g}_\phi = \hat{\boldsymbol{g}}_\phi + \bar{\boldsymbol{g}}_\phi$, where $\bar{\boldsymbol{g}}_\phi$ is equal to $\frac{1}{n}\sum_i \boldsymbol{g}_{\phi_i}$ in each entry, s.t. $\hat{\boldsymbol{g}}_\phi$ sums to zero. We then get

$$\mathrm{L}_2(\boldsymbol{g}_\phi, \boldsymbol{g}) = \|\hat{\boldsymbol{g}}_\phi - \boldsymbol{g}\|^2 + \|\bar{\boldsymbol{g}}_\phi\|^2, \quad (18)$$

as

$$\langle \hat{\boldsymbol{g}}_\phi - \boldsymbol{g}, \bar{\boldsymbol{g}}_\phi \rangle = 0.$$

Similarly, we get

$$\mathrm{L}_2(\boldsymbol{g}_\phi, \boldsymbol{g}_{\tau_k}) = \|\hat{\boldsymbol{g}}_\phi - \boldsymbol{g}_{\tau_k}\|^2 + \|\bar{\boldsymbol{g}}_\phi\|^2. \quad (19)$$

Combining equations (17), (18) and (19), we get

$$\begin{aligned}
\mathrm{L}_2(\boldsymbol{g}_\phi, \boldsymbol{g}) &= \|\hat{\boldsymbol{g}}_\phi - \boldsymbol{g}\|^2 + \|\bar{\boldsymbol{g}}_\phi\|^2 \\
&\leq c\,\mathrm{L}_2(\hat{\boldsymbol{g}}_\phi, \boldsymbol{g}_{\tau_k}) + \|\bar{\boldsymbol{g}}_\phi\|^2 \\
&= c\left(\mathrm{L}_2(\boldsymbol{g}_\phi, \boldsymbol{g}_{\tau_k}) - \|\bar{\boldsymbol{g}}_\phi\|^2\right) + \|\bar{\boldsymbol{g}}_\phi\|^2 \\
&= c\,\mathrm{L}_2(\boldsymbol{g}_\phi, \boldsymbol{g}_{\tau_k}) + (1 - c)\,\|\bar{\boldsymbol{g}}_\phi\|^2 \\
&\leq c\,\mathrm{L}_2(\boldsymbol{g}_\phi, \boldsymbol{g}_{\tau_k}),
\end{aligned}$$

where the last inequality follows from the fact that $1 - c < 0$. This finishes the proof. $\qquad\square$

*Remark* B.6. Looking at the proof of Proposition 5, one might wonder why we didn't opt for the Hilbert projective metric as the loss directly. We tried using it instead of L2, and it works quite well, but training with L2 seems to have an edge, probably because the indifference of the Hilbert projetive metric to constant shifts is not a helpful inductive bias for deep learning.

## C. Training Details

**Generator Architecture.** Recall that the generator is of the form

$$
\mathrm{G}_{\boldsymbol{\theta}} : \mathbb{R}^d \to \mathcal{P}(X) \times \mathcal{P}(X)
$$
$$
\boldsymbol{z} \sim \rho_{\boldsymbol{z}} \mapsto \mathrm{R}\left[\mathrm{ReLU}\left(\mathrm{NN}_{\boldsymbol{\theta}}(\boldsymbol{z}) + \lambda\,\mathrm{I}_{d,d'}(\boldsymbol{z})\right) + \delta\right],
$$

where we set $\lambda = 1.0$, $\delta = 1e\text{-}6$ (note we *first* normalize, then add $\delta$, and then *normalize again* in practice), and $\boldsymbol{z}$ is of size $2 \cdot 10 \times 10$. $R$ normalizes and randomly downsizes output distributions to resolutions between $10 \times 10$ and $64 \times 64$ (per distribution). This improves generalization of the FNO $\mathrm{S}_{\phi}$ across resolutions, which is true for FNOs in general (Li et al., 2024a). $\mathrm{NN}_{\boldsymbol{\theta}}$ is a five-layer fully connected MLP, where all hidden layers are of dimension $0.04 \cdot 64^2$, and the output is of dimension $2 \cdot 64^2$. All layers except the output layer contain Batch Normalization and ELU activations; the last layer has a sigmoid activation only. We note the architecture might seem strange, as the network is relatively deep, while the hidden layers are relatively narrow. However, this architecture worked best amongst an extensive sweep of architectures.

**Applying Theorem 3.** In the following, we discuss the relation between our generator $\mathrm{G}_{\boldsymbol{\theta}}$ and Theorem 3 in more detail. Note that Theorem 3 is not directly applicable to our setting for a few reasons: First, we add a small constant $\eta$ to the generator's output. This constant ensures that all training samples are positive everywhere, and vastly improves learning speed as it ensures that all inputs are active. However, this is not restrictive of the problem, as the Sinkhorn algorithm requires inputs to be positive anyways. Second, in Theorem 3 both in- and outputs to $\mathrm{G}_{\boldsymbol{\theta}}$ have the same dimension. This could be achieved in our setting by choosing the input dimension equal to the output dimension, i.e. $\mathrm{I}_{d,n}$ equal to the identity. However, in practice, using lower-dimensional inputs achieves significantly better results. This can be argued for by the manifold hypothesis (Fefferman et al., 2016), i.e. the fact that typically, datasets live on low-dimensional manifolds embedded in high-dimensional spaces. Depending on the application, i.e. the expected target dataset dimension, the dimension of the input can be adjusted accordingly. Finally, note that the theorem assumes that $\mathrm{NN}_{\boldsymbol{\theta}}$ is Lipschitz continuous with Lipschitz constant $L < \lambda$, where $\lambda$ is the scaling factor of the skip connection. We do not enforce this constraint, as not doing so yields empirically better results. Still, Theorem 3 goes to show that our algorithm's performance is not bottlenecked by the generator's inability to generalize. We note that a bound on the Lipschitz constant is not necessary for invertibility of ResNets; other approaches have been suggested in the literature, e.g. through the lens of ODEs (Chang et al., 2017) or by partitioning input dimensions (Jacobsen et al., 2018). It is also possible to directly divide by the Lipschitz constant of each layer (Serrurier et al., 2023); these approaches could be studied in future research.

We will now describe how one can bound the Lipschitz constant of the generator. Since $\lambda = 1.0$, we need to make sure that the Lipschitz constant of $\mathrm{net}_{\boldsymbol{\theta}}$ is smaller than $1$ in order for Theorem 3 to be applicable. Since the Lipschitz constant of a composition of functions is bounded by the product of the Lipschitz constants of each component function, this means we have to bound the product of the Lipschitz constants of components of $\mathrm{net}_{\boldsymbol{\theta}}$. ELU is Lipschitz continuous with constant $1$, whereas sigmoid's Lipschitz constant is $0.25$. Furthermore, for a batch normalization layer BN, we have

$$
\|\mathrm{BN}(x) - \mathrm{BN}(y)\| = \left\| \frac{x - \mu_b}{\sigma_b} - \frac{y - \mu_b}{\sigma_b} \right\| = \frac{1}{\sigma_b}\|x - y\|,
$$

where $\mu_b$ and $\sigma_b$ denote the empirical mean and standard deviation of the batch. Since we draw our data from a standard normal Gaussian, we have $\mathbb{E}[\sigma_b] = 1$, i.e. in expectation, the batch normalization layer is Lipschitz with constant $1$. Hence, all that remains is to bound the product of Lipschitz constants of the three linear layers by (any number smaller than) $4$ (because the constant of sigmoid is $0.25$, this will ensure that the network has a Lipschitz constant smaller than $1$), for which it suffices to bound the operator norms of the weight matrix of each layer. In practice, these can be approximated with the power method as in (Gouk et al., 2020) to find a lower bound on the Lipschitz constant of each linear layer, and these bounds can be used to add a soft constraint to the loss. Empirically, this suffices to bound the Lipschitz constant of the generator. Alternatively, one can use a hard constraint as outlined in (Behrmann et al., 2019). However, empirically, this proved detrimental to training, hence we did *not* control the Lipschitz constant during our training. Yet, Theorem 3 is still of value, as it goes to show that our algorithm's performance is not bottlenecked by the generator's inability to generalize. We leave properly enforcing the Lipschitz constraint for future research.

**Architecture of $\mathrm{S}_{\phi}$.** Our FNO architecture follows the general structure outlined in Section A.5. We set $d_{v_i} = 64$ for all $i$; recall this is the hidden dimension in the Fourier layer. We set the number of Fourier features selected from the Fourier transform to $10 \times 10$, i.e. 10 along each of the two dimensions of the domain. The (complex) weight matrices of the neural network in Fourier space, i.e. the one acting on the Fourier features, are tensors of shape $(d_{v_i}, 4d_{v_i}, N_{modes_x}, N_{modes_y}) =$

$(64, 256, 10, 10)$ and $(4d_{v_i}, d_{v_i}, N_{modes_x}, N_{modes_y}) = (256, 64, 10, 10)$ respectively, i.e. the hidden dimension is four times the hidden dimension of $v_i$. Note that since these are complex layers, each layer has two (real) weight tensors of this shape, one for the real and one for the complex part. These layers are the only complex layers in the network $S_\phi$. The inputs to the layer are of shape $(d_{v_i}, N_{modes_x}, N_{modes_y}) = (64, 10, 10)$ (in $\mathbb{C}$) and multiplied along all dimensions by the weights, i.e. for input $\hat{x} \in \mathbb{C}^{64,10,10}$ and weight matrix $A \in \mathbb{C}^{64,256,10,10}$ (the first of the two layers):

$$\hat{y}_{o,n,m} = \sum_i A_{i,o,n,m}\hat{x}_{i,n,m}.$$

The activation used within this network, as well as after each Fourier block, is GeLU. The lifting layer $P$, bypass layer $W$, and projection layer $Q$ are 2D convolutions with kernel size 1.

**Hyperparameters.** In Table 3 we present all relevant hyperparameters again for convenience.

*Table 3.* Training hyperparameters.

| Hyperparameter | Value |
|---|---|
| # params $G_\theta$ | 272k |
| # layers $G_\phi$ | 5 |
| hidden dims $G_\phi$ | (164, 164, 164, 164) |
| $\delta$ (eq. (6)) | 1e-6 |
| $\lambda$ (eq. (6)) | 1 |
| $d$ (dimension of latent $z$) | $2 \cdot 10 \times 10 = 200$ |
| optimizer $G_\phi$ | Adam |
| activations $G_\phi$ | ELU |
| $\beta_1$ (initial learning rate $G_\theta$) | 0.001 |
| learning rate decay $G_\theta$ | 1 |
| weight decay $G_\phi$ | 0 |
| # params $S_\phi$ | 26M |
| Number of Fourier layers | 4 |
| $d_{v_i}$ (dim. in Fourier blocks) | 64 |
| hidden dim. of Fourier NN | 256 |
| # layers in Fourier NN | 2 |
| $N_{modes_x}$ (# Fourier modes) | 10 |
| $N_{modes_y}$ (# Fourier modes) | 10 |
| optimizer $S_\phi$ | AdamW |
| $\sigma$ (activation in $S_\phi$) | GeLU |
| $\alpha_1$ (initial learning rate $S_\phi$) | 1e-4 |
| learning rate decay $S_\phi$ | 0.9999 |
| weight decay $S_\theta$ | 1e-4 |
| minimum training sample size | $10 \times 10$ |
| maximum training sample size | $64 \times 64$ |
| # training samples | 200M |
| batch size | 5000 |
| mini batch size | 64 |
| T (number batches) | 40k |
| $\epsilon$ (for Sinkhorn targets) | 0.01 |
| $k$ (# Sinkhorn iterations for targets) | 5 |

**Code.** Source code for UNOT, including the weights for the model used in the experiments, can be found at `https://github.com/GregorKornhardt/UNOT`.

# D. Additional Experiments and Materials

## D.1. Test Sets

In Figure 10 we show samples from our test datasets. For some of the experiments in the appendix, we included two additional datasets, the "cars" class which is also from the Quick, Draw! dataset, and the Facial Expressions dataset (Hashan, 2022), which consists of 48×48-dimensional greyscale images. The datasets are very diverse, and range in dimensionality from very low (MNIST) to fairly low (BEARS, CARS), medium high (CIFAR) and very high (EXPRESSIONS, LFW).

Figure 11 shows samples from our spherical datasets (where only part of the sphere is visible here). To create a grid on the sphere, we sample elevation angles $\theta$ uniformly in $\left[-\frac{\pi}{2}, \frac{\pi}{2}\right]$ and azimuthal angles $\varphi$ uniformly in $[0, 2\pi]$. Concretely, we set

$$\theta_i = -\frac{\pi}{2} + \frac{i}{n-1}\pi, \quad \varphi_j = \frac{2\pi j}{n-1}, \quad i,j = 0, \ldots, n-1,$$

and form the $n \times n$ grid $\left\{(\theta_i, \varphi_j)\right\}_{i,j}$. Each pair $(\theta_i, \varphi_j)$ is mapped to a point on the sphere by

$$x = \cos(\theta_i)\cos(\varphi_j), \quad y = \cos(\theta_i)\sin(\varphi_j), \quad z = \sin(\theta_i).$$

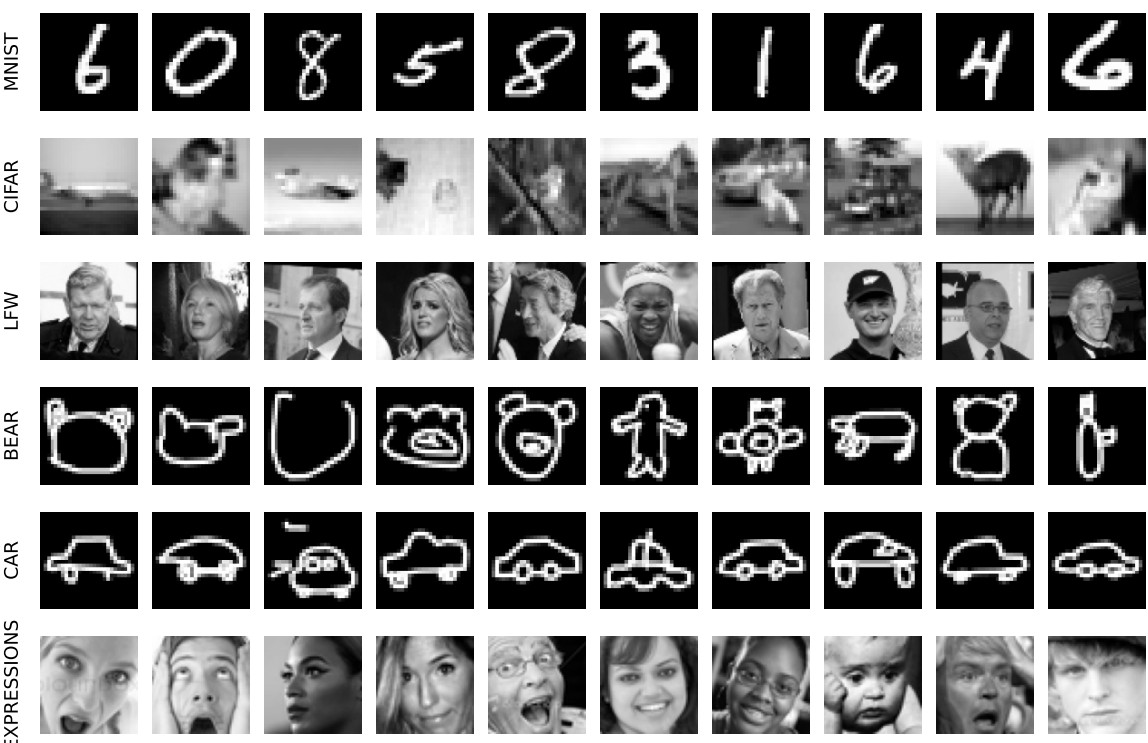

*Figure 10.* Test dataset samples on the unit square.

## D.2. Comparison with Meta OT

We trained a Meta OT (Amos et al., 2023) network with the official GitHub implementation[12] and compared it against UNOT on our test datasets, where we rescaled all datasets to $28 \times 28$, as Meta OT does not natively support inputs of varying sizes. In Table D.2, we report the relative errors on the OT distance (in %) after a single Sinkhorn iteration.

---

[12]https://github.com/facebookresearch/meta-ot

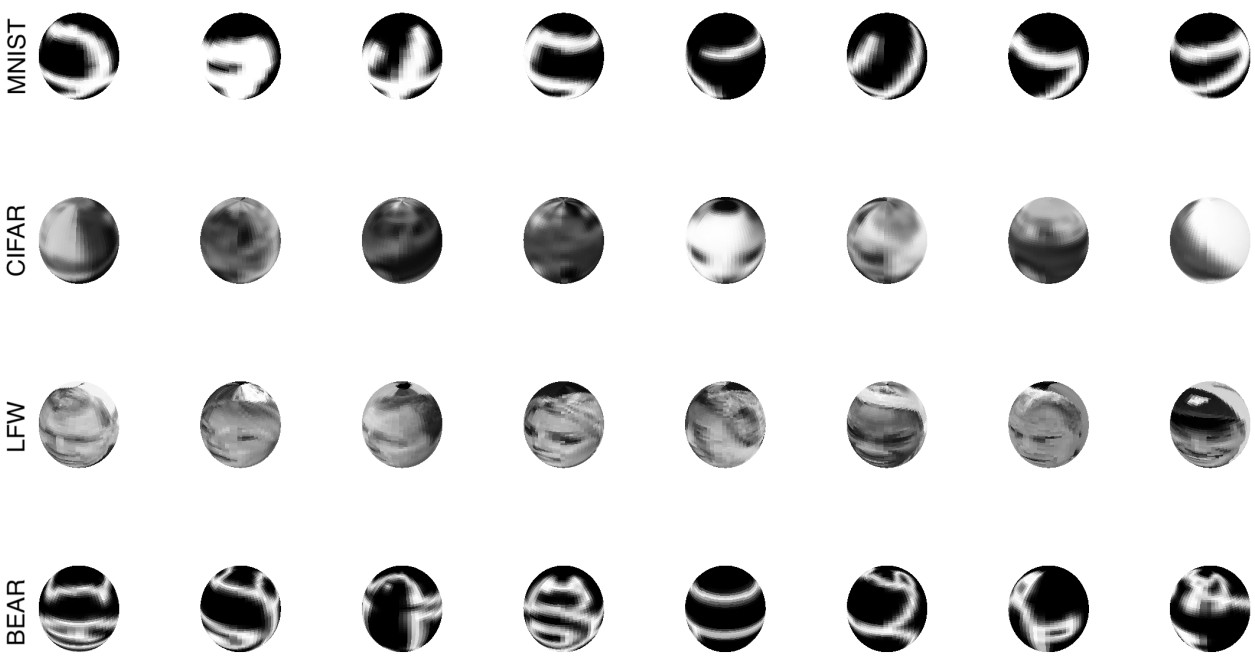

*Figure 11.* Test dataset samples on the sphere.

*Table 4.* Relative Errors on the OT distance (in %) after a single Sinkhorn iteration with UNOT's initialization, compared to Meta OT (Amos et al., 2023), the Gaussian initialization (Thornton & Cuturi, 2022), and the default initialization. Datasets rescaled to $28 \times 28$ such that the Meta OT network can process them.

|  | MNIST | CIFAR | MNIST-CIFAR | LFW | BEAR | LFW-BEAR |
|---|---|---|---|---|---|---|
| **UNOT (ours)** | $2.7 \pm 2.4$ | $\mathbf{1.3 \pm 1.1}$ | $\mathbf{2.8 \pm 2.6}$ | $\mathbf{1.5 \pm 1.3}$ | $\mathbf{2.0 \pm 1.6}$ | $\mathbf{1.8 \pm 1.3}$ |
| **MetaOT** | $\mathbf{2.4 \pm 1.8}$ | $23.1 \pm 15.7$ | $11.4 \pm 5.8$ | $24.6 \pm 15.7$ | $11.8 \pm 8.3$ | $31.0 \pm 14.8$ |
| **Gauss** | $18.1 \pm 10.0$ | $19.7 \pm 7.6$ | $32.2 \pm 8.7$ | $21.1 \pm 6.5$ | $20.4 \pm 8.3$ | $19.3 \pm 6.4$ |
| **Ones** | $39.5 \pm 13.4$ | $47.4 \pm 20.2$ | $74.5 \pm 6.9$ | $56.9 \pm 15.4$ | $54.2 \pm 13.5$ | $66.4 \pm 10.8$ |

We see that UNOT outperforms Meta OT on all datasets except MNIST, which is to be expected, as Meta OT is explicitly trained on MNIST, while UNOT is not trained on any MNIST data. However, surprisingly, we see that UNOT almost matches Meta OT's performance on MNIST, suggesting strong coverage of MNIST-like distributions by our generator network during training.

### D.3. MLP-UNOT

We mention that in applications of fixed-size distributions, one can replace the Neural Operator with an MLP and achieve similar results for a fraction of the training cost. We note that since the MLP acts on a fixed discrete space one does not need to have equispaced samples. In experiments, we found the MLP approach to also be very reliable for fixed-size inputs, and to vastly outperform the standard initialization of the Sinkhorn algorithm. Notably, it can be trained in just a few minutes to relative errors below 5%.

## D.4. Generalization across Resolutions

In this section, we show that UNOT successfully generalizes across resolutions. To this end, we downsample resp. upsample our test datasets to resolutions between $10 \times 10$ and $64 \times 64$. Figure 12 shows the relative errors on the transport distance over this range of resolutions after a single Sinkhorn iteration, compared against the default and the Gaussian initializations. (In Section D.5, we also provide some results on upsampling the dimension of the data beyond $64 \times 64$, i.e. beyond the largest resolution that the network saw during training.) We see that UNOT generalizes very well across all resolutions between $10 \times 10$ and $64 \times 64$.

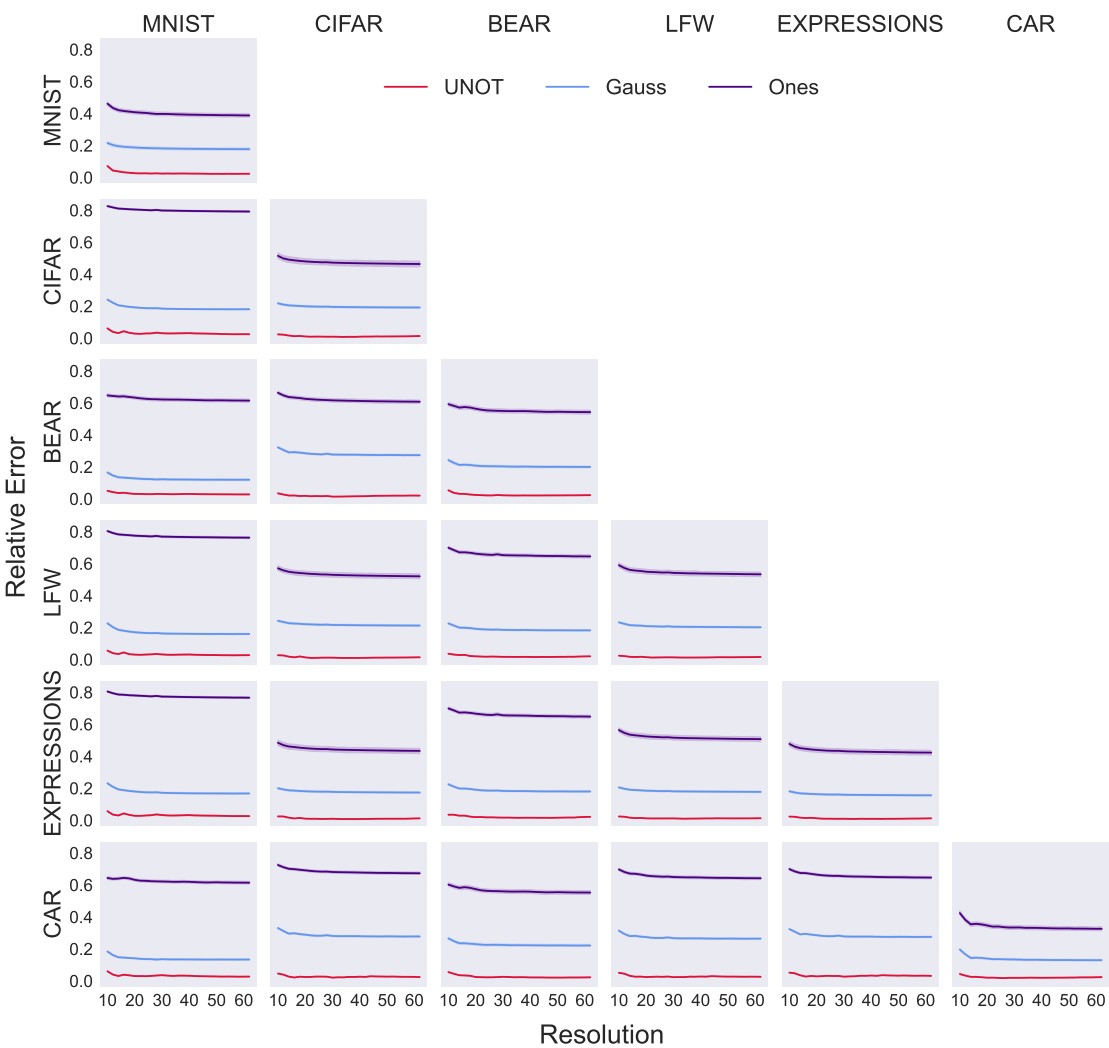

*Figure 12.* Relative error on the transport distance over the image resolution, ranging from $10 \times 10$ to $64 \times 64$.

## D.5. Variable Epsilon

In this section, we provide experimental results on a variant of UNOT that also receives the parameter $\epsilon$ as an input. Instead of the pair of measures $(\boldsymbol{\mu}, \boldsymbol{\nu})$ encoded as a tensor of size $(B, 2, n, n)$, we use an input size of $(B, 3, n, n)$, where the third channel is equal to $\epsilon$ everywhere. During training, we sample epsilon randomly per sample from a distribution with values between $0.01$ and $1$. Otherwise, training is identical to the training of regular UNOT. In Figure 13, we plot the relative errors over $\epsilon$ ranging from $0.01$ to $1$ on the x-axis, and the resolution of the data ranging from $10 \times 10$ to $70 \times 70$ on the y-axis (where we downsample resp. upsample the data to these dimensions, cf. Section D.4; note that we still only trained on image resolutions between $10 \times 10$ and $64 \times 64$). This variant of UNOT seems to do surprisingly well across different values of $\epsilon$ and across a wide range of resolutions, with relatively stable performance across different values of $\epsilon$. However, we can see that when the resolution gets smaller than around $15 \times 15$, or close to $70 \times 70$, the error increases.

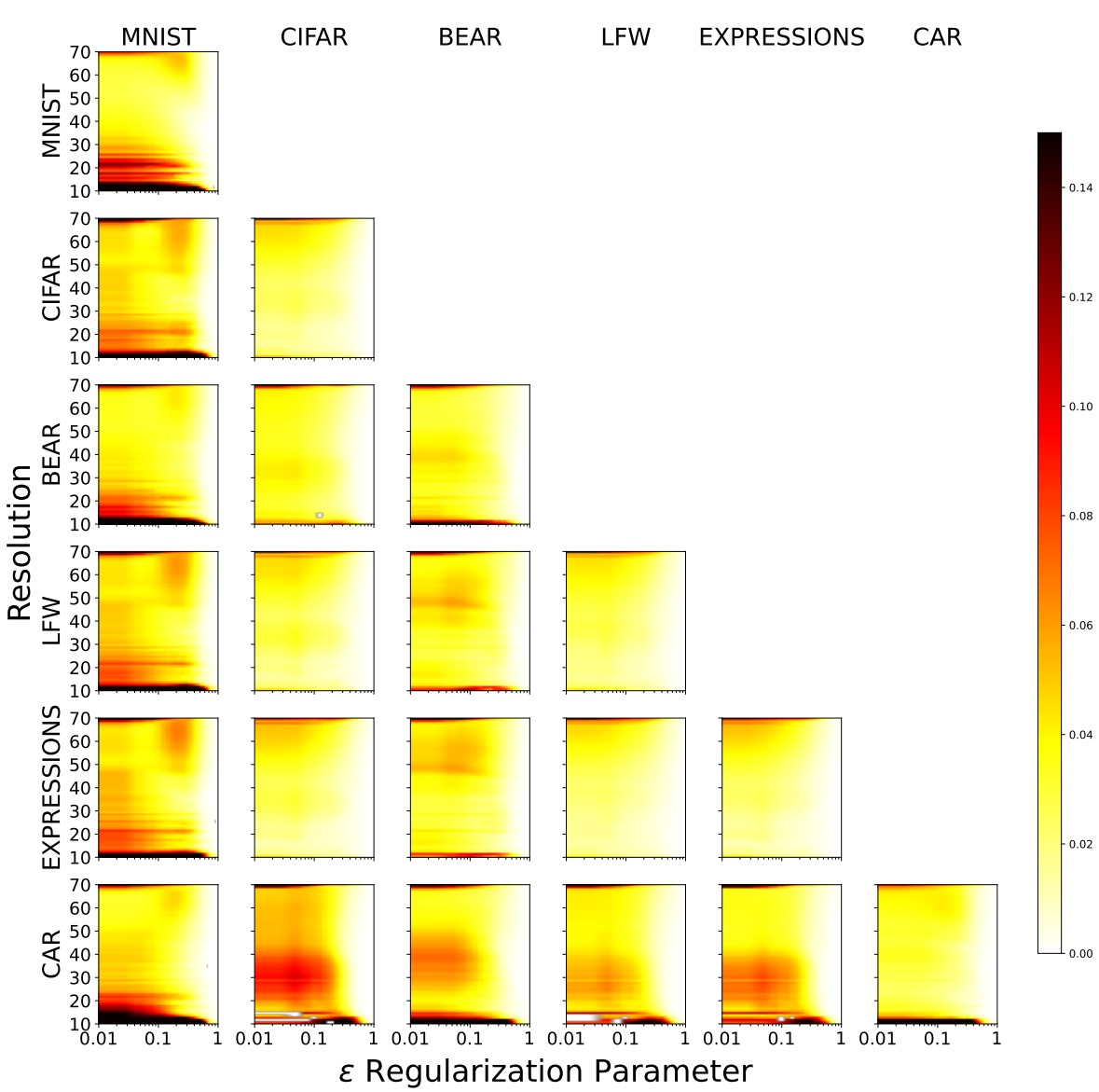

*Figure 13.* Relative error on the transport distance, over the resolution and varying values of $\epsilon$.

### D.6. Generated Measures

Figure 14 shows images created by the generator. The generator creates very different images over the course of training, including highly structured distributions, large areas of mass, and distributions with mass concentrated in very small areas.

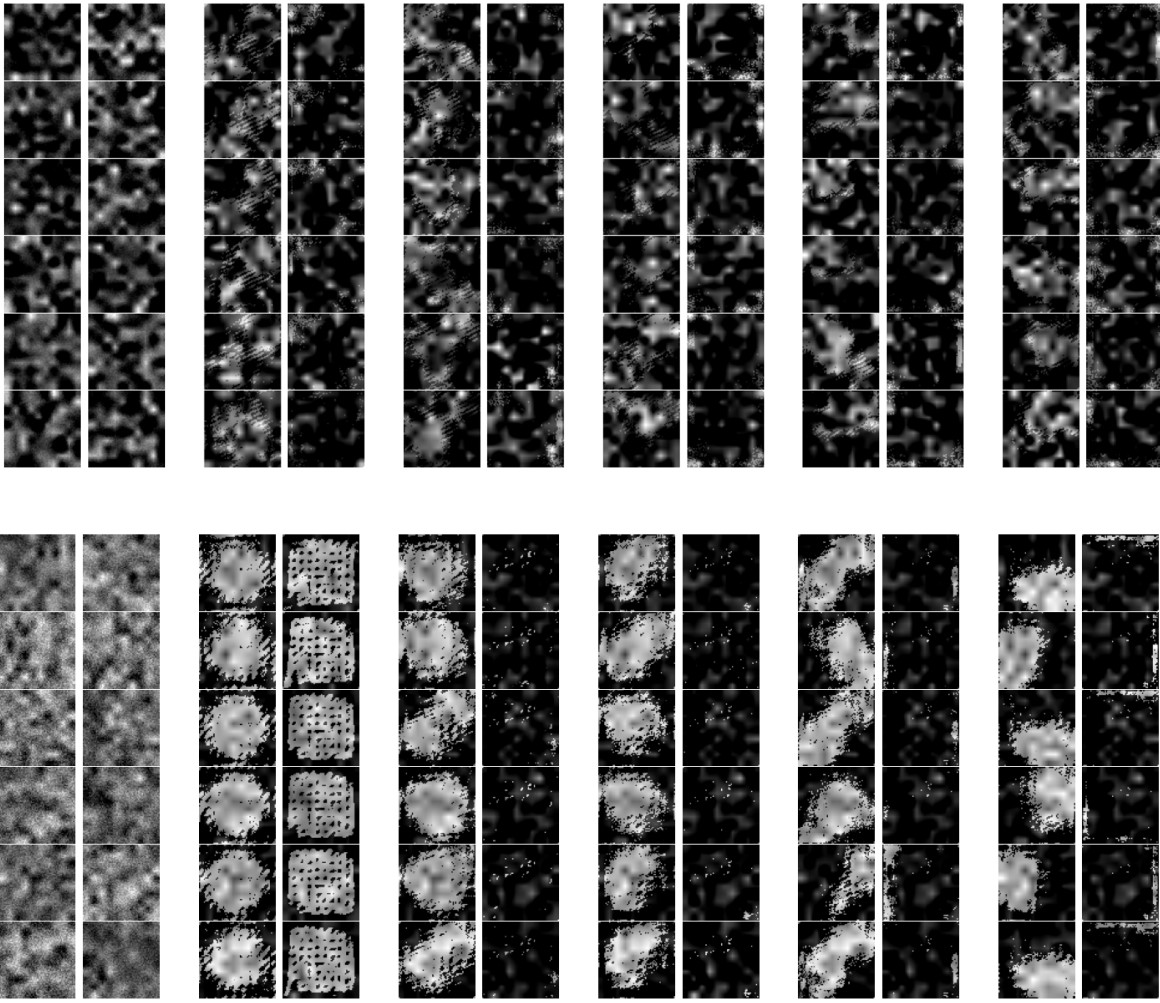

*Figure 14.* Pairs of training samples before and after 20%, 40%, 60%, 80%, and 100% of training, from left to right (lighter=more mass). Top row: actual training images; bottom row: training samples visualized with a smaller skip constant $\lambda$ to accentuate learned features.

In Table D.6, we also report the average OT distance error of samples created by the generator at various stages of training. We can see that the generator indeed creates samples that are initially difficult, but that it quickly picks up on them, and by the end of training is capable of predictions for samples from all stages of training.

*Table 5.* Relative error on OT distance for samples created after 10, 20, ..., 70% of training. Errors for all samples computed at the time of their creation (i.e., after 10, 20, ...% of training) and at the end of training.

| Error after ...% of Training | 0% | 10% | 20% | 30% | 40% | 50% | 60% | 70% |
|---|---|---|---|---|---|---|---|---|
| **At Generation** | 53.2% | 3.1% | 2.1% | 1.6% | 1.8% | 1.7% | 2.1% | 1.9% |
| **At End of Training** | 2.0% | 1.6% | 1.4% | 1.1% | 1.6% | 1.5% | 2.0% | 1.9% |

## D.7. Additional Experiments

We provide additional results from our experiments in this section. In Table 6, we show the average Wasserstein-2 distance of barycenters computed by gradient descent using equation (10) to the true barycenter, where we compute the gradient in equation (10) from the different initializations and a single Sinkhorn iteration. Figures 15 and 16 show the relative error on the OT distance over Sinkhorn iterations for $c(\boldsymbol{x}, \boldsymbol{y}) = \|\boldsymbol{x} - \boldsymbol{y}\|$ (on the square) and $c(\boldsymbol{x}, \boldsymbol{y}) = \arccos(\langle \boldsymbol{x}, \boldsymbol{y} \rangle)$ (on the sphere) resp., complementing Figure 3.

In Figure 17, we plot the relative error on the transport distance w.r.t. computation time when initializing the Sinkhorn algorithm with UNOT, and compare against the default initialization. We see that particularly on higher dimensional data, UNOT is significantly faster than Sinkhorn. However, interestingly, on MNIST the default initialization actually seems to be faster. We note that these results heavily depend on the hardware used, and that we did not optimize our FNO architecture for performance, so a more efficient architecture would probably lead to even more significant speedups. We have not included the initialization from (Thornton & Cuturi, 2022) in the plots, as it was very slow for us, even slower than the standard initialization, despite our best efforts to implement it as efficiently as possible. However, from (Thornton & Cuturi, 2022; Amos et al., 2023) it seems like the speedup should be somewhere between 1.1x and 2x, depending on the dataset, which would make it significantly slower than UNOT on most of our datasets. We mention again that FNOs process complex numbers, but PyTorch is heavily optimized for real number operations. With kernel support for complex numbers, UNOT will likely be much faster.

Finally, in Figures 18 and 19, we plot the *marginal constraint violation* (MCV), defined as

$$\frac{\left\|1_m^\top \Pi - \boldsymbol{\nu}^\top\right\|_1 + \|\Pi 1_n - \boldsymbol{\mu}\|_1}{2} \tag{20}$$

for a transport plan $\Pi$, again for a single Sinkhorn iteration (Figure 18) and over iterations (Figure 19). The MCV measures how far the transport plan is from the marginals $\boldsymbol{\mu}$ and $\boldsymbol{\nu}$. It is often used as a stopping criterion for the Sinkhorn algorithm, as the ground truth OT distance is unknown in practice. We compute the predicted transport plan for UNOT via equation (4).

*Table 6.* Average $W_2$ distance from the predicted barycenter to the true barycenter on MNIST after 100 gradient steps.

|  | $W_2$ Distance |
|---|---|
| **UNOT (Ours)** | $0.021 \pm 0.011$ |
| **Gauss** | $0.033 \pm 0.018$ |
| **Ones** | $0.057 \pm 0.034$ |

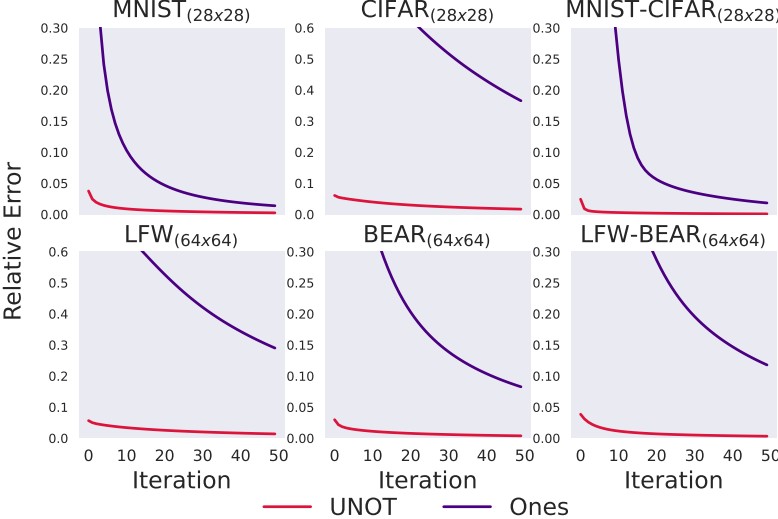

*Figure 15.* Relative Error on the OT distance on the unit square with $c(\boldsymbol{x}, \boldsymbol{y}) = \|\boldsymbol{x} - \boldsymbol{y}\|$ for the UNOT initialization compared to the default one, over number of Sinkhorn iterations. Note the y-axis has been rescaled for CIFAR and LFW to fit the curve for the default initialization, and that the Gaussian initialization does not exist for the Euclidean cost function.

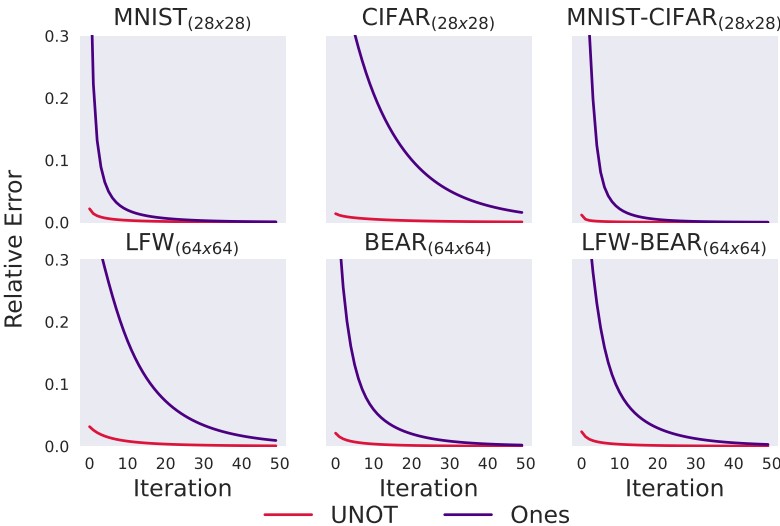

Figure 16. Relative Error on the OT distance on the unit sphere with $c(\boldsymbol{x}, \boldsymbol{y}) = \arccos(\langle \boldsymbol{x}, \boldsymbol{y} \rangle)$ for the UNOT initialization compared to the default one, over number of Sinkhorn iterations. Note that the Gaussian initialization does not exist for the spherical cost function.

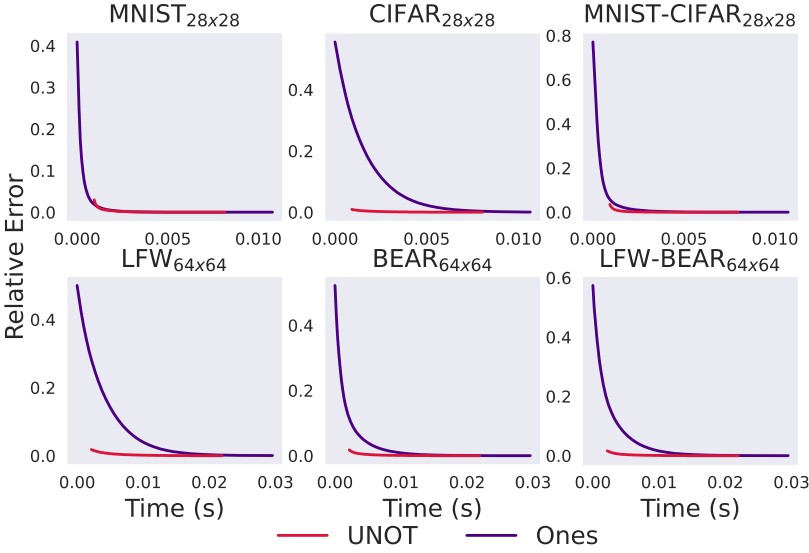

Figure 17. Comparison of relative errors on the transport distance over computation time in seconds. Evaluated on an NVIDIA 4090. The $x$-offset of the UNOT curves corresponds to the time needed for the forward pass through $S_{\boldsymbol{\phi}}$.

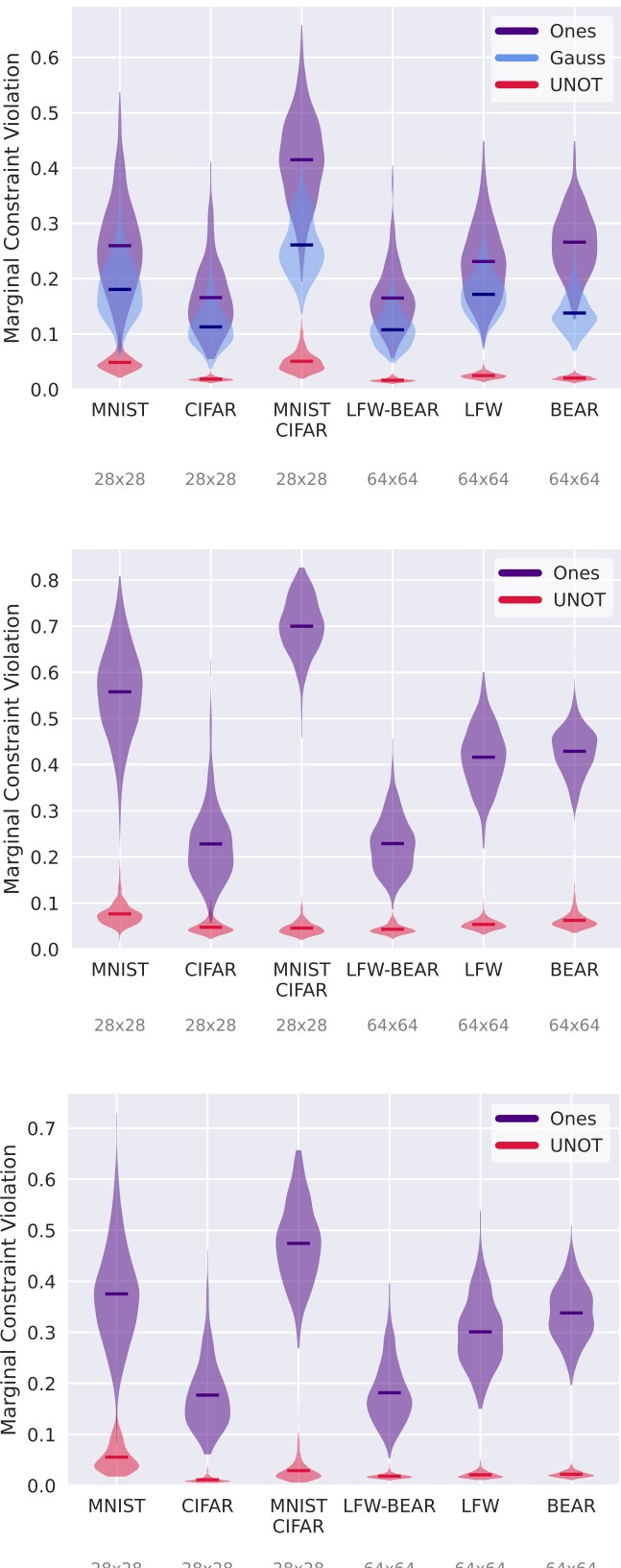

*Figure 18.* Average marginal constraint violation (see eq. (20)) after a single Sinkhorn iteration, for the unit square domain with $c(\boldsymbol{x}, \boldsymbol{y}) = \|\boldsymbol{x} - \boldsymbol{y}\|^2$ (top) and $c(\boldsymbol{x}, \boldsymbol{y}) = \|\boldsymbol{x} - \boldsymbol{y}\|$ (middle), and the unit sphere with $c(\boldsymbol{x}, \boldsymbol{y}) = \arccos(\langle \boldsymbol{x}, \boldsymbol{y} \rangle)$ (bottom). Note that the Gaussian initialization exists only for the squared Euclidean distance cost.

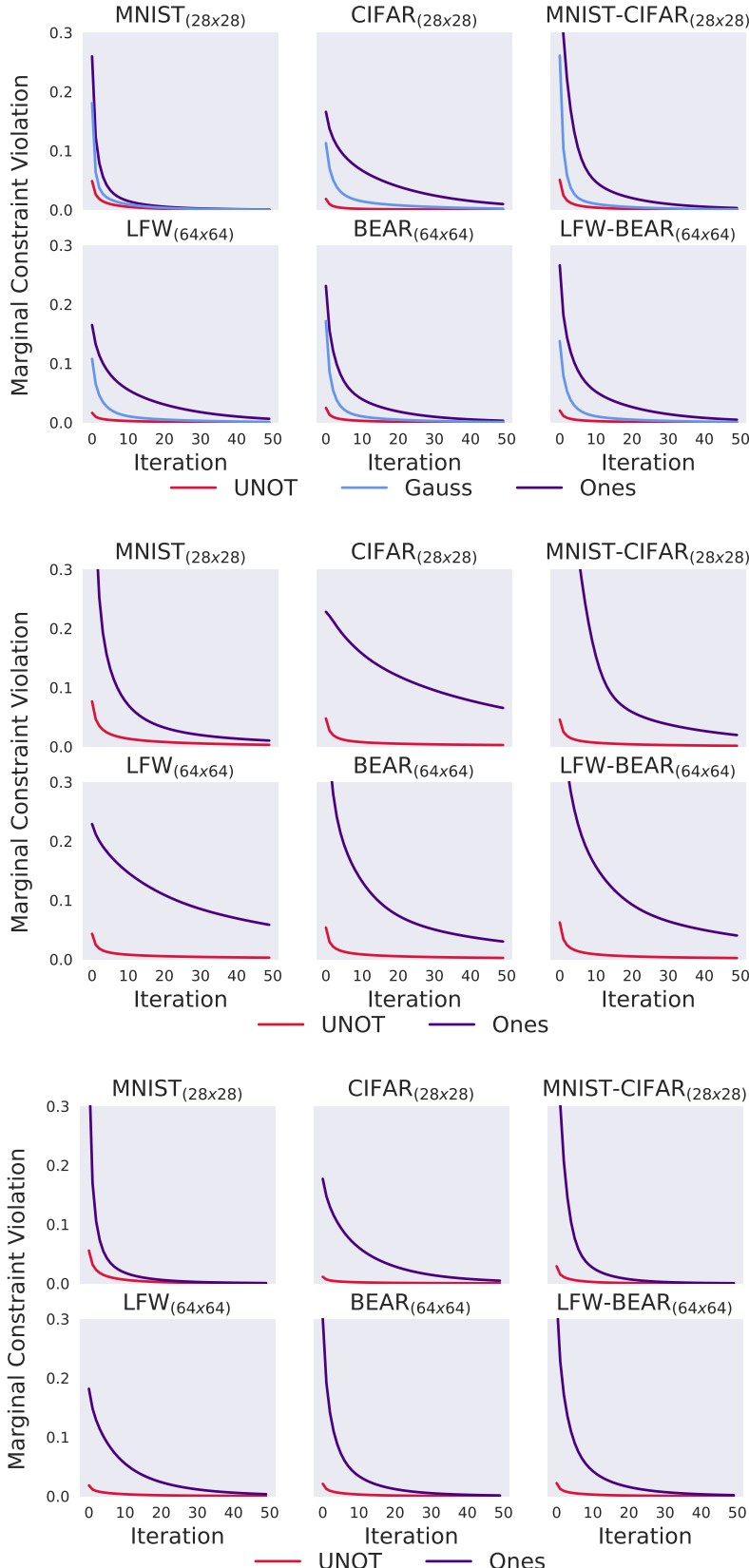

*Figure 19.* Average marginal constraint violation (see eq. (20)) over number of Sinkhorn iterations, for the unit square domain with $c(\boldsymbol{x}, \boldsymbol{y}) = \|\boldsymbol{x} - \boldsymbol{y}\|^2$ (top) and $c(\boldsymbol{x}, \boldsymbol{y}) = \|\boldsymbol{x} - \boldsymbol{y}\|$ (middle), and the unit sphere with $c(\boldsymbol{x}, \boldsymbol{y}) = \arccos(\langle \boldsymbol{x}, \boldsymbol{y} \rangle)$ (bottom). Note that the Gaussian initialization exists only for the squared Euclidean distance cost.

