# OpenReview forum: "Universal Neural Optimal Transport"
_ICML.cc/2025/Conference — ICML 2025 poster_

### Official Review · Reviewer_xg8H · 2025-03-12

**Overall Recommendation:** 2

**Summary:**

The authors introduced UNOT (Universal Neural Optimal Transport), a novel framework designed to efficiently predict entropic optimal transport distances and plans between discrete measures of varying resolutions. Motivated by the universal domain adaptation methodology, they utilized Fourier Neural Operators, which are capable of processing inputs of different sizes, allowing UNOT to generalize across diverse datasets.

**Claims And Evidence:**

The paper claims that the proposed self-supervised bootstrapping loss minimize the ground truth loss. However, the theoretical justification is somewhat vague, and while Proposition 5 suggests a relationship between minimize the bootstrapping loss and the ground truth loss, it depends on assumptions that may not hold in all scenarios.

**Essential References Not Discussed:**

The related work section covers the research area.

**Experimental Designs Or Analyses:**

The selected datasets are toy and low-dimensional, and the authors did not provide a comparison with the universal domain-adaptation methods https://openaccess.thecvf.com/content_CVPR_2019/papers/You_Universal_Domain_Adaptation_CVPR_2019_paper.pdf. The calculation of Wasserstein barycentres and their accuracy was only compared visually with the ground truth barycentres.

**Methods And Evaluation Criteria:**

The evaluation criteria is questionable. "0.01 relative error on the OT distance" is not the best metric because it is very sensitive to the preturbations. On the image domain to calculate the classic FID distance is a more reasonable choice.

**Other Comments Or Suggestions:**

None

**Other Strengths And Weaknesses:**

The model's performance may degrade with significantly higher resolutions than those encountered during training, indicating a potential scalability issue that needs to be addressed. While UNOT is claimed to generalize across datasets, the paper does not provide sufficient evidence for its effectiveness in highly diverse or out-of-distribution scenarios

**Questions For Authors:**

None

**Relation To Broader Scientific Literature:**

The paper is based on the adaptation of the well known Neural Optimal Transport framework and universal domain adaptation methods.

**Theoretical Claims:**

Theoretical claims and supporting evidence are correct and provide the necessary information about the method.

---

> ### Author Rebuttal · Authors · 2025-04-01
>
> Thank you for your thorough review! We will address your concerns in the following.
>
> >**Proposition 5 [...] depends on assumptions that may not hold**
>
> The only assumption we make in Proposition 5 is that $\sum_i g_i=\sum_i g_{\phi_i}=\sum_i g_{\tau_{k_i}}=0$. Since optimal potentials of the dual OT problem can be shifted by constant factors, we simply choose $g$ and $g_{\tau_k}$ to have zero sum. We could also shift $g_\theta$ to have zero sum in practice. However, we have updated the proof and show that the same result with the same constant holds _without_ assuming that $\sum_i g_{\phi_i}=0$, i.e. only assuming $\sum_i g_i=\sum_i g_{\tau_{k_i}}=0$.
>
>
> >**"0.01 relative error on the OT distance" is not the best metric [...] the classic FID [...] is [...] more reasonable**
>
> The purpose of UNOT is to predict OT distances with high accuracy. Hence, using the OT distance itself for evaluation seems reasonable, and is in line with other works. FID, however, is usually used to evaluate the quality of generated distributions for generative image models. Since UNOT does not generate images, we do not see how FID could be applied in our setting, but if you could explain what you mean, we are happy to include it.
>
>
> >**The selected datasets are toy and low-dimensional**
>
> While the datasets are toy datasets, we believe they are not low-dimensional. We evaluate on $64\times 64$-dimensional images, which corresponds to distributions in $4096$ dimensions (note, in particular, that in this case the cost matrix and the transport plan live in $4096^2\approx 17M$ dimensions). In comparison, MetaOT only trains and tests on $28\times 28=784$-dimensional images, more than $5$ times smaller than us. We have also run experiments on $128\times 128=16384$-dimensional data (here the cost matrix is already ~$268M$-dimensional), and the results look very promising so far, with accuracy almost recovering that of the $64\times 64$-dimensional setting. We also refer to our response to Reviewer **JPz7** regarding additional experiments with varying costs and on the unit sphere domain, and we will include real-world Euclidean and non-Euclidean test datasets in the evaluation.
>
> >**the authors did not provide a comparison with the universal domain-adaptation methods**
>
> We assume that you mention this paper because their method also features an adversarial training objective between a predictor and two discriminators (equation (4) therein). However, our objective is more closely related to that of GANs, and the similarities to the paper you link are fairly limited as far as we can tell.
>
> >**The calculation of Wasserstein barycentres and their accuracy was only compared visually**
>
> We did not include quantitative results as we thought they were difficult to interpret, but we have added them here for completeness and can add them to the camera-ready version too. We report the average $W_2$ distance from the predicted barycenter (after 1 Sinkhorn iteration) to the true barycenter on MNIST.
>
> |||
> |-|-|
> | **UNOT**|0.021 ± 0.011|
> |**Gauss**|0.033 ± 0.018|
> | **Ones**|0.057 ± 0.034|
>
>
> >**The model's performance may degrade with significantly higher resolutions than those encountered during training**
>
> We evaluate UNOT in much higher dimension than e.g. MetaOT, and we have conducted additional experiments in higher-dimensional settings ($128\times 128$) with very promising results, suggesting that UNOT can be scaled up. However, you are correct that the performance may degrade with significantly higher-dimensional inputs than those seen during training, but this is not due to an inherent limitation of our approach, but rather an impossible learning task. If a model only sees up to $n\times n$-dimensional distributions during training, it is impossible to accurately learn $(n+1)\times (n+1)$ distributions (you can always come up with two different $(n+1)\times (n+1)$ distributions that look identical when rescaled to $n\times n$). Empirically, however, UNOT performs well on inputs up to 10% larger than the largest distributions it has seen during training without loss in performance.
>
> >**While UNOT is claimed to generalize across datasets, the paper does not provide sufficient evidence for its effectiveness in highly diverse or out-of-distribution scenarios**
>
> We respectfully disagree, as we believe our test datasets are highly diverse. Not only are the individual datasets very different (compare, e.g., BEAR with low intrinsic dimension to LFW with high intrinsic dimension, cf. Figure 11), but by adding cross-datasets such as BEAR-LFW, we cover a very diverse subset of the space. Furthermore, our test datasets cover a wide range of dimensions ($28\times 28$, $48\times 48$, and $64\times 64$). UNOT performs very well on _all_ test datasets, suggesting that its generalization capabilities are strong. Also note that _all_ our test datasets are out-of-distribution, as the model does not see any of them during training.
>
> We hope this answers your questions!

---

### Official Review · Reviewer_rgYo · 2025-03-13

**Overall Recommendation:** 2

**Summary:**

This paper introduces the universal neural optimal transport (UNOT) solver, a framework designed to efficiently solve entropic-regularized optimal transport (OT) problems. Unlike existing neural OT methods that can only handle input distributions of a fixed dimension, UNOT leverages Fourier neural operators (FNOs) to predict transport plans of variable resolutions, enabling generalization across datasets and input dimensions.

## Update after rebuttal

I have raised my score after the authors fix the presentation of the theorem, but I still think the significance of Theorem 3 is limited, and the paper needs to be restructured to better highlight its novelty in other aspects.

**Claims And Evidence:**

I suspect that some of the claims in the paper are not well supported by convincing evidences, especially in the theoretical analysis. Please see the "Theoretical Claims" section below for details.

**Essential References Not Discussed:**

None.

**Experimental Designs Or Analyses:**

The experimental design seems reasonable.

**Methods And Evaluation Criteria:**

The numerical evaluation seems reasonable to me.

**Other Comments Or Suggestions:**

I think some of the notation can be made clearer and more precise. For example, in Theorem 3, the meaning of the function $\rho(x)$ is unclear, although it can be inferred that it may stand for the density function. Similarly, it should be made clear that $\mathcal{N}(\cdot|0,I)$ stands for the density function of the standard normal distribution.

**Other Strengths And Weaknesses:**

None.

**Questions For Authors:**

I suggest the author(s) checking the theoretical claims and fixing any potential flaws in the proofs.

**Relation To Broader Scientific Literature:**

The proposed UNOT method may be useful for machine learning models that involve OT computation.

**Theoretical Claims:**

I have checked the proofs of the theorems, and I feel some of the claims may be incorrect. In Theorem 3, the inequality involves the term $G_{\theta}^{-1}(x)$, and I suppose it stands for the inverse of the mapping $G_{\theta}:\mathbb{R}^d\rightarrow\mathbb{R}^d$. So basically the theorem implicitly claims that $G_{\theta}$ is invertible, but we can get a simple counterexample. By construction, $G_{\theta}$ is the output of a ReLU activation, so by carefully choosing $z$, $\mathrm{NN}\_\theta(z)+\lambda z$ can be made negative, and then the output of $G_{\theta}(z)$ would be a zero vector. By slightly perturbing the value of $z$ to $z'\neq z$, $G_{\theta}(z')$ will still be a zero vector, which shows that $G_{\theta}(z)$ is not invertible.

---

> ### Author Rebuttal · Authors · 2025-04-01
>
> Thank you for your helpful review!
>
> We will include definitions of $\rho(x)$ and $\mathcal{N}(\cdot |0,I)$ in Theorem 3, thank you for catching that. Otherwise, your main concern seems to be regarding the proof of Theorem 3. **The statement of Theorem 3 is indeed correct (if you interpret $G^{-1}(x)$ as a particular point in the preimage, see below).** However, this is only immediately clear for $x\in\mathbb{R}^d_{>0}$, as opposed to $x\in\mathbb{R}^d_{\ge 0}$ as stated in the paper; indeed, your counterexample shows that $G_\theta$ is not invertible if $x=0$ (or at least one entry in $x$ equals $0$). Hence, we could simply rewrite Theorem 3 for $x\in\mathbb{R}^d_{>0}$ instead of $x\in\mathbb{R}^d_{\ge 0}$ (on this set, $G$ is indeed invertible), which would be sufficient for our purposes as we use the Sinkhorn algorithm to generate targets from the generated samples, and the inputs to the Sinkhorn algorithm need to be positive everywhere. However, it is also straightforward to adapt the proof of Theorem 3 to work for any non-negative $x\in\mathbb{R}^d_{\ge 0}$:
>
> Let $\tilde{G}(z)=\text{NN}(z)+\lambda z$ (we drop the subscript $\theta$ for ease of notation here), i.e. $\tilde{G}$ is equal to $G$ without the ReLU activation. Denote by $\rho_{\tilde{G},\rho_z}$ the density of $\rho_z$ pushed forward by $\tilde{G}$ (this interface won't render the pushforward symbol '#' correctly, hence we slightly adapt the notation compared to the paper). Then the exact same proof as for Theorem 3, but applied to $\tilde{G}$, yields
> $$
> \rho_{\tilde{G},\rho_{z}}(x)\ge \frac{1}{(L+\lambda)^d}\mathcal{N}\left({\tilde{G}}^{-1}(x)|0,I\right)
> $$
> for _any_ $x\in\mathbb{R}^d$, where we get invertibility of $\tilde{G}$ by Theorem 1 in [1] (cmp. our proof of Theorem 3 in Appendix B). Now for any non-negative $x\in\mathbb{R}^d_{\ge 0}$, we clearly have
> $$
> \rho_{G,\rho_{z}}(x)\ge \rho_{\tilde{G},\rho_{z}}(x),
> $$
> as for any $z$ with $\tilde{G}(z)=x$, we also have $G(z)=x$. Combining these two inequalities yields
> $$
> \rho_{G,\rho_{z}}(x)\ge\frac{1}{(L+\lambda)^d}\mathcal{N}\left({\tilde{G}}^{-1}(x)|0,I\right)
> $$
> for any non-negative $x\in\mathbb{R}^d_{\ge 0}$. **This shows that the statement in Theorem 3 is indeed correct, but there could be many points in the preimage $G^{-1}(x)$, and we need to pick the unique $\tilde{G}^{-1}(x)$ amongst those for the statement to hold.**
> Thank you for pointing out this inaccuracy, and we will adapt the statement and proof accordingly in the updated version of the paper, and also update the wording of Theorem 3 to make it more clear that $\tilde{G}$ is indeed invertible under the assumptions of Theorem 3. (Corollary 4 will be updated accordingly as well.)
>
> As this seems to have been your only concern, we would kindly ask you to reconsider your score.
>
> [1] Behrmann, J., Grathwohl ,W., Chen, R.T.Q., Duvenaud, D., and Jacobsen, J.-H. Invertible Residual Networks. Proceedings of the International Conference on Machine Learning, 2019. doi:10.48550/ARXIV.1811.00995. URL https://arxiv.org/abs/1811.00995.

---

> > ### Comment · Reviewer_rgYo · 2025-04-04
> >
> > I thank the authors for the further clarification, after which I better understand what the authors want to convey in Theorem 3, and I will raise my score to reflect the fix. However, now I think the theorem is less significant. It basically says that the pushforward distribution $G_\theta\sharp\rho_z$ has positive densities on $\mathbb{R} _{\ge 0}^d$, so that it has the possibility to generate any nonnegative vector $x\in \mathbb{R} _{\ge 0}^d$, which is further used to construct discrete distributions $(\mu,\nu)$. However, this "universal generator" is trivial, since any continuous mapping from $\mathbb{R}^d$ to $\mathbb{R} _{\ge 0}^d$ results in a continuous distribution that is supported on $\mathbb{R} _{\ge 0}^d$. For example, simply take the elementwise absolute value of a normal random vector will have the same effect. I would not say the theorem is incorrect, but it just does not characterize important properties of the generator.
> >
> > ======
> >
> > I just realize that my additional reply is not visible to the authors, so I post it here.
> >
> > Thanks for the additional explanations. I recognize the various applications of UNOT, but just want to first make some of the claims precise. Back to the problem, what I mean is that if a continuous function $f$ can map the set $\mathbb{R}^d$ to $\mathbb{R} _{\ge 0}^d$, or more precisely, $f(\mathbb{R}^d)\coloneqq\\{f(x):x\in \mathbb{R}^d\\}=\mathbb{R} _{\ge 0}^d$, then $f\sharp \rho_z$ should have a positive density on $\mathbb{R} _{\ge 0}^d$. This can be shown in the following way (please correct me if there is any mistake). Let $Y=f(Z)$, where $Z\sim N(0,I)$. Suppose that there is an open set $D\subset \mathbb{R} _{\ge 0}^d$ such that the density of $Y$ is zero, then $P(Y\in D)=0$. Let $O=f^{-1}(D)=\\{z:f(z)\in D\\}$, and then by the definition of continuous functions, $O$ must be an open set, so we must have $P(Z\in O)>0$. However, by definition, $P(Z\in O)=P(Y\in D)$, so there is a contradiction. As a result, we must have that $Y$ has positive densities on every open set of $\mathbb{R} _{\ge 0}^d$.
> >
> > If I do it correctly, then it means that as long as the range of $f$ is exactly $\mathbb{R} _{\ge 0}^d$, then $f\sharp \rho_z$ meets your requirement. This is true for many neural networks, since a simple linear function with a ReLU activation suffices. The absolute value function satisfies since its range is correct, whereas the example $f(z)=0$ you mentioned does not since its range is not $\mathbb{R} _{\ge 0}^d$.
> >
> > To summarize, what I mean is that Theorem 3 is not strongly associated with the design of your generator. A huge class of neural networks would also have this property. Of course, the lower bound requires stronger conditions, but given that a normal distribution is mostly concentrated around its mean, for most areas in $\mathbb{R} _{\ge 0}^d$, the density is quite small. Hence, the lower bound is practically very close to zero on those areas.

---

> > > ### Author Response · Authors · 2025-04-04
> > >
> > > Thank you for raising your score in response to our clarification!
> > >
> > > There still seems to be some misunderstanding about the implications of Theorem 3, which we address in the following. In particular, **we argue that the theorem is not at all trivial.**
> > >
> > > > **Any continuous mapping from $\mathbb{R}^d$ to $\mathbb{R}^d_{\ge 0}$ results in a continuous distribution that is supported on $\mathbb{R}^d_{\ge 0}$. For example, simply take the elementwise absolute value of a normal random vector will have the same effect.**
> > >
> > > This statement sounds a bit misleading. While it is true that the pushforward of a Gaussian $\rho_z$ under _any_ measurable map $f$ (not necessarily continuous) from $\mathbb{R}^d$ to $\mathbb{R}^d_{\ge 0}$ is supported on $\mathbb{R}^d_{\ge 0}$ (simply by virtue of $f$ mapping into $\mathbb{R}^d_{\ge 0}$ by definition), this does _not_ imply that $f$#$\rho_z(x) > 0$ for all $x\in\mathbb{R}^d_{\ge 0}$. In the example you give ($f_i(z)=|z_i|$, i.e. the element-wise absolute value) we indeed have $f$#$\rho_z(x) > 0$ for all $x\in\mathbb{R}^d_{\ge 0}$. However, it is easy to see that this is not the case for any measurable function $f$, also not if $f$ is continuous (consider the simple case $f(z)=0$). So you are right that constructing a function $f:\mathbb{R}^d\to\mathbb{R}^d_{\ge 0}$ such that $f$#$\rho_z(x) > 0$ for all $x\in\mathbb{R}^d_{\ge 0}$ is trivial, but given a function $f$ - such as our generator $G_\theta$ - it is by no means trivial to show that the condition $f$#$\rho_z(x) > 0$ for all $x\in\mathbb{R}^d_{\ge 0}$ holds. In fact, **this will _not_ be true in general for an arbitrary (trained) neural network.** However, we prove that **our generator has this property independent of the weights $\theta$** (we also provide a specific lower bound on the density, which is stronger than just showing that it is positive everywhere), **meaning _at any stage_ in training, it can generate _any_ non-negative $x\in\mathbb{R}^d_{\ge 0}$** (and thus, any pair of discrete distributions $(\mu,\nu)$ of the right dimension). This shows that Theorem 3 states a non-trivial property of the generator.
> > >
> > > We hope this addresses your concerns about Theorem 3. As the proof and implications of Theorem 3 seem to have been your only concerns about the paper, we hope our clarifications will let you consider accepting the paper. In light of this, some details from the other rebuttals might be relevant: as you mention in your review that UNOT “may be useful for machine learning models that involve OT computation”, we refer to our reply to Reviewer **dxme**, where we highlight various potential applications of UNOT in and beyond machine learning (and also mention that we sped up our previous implementation by 2.5x). Furthermore, additional experiments in response to Reviewer **JPz7** show that UNOT works very well out-of-the-box with other costs and even on the unit sphere domain with spherical data; in the camera-ready version of the paper, we will include this setting with real-world spherical datasets.
> > >
> > > One again, thank you for taking the time to review!
> > >
> > > =====================
> > >
> > > EDIT: Thank you for updating your comment! We reply to your update below.
> > >
> > > >**[...] as long as the range of $f$ is exactly $\mathbb{R}^d_{\ge 0}$, then $f$#$\rho_z$ meets your requirement. This is true for many neural networks, since a simple linear function with a ReLU activation suffices.**
> > >
> > > Your proof that $f$#$\rho_z$ has positive density on $\mathbb{R}^d_{\ge 0}$ if $f(\mathbb{R}^d)=\mathbb{R}^d_{\ge 0}$ seems correct. However, there are two important distinctions between this statement and Theorem 3: **1)** The assumption that **$f(\mathbb{R}^d)=\mathbb{R}^d_{\ge 0}$ does _not_ hold in general for _any_ neural network, _not even for a (one-layer) linear network with ReLU_**; again, $f(z)=0$ is a (linear) counterexample, but more generally, linear functions $f:\mathbb{R}^d\to \mathbb{R}^d_{\ge 0}$ are _not_ guaranteed to fulfill $f(\mathbb{R}^d)=\mathbb{R}^d_{\ge 0}$ (as they can map to a lower-dimensional subspace of $\mathbb{R}^d_{\ge 0}$). Similarly, multi-layer non-linear neural networks are not guaranteed to fulfill this property either (and in practice, will not). Hence, this is a non-trivial property of our generator. **2)** In addition to showing that the density is positive everywhere, we also provide a specific lower bound on the density.
> > >
> > > You are correct that this lower bound can be very small in practice. The theorem still shows that the generator _can_ produce any pair of distributions during training (i.e., is not restricted by its architecture), and due to the adversarial training formulation, it will learn to generate the most useful distributions automatically (which will, in practice, not cover the entire space of course).
> > >
> > > We hope this clarifies Theorem 3, and hope that you do not have any remaining concerns for the paper. As the rebuttal period is coming to an end, once again thank you for your thorough responses!

---

### Official Review · Reviewer_dxme · 2025-03-13

**Overall Recommendation:** 3

**Summary:**

The paper presents UNOT- a method to learn universal discrete OT plans/potentials/costs approximator. Interestingly, the method may deal with different discrete resolution scales simultaneously by specific parameterization of the universal learned OT potential (through FNO operator). Several experiments on images (considered as discrete 2d distributions of different resolutions) validate the proposed methodology.

**Claims And Evidence:**

Ok

**Essential References Not Discussed:**

No

**Experimental Designs Or Analyses:**

For the problem on hand, the experimental section seems to be comprehensive enough. However, I have several comments.

**General**:
1. I do not understand the practical implication of section 4.4. Figure 7 shows that, given an image, we can transform from noise to this image. Actually, we can do the same by adding noise to the image…
2. Table 2: time for LFW: $10^{-2}$; time for BEAR: $8.0\cdot10^{-3}$. Then, time for LFW-BEAR should be ~$9\cdot10^{-3}$?

**UNOT vs. MetaOT**
1. I think the authors should add MetaOT [Amos et. al., 2022] to the comparison. Actually, this method could be used in the all provided experimental setups (except, probably, 4.1 with mixed distributions mnist + cifar/lfw+bear: but even in this case we can just learn two MetaOT models with dimensions 28x28 and 64x64)
2. Quite a strange phrase (line 318): “the relative speedup achieved in [Amos et. al., 2022]” is 1.96. But this speedup in [Amos et. al., 2022] corresponds to MNIST dataset, where your method has a deterioration compared with “ones” initialization. Also, it is interesting that your reported time for MNIST with “ones” initialization is smaller compared to [Amos et. al., 2022].

**Methods And Evaluation Criteria:**

Yes

**Other Comments Or Suggestions:**

1. lines 43-45, first column: line 44 is skipped.
2. Line 179, second column: “... any pair of discrete probability measures ($\mu$, $\nu$) can be generated by $G_{\theta}$.’’ - this is not quite right, because $G_{\theta}$ can generate only discrete distributions with dimensions $n, m$ such that $n + m \leq d’$.
3. I found it strange that no information on FNO $S_{\phi}$ is given in the main text. I think, a quick introduction of how they are constructed (in section 3.2.) would enhance the flow of the manuscript

**Other Strengths And Weaknesses:**

The paper is generally good written. However, to be honest, I do not see serious practical applications of the proposed method. How can we use UNOT? Importantly, the UNOT is trained for a fixed cost matrix. Also, I think that in the vast majority of situations, UNOT has no advantages compared to MetaOT, while being more time-consuming for training.

**Questions For Authors:**

Why do you need the generator network $G_{\theta}$, and, subsequently, adversarial max-min objective? Why not just sample $\mu$ and $\nu$ at random, treating this as Meta distribution, similar to MetaOT [Amos et. al., 2022], in eq. (6)?

**Relation To Broader Scientific Literature:**

The closest approach for UNOT is MetaOT, It would be good to have a detailed comparison with this approach (see “experimental design and analysis” section). Note on the related works section: a good work on OT in flow matching: [1], an interesting alternative approach for Neural OT: [2].

[1] Optimal Flow Matching, NeurIPS’24

[2] On amortizing convex conjugates for optimal transport, ICLR’23

**Theoretical Claims:**

I didn’t check carefully the proofs, but the results seems to be reasonable

---

> ### Author Rebuttal · Authors · 2025-04-01
>
> Thank you for your thorough and helpful review! In the following we will try to answer your questions.
>
> >**I do not understand the practical implication of section 4.4. [...] we can do the same by adding noise to the image**
>
> Section 4.4 shows that UNOT can _solve the OT problem between distributions over images_, which e.g. arises in training of generative image models. This means UNOT can a) optimally match the images in the two marginals; b) interpolate the transport along this optimal trajectory, which essentially solves the discretized Wasserstein gradient flow w.r.t. the functional $W_2^2$. Neither marginal has to be noise; we will add another figure where both marginals are image distributions. We hope this clarifies that Section 4.4 is fundamentally different from adding noise to images (adding noise does not solve OT, and in particular, cannot interpolate between two non-noisy marginals).
>
> >**Table 2: time for LFW: 10^-2; time for BEAR: 8\*10^-3. Then, time for LFW-BEAR should be ~9*10^-3?**
>
> The hardness of solving OT between two datasets (LFW and BEAR in this case) does not interpolate between the hardness of each of the individual datasets. Imagine one of the two datasets consisted only of Dirac measures, then solving OT between this dataset and any other dataset is also trivial. Hence, the time for LFW-BEAR need not equal 9*10^-3.
>
> >**I think the authors should add MetaOT [Amos et. al., 2022] to the comparison**
>
> We did not include a comparison to MetaOT as it is inherently restricted to a single dataset of fixed dimension, but for completeness, we trained MetaOT on MNIST with the official implementation (https://github.com/facebookresearch/meta-ot) and compared it against UNOT on our test datasets (rescaled to 28*28 s.t. MetaOT can process them). Here are the relative errors on the OT distance (in %):
>
> | |MNIST| CIFAR|MNIST-CIFAR|LFW|BEAR|LFW-BEAR|
> |-|-|-|-|-|-|-|
> | **UNOT** | 2.7 ± 2.4 | 1.3 ± 1.1 | 2.8 ±2.6 | 1.5 ± 1.3 | 2.0 ± 1.6 | 1.8 ± 1.3 |
> | **MetaOT**| 2.4 ± 1.8   | 23.1 ± 15.7 | 11.4 ± 5.8 | 24.6 ± 15.7 | 11.8 ± 8.3 | 31.0 ± 14.8 |
> | **Gauss** |18.1 ± 10.0|19.7 ± 7.6 | 32.2 ± 8.7 | 21.1 ± 6.5 | 20.4 ± 8.3 | 19.3 ± 6.4 |
> | **Ones** | 39.5 ± 13.4 | 47.4 ± 20.2 | 74.5 ± 6.9 | 56.9 ± 15.4 | 54.2 ± 13.5 | 66.4 ± 10.8 |
>
> Surprisingly, **UNOT is almost on par with MetaOT on MNIST, despite _never having seen a single MNIST sample during training_, whereas MetaOT was _only_ trained on MNIST**, and we see that MetaOT breaks down on all other datasets. More on UNOT vs MetaOT in our response below.
>
> >**How can we use UNOT? Importantly, the UNOT is trained for a fixed cost [...] I think that [...] UNOT has no advantages compared to MetaOT**
>
> We provide the trained UNOT model in our GitHub (which we cannot link here for anonymity reasons), so it can be used out-of-the box as a general-purpose OT solver without any additional training required. You are right that the model is trained for a fixed cost. However, the squared Euclidean cost function is the most widely used cost function in practice (it also gives rise to $W_2$). We also refer to our response to reviewer **JPz7** for experiments with other costs and on spherical domains (we will provide these models on GitHub once they are ready).
> Comparing UNOT to MetaOT, we note that MetaOT can only be trained on a single dataset of fixed dimension. This means the downstream task needs to be known in advance, and crucially, _a training dataset needs to exist for this task_. For a new task, a new model needs to be trained from scratch. UNOT, on the other hand, can be applied to any downstream dataset (for a given cost) once trained, and it even matches MetaOT in terms of performance on the MetaOT training dataset (cmp. our response above).
>
>
>
> >**[...] this speedup in [Amos et. al., 2022] corresponds to MNIST dataset, where your method has a deterioration compared with “ones” initialization. Also [...] your reported time [..] with “ones” [...] is smaller**
>
> Runtimes depend on various factors (e.g. hardware) and are difficult to compare.
> Also, our FNO uses complex (number) layers, which are known to be sub-optimally implemented in PyTorch. With faster kernels we expect a drastic speedup. Since the submission, **we have also optimized UNOT with JAX and a better architecture, achieving a 2.5x speedup over the previous version without loss in performance.**
>
> >**“... any pair of discrete probability measures $(\mu,\nu)$ can be generated by $G_\theta$." - this is not quite right**
>
> Thank you for pointing this out, we’ll improve the wording!
>
> >**I found it strange that no information on FNO $S_\phi$ is given in the main text**
>
> Yes, a more detailed description of $S_\phi$ should be given in the main text, we’ll add this!
>
> >**Why do you need the generator network $G_\theta$ [...]? Why not just sample $\mu$ and $\nu$ at random [...]?**
>
> We refer you to our response to the same question by Reviewer **JPz7**.
>
> We hope this answers all your questions!

---

> > ### Comment · Reviewer_dxme · 2025-04-02
> >
> > I thank the authors for the answers and for the work done. I will rise my score, but think the paper is exactly borderline.
> >
> > Overall, I think that the practical merits of the proposed method are overestimated.
> >
> > `UNOT can solve the OT problem between distributions over images` - This phrase is amibiguous. I understand that you may interpolate between not only noise and image, but also between image and image, but you treat each image already as a distribution. So you pick one image to be the first distribution, the second image to be the second distribution. To my understanding, it has nothing with generative image models.
> >
> > Using `fixed cost matrix` - the authors misunderstood me. Here I mean not that the authors are actually using euclidean cost function (in principle, the authors could use any cost function), but that the "grid" of support points is fixed. This prevents using UNOT (and MetaOT, too) for some nice applications, e.g., speeding-up minibatch methods for training generative models (Here I don't mean that authors mention such applications, it is just my thoughts about possible applications of such methods as UNOT and MetaOT).
> >
> > `it can be used out-of-the box as a general-purpose OT solver` - this phrase, in my view, overestimates UNOT. In particular, you can not solve generative problem between distribution of images.
> >
> > So, currently I do not see some solid applications of the method.
> >
> > Best, dxme

---

> > > ### Author Response · Authors · 2025-04-03
> > >
> > > Thank you for your response, and for raising your score! We will respond to your remaining concerns in the following.
> > >
> > > >**you pick one image to be the first distribution, the second image to be the second distribution. To my understanding, it has nothing with generative image models [...] In particular, you can not solve generative problem between distribution of images**
> > >
> > > **Solving OT between (finite) distributions of images and generative modeling are two different (albeit related) tasks.
> > > We are not claiming that UNOT can be used as a generative model.** In Section 4.4, we show that UNOT can approximate the OT plan between two distributions over images by following the Sinkhorn divergence gradient flow between the marginal distributions. In particular, we are not just randomly matching the images between the marginals and solving the OT problem on a per-image level, but we are _optimally matching the images_ of both marginals. Thus, while UNOT is applied on a per-image level, we solve the OT problem between the distributions over images. You are right that this is not generative image modeling - it is solving the (time-continuous) OT problem between the two marginals. UNOT can, however, have applications in generative modeling, e.g. in flow matching, where batches of prior samples need to be matched with data samples, and several works have explored using the OT matching [1], [2], [3], or when using OT in the loss function of generative models [4]. We provide more potential applications for UNOT below.
> > >
> > > Since Figure 7 from the paper led to ambiguity of what we are trying to show, we will add another figure where both marginals are distributions over images to make this more clear.
> > >
> > > ## Applications of UNOT
> > >
> > > Since you expressed concern about applications of UNOT, we will list some additional potential applications in the following. We consider training time applications (where UNOT could be used to guide model training) and inference time applications. Note that for training time applications, UNOT can be integrated into loss functions, as it is fully differentiable.
> > >
> > > **Neuroimaging Data (Inference Time):**
> > > OT plays an important role in the evaluation of neuroimaging data, such as in computing Wasserstein barycenters of sets of MRI scans [5]. As shown in Section 4.2, UNOT can be used to efficiently compute barycenters of images.
> > >
> > > **Remote Sensing (Inference or Training Time):**
> > > In remote sensing applications, one often compares time series data, where the OT distance can be used to inform the similarity of two time series [6]. UNOT on regular grids in one dimension could efficiently solve this task.
> > >
> > > **Climate Models on the Sphere (Inference Time):**
> > > Climate models make predictions on the sphere, and OT distances can be used in various ways, such as validating climate models through Wasserstein distances between their predictions and data [7], for which spherical UNOTs (cf. our response to Reviewer **JPz7**) could be used.
> > >
> > > **Representation Learning (Training Time):**
> > > Wasserstein barycenters can be used to adapt dictionary learning with OT [8], where barycenters can again be computed with UNOT.
> > >
> > > **Imitation Learning (Training Time):**
> > > In imitation learning, the Wasserstein distance between time series data can be used to learn to mimic an expert’s behavior [9], and UNOT could be used to efficiently estimate these Wasserstein distances.
> > >
> > > These are just some of the potential applications UNOT could be used for, but myriads of other works use OT between discrete distributions on regular grids in diverse contexts, and the above list is far from being exhaustive.
> > >
> > > We hope this addresses your concerns about the potential applications of UNOT, and thank you again for taking the time to review!
> > >
> > > ## References
> > >
> > > [1]  Lipman, Y., et al. Flow matching for generative modeling, 2023. https://arxiv.org/abs/2210.02747.
> > >
> > > [2] Pooladian, A.-A., et al. Multi Sample Flow Matching: Straightening Flows with minibatch couplings, 2023. https://arxiv.org/abs/2304.14772.
> > >
> > > [3] Tong, A., et al. Improving and generalizing flow-based generative models with minibatch optimal transport, 2024. https://arxiv.org/abs/2302.00482.
> > >
> > > [4] Genevay, A., et al. Learning Generative Models with Sinkhorn Divergences, 2017. https://arxiv.org/abs/1706.00292.
> > >
> > > [5] Gramfort, A., et al. Fast Optimal Transport Averaging of Neuroimaging Data, 2015. https://arxiv.org/pdf/1503.08596.
> > >
> > > [6] Courty, N., et al. Optimal Transport for Data Fusion in Remote Sensing, 2016. https://ieeexplore.ieee.org/document/7729925.
> > >
> > > [7] Garrett, R. C., et al. Validating Climate Models with Spherical Convolutional Wasserstein Distance, 2024. https://arxiv.org/pdf/2401.14657v1.
> > >
> > > [8] Schmitz, M., et al. Optimal transport-based dictionary learning and its application to Euclid-like Point Spread Function representation. Wavelets and Sparsity XVII, Aug 2017, San Diego, United States.
> > >
> > > [9] Dadashi, R., et al. Primal Wasserstein Imitation Learning, 2020. https://arxiv.org/pdf/2006.04678.

---

### Official Review · Reviewer_JPz7 · 2025-03-14

**Overall Recommendation:** 4

**Summary:**

The paper suggests UNOT (Universal Neural Optimal Transport) a method for (single forward-pass) prediction of OT distances, which are typically computed by iterative methods like the Sinkhorn algorithm. The network architecture is based on Fourier Neural Operators (FNOs), that provide discretization-invariance, which is useful for dealing with discrete distributions of different dimensions. Training is based on a novel GAN-like adversarial scheme, where a generating network is made to provide challenging instances, which the prediction network is optimized over using a self-supervised bootstrapping loss, theoretically proven to minimize the ground truth loss. Extensive experiments show that UNOT outperforms prior methods in several aspects of the problem, including quality of approximation, especially as an initialization to Sinkhorn iterations, but also when considering the implied geodesics and barycenters in the Wasserstein space.

**Claims And Evidence:**

Yes. The work is well motivated, since as claimed, the use of OT is constantly growing in the field of ML, and the iterative natured computations are known to be expensive, especially for large instances of the problem, for which alternative approximations (such as projections to lower-dimensional spaces, or use of closed-form OT formulations) are typically used to save time. Therefore, good approximations, or ones that can be used for starting local optimization, are of high demand.
The proposed method indeed shows excellent results, over the different datasets and usage cases, that make significant progress with respect to the goals that were set.

**Essential References Not Discussed:**

No.

**Experimental Designs Or Analyses:**

.

**Methods And Evaluation Criteria:**

The solution is well explained and its merits are demonstrated very clearly, with clear improvements across the experimentation.
The paper overall is very well written, with a very good separation between the essential parts in the main paper vs. the technicalities that are deferred to the appendix.
The solution proposed is elegant, with several new ideas that provide independent contributions. The choice of predicting the dual potential, from which everything else can be recovered, is very natural. The way that the supervision is achieved by running several Sinkhorn iterations over the prediction is a good idea, that is also justified theoretically.
There are some design choices (in both method and experimentation setup) that are well explained in the paper, but the choices (among possible alternatives) are not sufficiently discussed and justified.
One main point I wonder about is why the method is demonstrated only on the domain of images and as a consequence - How much is it suited (or limited) to this setting (where e.g. ground distances have a very particular structure).
Another is the choice to use the adversarial type training over synthetic data. Such an approach has typically stability issues in training, which are a downside of methods like GAN, where specific attention is required to avoid different failures (such as mode collapse). Additionally, it is unclear how the generated samples are related to natural image distributions and therefore how efficient the training and generalization is. Even when inspecting the generated examples - it is hard to understand why they appear as they do and how they develop during training.

**Other Comments Or Suggestions:**

I would try to clarity (and slightly expand) the description of the architecture (both the generator and FNO), since the current one is somewhat cryptic and I need to refer to the appendix for clarifications. These are main components of the solution and should be more explicit.
Regarding FNOs - a few sentences on the idea itself (lifting and operations in Fourier space) rather than only referring to the discretization invariance property. Regarding the generator - " R denotes renormalizing to two probability measures and downsampling them to random dimensions in a set range" is not very clear, even though it is correct.

Figure 3 and 13/14 - seems like you switched "Ones" and "Gauss"

Figure 5 - Is it intentional that 3 out of the 4 corner images are blurry. It would be better to interpolated between sharp images.

Figure 6 - What is "ground truth"? Only for t=1? And why are don't the interpolations coincide with the inputs at the endpoints (t=0/1)?

**Other Strengths And Weaknesses:**

.

**Questions For Authors:**

I am very positive about the paper. Given clarifications to the following questions, I might reconsider my score.
1. The focus on images. I think that it is a very good domain for explaining and demonstrating the method. But how general is the solution? Have you tested it on other domains (especially ones with different ground costs - ones that are not Euclidean, not a metric)?
2. The choice of adversarial training. Have you considered alternatives to this? Wouldn't training on the (real) images of a single particular dataset (or multiple ones) give better results (perhaps at the cost of generalization)? Did you encounter stability issues in training (e.g. dying/vanishing gradients, mode collapse) and if so what did you do to avoid them? Do the loss dynamics follow those typical in GANs?
3. Generated training distributions. The example generated pairs are nicely visualized, but I find the explanation as to their appearance and its evolution very unclear. Why does it look like it does? Is it a result of the specific architecture? Or are these actually instances that are difficult for the predictor? Does the predictor eventually solve these instances well? Does is suffer from forgetting? (e.g. solve the first examples at their time, but fail to do so later on)

**Relation To Broader Scientific Literature:**

The contributions are quite general and will have an impact on related research.
The proposed method follows a line of works that try to improve Sinkhorn initialization and sets a new standard in this respect.
Other contributions, such as the theoretical understandings that take care of the discretization issues (with the practical FNO based solution) and the bootstrapping based loss, should be of general interest for designing and training neural networks in this domain.

**Theoretical Claims:**

I didn't fully check the proofs. But the appendix does contain the necessary formulated background and proofs of all claims made in the paper.
One claim that I have not comfortably understood is regarding the universality of the generator. I understand that any input can be generated, due to the positive density of the generated distribution across the domain. But this density can be extremely non-uniform in practice and furthermore, the adversarial training strategy limits the ability to cover the space even further.

---

> ### Author Rebuttal · Authors · 2025-04-01
>
> Thank you for your thorough and positive review!
>
> The plots in Figures 3, 13, and 14 indeed got mixed up, thanks for catching! We will also include more details about FNOs and our architectures in the main text and define $R$ in the generator more clearly.
>
> **Figure 5:** the corners should not be blurry; we will fix the plot.
>
> **Figure 6:** "Ground truth" are the input images at times $t=0$ and $t=1$. Each row corresponds to a different transport plan between the ground truth images which transports the ground truth image from time $t=0$ to the predicted image at time $t=1$. This means the interpolation at time $t=0$ is equal to the ground truth by design (at $t=0$, the images should be identical, we will fix the plot), but the interpolation at time $t=1$ can be different from the ground truth if the transport plan is not accurate. This is why we also provide the ground truth for $t=1$. We hope this answers your question.
>
> >**The focus on images [...] how general is the solution? Have you tested it on other domains [...]?**
>
> We note that the model is “agnostic” to the data modality and should work with data of any modality with the same cost function and grid structure. We ran additional experiments in the Euclidean image setting with different costs ($L_1$ and $L_2$, which are metrics unlike the squared Euclidean cost $L_2^2$ from the paper), and out-of-the-box (without any hyperparameter finetuning) it matches the relative errors on the transport distance for $L_2^2$. We also ran an additional experiment with the **unit sphere in $\mathbb{R}^3$ as the domain, where we parametrize the network with Spherical FNOs** (https://arxiv.org/abs/2306.03838), and use the spherical distance $c(x,y)=\text{arccos}(<x, y,>)$ as the cost. Due to time constraints, we have only tested this on our image datasets (where we define an angular grid on the sphere, and then simply map the images onto this grid). Without any hyperparameter tuning, this works very well out-of-the-box and **even outperforms Euclidean UNOT** averaged over the test datasets. With proper hyperparameter tuning, these results can probably be even improved. For the camera-ready version, we will test UNOT on real-world spherical datasets.
>
> >**The choice of adversarial training. Have you considered alternatives to this? [...] Did you encounter stability issues in training [...]?**
>
> Initially, we tried training on synthetically generated data which works, but the adversarial approach works better. Additionally, synthetic data requires significant “fine-tuning” to capture the space of distributions well. Training on a single dataset would destroy generalization capabilities (cf. the comparison between UNOT and MetaOT in response to reviewer **dxme**). This shows that a general-purpose model requires our universal training approach. (However, it is probably possible to fine-tune a pre-trained UNOT model on downstream datasets.)
> We have not encountered any stability issues during training, and the training loss, as well as the test losses, decrease very stably during training. One stabilizing factor compared to GANs could be that our generator has a skip connection, s.t. its outputs can never completely collapse.
>
> >**I find the explanation as to [the generated pairs’] appearance [...] unclear. Why does it look like it does? [...] Does the predictor eventually solve these instances well? Does it suffer from forgetting?**
>
> Since our generator has a skip connection, part of the appearance of the distributions is indeed due to the architecture. However, the learned network in the generator ensures that training samples are generated in such a way that they are difficult for the predictor. We ran additional experiments which show that **a) generated samples are indeed difficult for the predictor initially, b) over the course of training, the predictor eventually solves them, c) forgetting of training data seems not to be happening.** Specifically, at the start of training, as well as after each 10% of training, we save a training batch and track the relative error on the OT distance on each. The following table shows the initial error on these samples as well as the final error on them at the end of training (with stable improvements on all of them during training).
>
> | Relative OT Distance Error | 0% Training | 10% Training | 20% Training | 30% Training | 40% Training | 50% Training | 60% Training | 70% Training |
> |-|-|-|-|-|-|-|-|-|
> | At Generation| 53.2%| 3.1%|2.1%|1.6%|1.8%|1.7%| 2.1%|1.9%|
> | At End of Training| 2.0%| 1.6%|1.4%|1.1%| 1.6%| 1.5%|2.0%|1.9%|
>
> We also experimented with keeping a cache of previous training samples and re-feeding them to the model over the course of training to prevent potential forgetting, similar to what is typically done when fine-tuning pre-trained models. However, this did not improve our training, probably because there does not seem to be any forgetting in the first place.
>
> We hope this answers all your questions!

---

### Decision · Program_Chairs · 2025-05-01

**Decision:**

Accept (poster)

**Comment:**

The paper proposes a one-shot prediction method for optimal transport (OT) distances called UNOT. By applying Fourier Neural Operators, the authors enable the UNOT to work with discrete distributions of varying dimensions. This method gives a state-of-the-art approximation of initial distributions for Sinkhorn iterations, thereby reducing the computational complexity of the Sinkhorn algorithm in high-dimensional cases. Moreover, the authors show that UNOT also captures Wasserstein geometry via barycenters, and approximates geodesics through both barycenters and predicted transport plans.

Regarding the reviewer JPz7, most of the questions and concerns were addressed during the rebuttal phase. JPz7 notes that such tasks as using initializations for solving OT are in demand and contributions are general, mentioning that the idea follows previous works aimed at improving Sinhorn initializations. The authors responded to all of the questions and concerns of the reviewer, including issues with domain preferences in experiments, choice of adversarial training, and the understanding of generated samples. Most of these responses were supported with solid evidence. However, I found that one questions about adversarial training and other domains were addressed without experimental evidence.

The reviewer dxme primarily highlights two issues: the lack of comparison with related work, MetaOT [Amos et. al., 2022], and the limited range of applications. The former concern was addressed by the authors with additional experiments comparing MetaOT [Amos et. al., 2022] and the proposed method. By the table, we can conclude that such a universal approach indeed works comparably, even without being trained on MNIST. Obviously, on out-of-domain datasets, MetaOT [Amos et. al., 2022] gives worse results than the proposed method. The latter concern is more substantial. The reviewer questions whether this method could be applied as the OT solver for mini-batch flow matching setups [Pooladian, A.-A., et al., Tong, A., et al.]. Despite the authors’ explanation of this use case, dxme remains unconvinced. Nevertheless, the authors included additional examples of UNOT application, which, in my opinion, adequately address the reviewer’s concerns. In this case, I believe the issue arises from the unique formulation presented in Section 3, which differs from the formulations commonly used in other articles addressing mini-batch applications of OT. As the author, I do not see a fundamental problem with the discussed application.

The third reviewer, rgYo, had only one question concerning the application of Theorem 3. Despite the response, the reviewer ended up convinced that the significance of Theorem 3 is limited. I agree to some extent that the theorem could benefit from a clearer interpretation to aid understanding. However, this appears to be more of a presentational issue rather than a flaw in the proposed method itself.

Finally, in my opinion, the authors successfully addressed all the concerns raised by the reviewers. Only minor textual/presentational issues remain, which do not seem to require substantial rewriting. Moreover, it is important to note that the issues that appeared across the reviews are mostly unique, such that both the experimental (JPz7, dxme, xg8H) and theoretical (JPz7, rgYo, xg8H) aspects of the work were comprehensively covered. Based on 4 reviews, no clear or critical problems were found that were repeatedly mentioned in all reviewers.